# Circular extrachromosomal DNA promotes tumor heterogeneity in high-risk medulloblastoma

Owen S. Chapman[1,2,3], Jens Luebeck[1,4], Sunita Sridhar[2,5], Ivy Tsz-Lo Wong [6,7], Deobrat Dixit[3,8], Shanqing Wang[4], Gino Prasad[4], Utkrisht Rajkumar[4], Meghana S. Pagadala [9,10], Jon D. Larson [3], Britney Jiayu He [11], King L. Hung [11], Joshua T. Lange[6,7], Siavash R. Dehkordi[4], Sahaana Chandran[12], Miriam Adam[13], Ling Morgan[2], Sameena Wani[3], Ashutosh Tiwari[3], Caitlin Guccione[1,2], Yingxi Lin[4], Aditi Dutta[4], Yan Yuen Lo[3,14], Edwin Juarez[2], James T. Robinson[2], Andrey Korshunov[15], John-Edward A. Michaels[16], Yoon-Jae Cho[16], Denise M. Malicki[17], Nicole G. Coufal [5], Michael L. Levy[17], Charlotte Hobbs[14], Richard H. Scheuermann[18,19], John R. Crawford[20], Scott L. Pomeroy[21,22,23], Jeremy N. Rich[24,25], Xinlian Zhang[26], Howard Y. Chang [11,27,28], Jesse R. Dixon [12], Anindya Bagchi[3], Aniruddha J. Deshpande [3], Hannah Carter [2,29], Ernest Fraenkel [13,21], Paul S. Mischel [6,7], Robert J. Wechsler-Reya [3,8], Vineet Bafna [4,29], Jill P. Mesirov [2,29,30] & Lukas Chavez [2,3,14,29,30] ✉

Circular extrachromosomal DNA (ecDNA) in patient tumors is an important driver of oncogenic gene expression, evolution of drug resistance and poor patient outcomes. Applying computational methods for the detection and reconstruction of ecDNA across a retrospective cohort of 481 medulloblastoma tumors from 465 patients, we identify circular ecDNA in 82 patients (18%). Patients with ecDNA-positive medulloblastoma were more than twice as likely to relapse and three times as likely to die within 5 years of diagnosis. A subset of tumors harbored multiple ecDNA lineages, each containing distinct amplified oncogenes. Multimodal sequencing, imaging and CRISPR inhibition experiments in medulloblastoma models reveal intratumoral heterogeneity of ecDNA copy number per cell and frequent putative 'enhancer rewiring' events on ecDNA. This study reveals the frequency and diversity of ecDNA in medulloblastoma, stratified into molecular subgroups, and suggests copy number heterogeneity and enhancer rewiring as oncogenic features of ecDNA.

Circular ecDNA molecules, also known as double minutes, have been described in isolated tumor and tumor-derived cells since the 1960s (ref. 1). Recent results have shown ecDNA to be far more common in human cancer than previously assumed[2,3]. Commonly defined as circular, acentric chromatin bodies tens of kilobases to tens of megabase pairs (Mbp) in length, circular ecDNA is now understood to be a major contributor to intratumoral heterogeneity and is implicated in oncogenesis, tumor evolution and the evolution of drug resistance[4–7].

Circular ecDNA is a frequent form of high-copy oncogene amplification[3] and a prognostic biomarker in many tumor types[8–10], and it allows amplified oncogenes to 'hijack' noncoding regulatory enhancers that would be inaccessible under normal karyotypic topology[11–13].

Medulloblastomas were represented among the first patient case reports describing ecDNA[1]. Few effective targeted molecular treatments exist for medulloblastoma, and the current standard of care carries a substantial risk of cognitive disorders, neurological damage and secondary malignancy[14]. There are four major molecular subgroups of medulloblastoma: WNT, Sonic hedgehog (SHH), Group 3 and Group 4 (ref. 15). Prognosis is especially poor for a subset of MYC-activated Group 3 tumors and for *TP53*-mutant SHH subgroup tumors[16–19]. The mutational landscape of medulloblastoma subgroups has recently been characterized[18]; however, the frequency of ecDNA in the different molecular medulloblastoma subgroups, the amplified genomic regions and their impact on patient outcomes are not well understood. Furthermore, the contribution of ecDNA to intertumoral and intratumoral heterogeneity as well as the potential role for enhancer hijacking by ecDNA in medulloblastoma remain open questions. Here, we resolve ecDNA content and structure using next-generation sequencing, optical mapping, CRISPR-CATCH and microscopy of ecDNA in medulloblastoma cells. We estimate intratumoral heterogeneity using computational approaches applied to microscopy and single-cell sequencing data. We perform epigenetic profiling to examine the transcriptional regulatory circuitry of ecDNA sequences and interrogate functional transcriptional enhancers on an ecDNA using CRISPRi. Our results demonstrate that ecDNA confers shorter survival for a subset of patients with medulloblastoma and illuminate molecular roles for ecDNA in medulloblastoma pathogenesis.

## ecDNA amplifies medulloblastoma oncogenes

To examine the landscape of ecDNA in medulloblastoma, we accessed whole genome sequencing (WGS) data of tumors available in three cloud cancer genomics platforms[20–22]. In addition, we included 43 tumors from a previous proteomic analysis[23] and 8 tumors diagnosed at the Rady Children's Hospital, San Diego. In total, our retrospective cohort comprised 481 tumor biopsies from 468 patients. Using DNA fingerprint analysis, we ensured that the combined cohort contained no duplicates. Clinical metadata were available for most patients and included age at diagnosis, sex, medulloblastoma molecular subgroup and survival (Supplementary Tables 1–4). Using the AmpliconArchitect algorithm[24], we detected 102 putative ecDNA sequences in tumor samples from 82 out of 468 (18%) patients. By molecular subgroup, patients with ecDNA-positive (ecDNA+) tumors were distributed as follows: WNT, 0 out of 22; SHH, 30 out of 112 (27%); Group 3, 19 out of 107 (18%); and Group 4, 26 out of 181 (14%) (Fig. 1a). SHH subgroup tumors were significantly more likely to contain ecDNA than tumors from the other medulloblastoma subgroups ($\chi^2 = 7.66$, $P = 0.006$). Among the ecDNA-amplified genes occurring in two or more samples in this cohort were known or suspected medulloblastoma oncogenes *MYC*, *MYCN*, *MYCL*, *TERT*, *GLI2*, *CCND2* (ref. 25), *PPM1D* (WIP1) (ref. 26) and *ACVR2B* (ref. 27); genes encoding DNA repair machinery (*RAD51AP1* and *RAD21*); and genes encoding *TP53* pathway inhibitors (*PPM1D*[28] and *CDK6* (ref. 29)) (Fig. 1b). Of *MYC* oncogene family amplifications, 19 out of 23 *MYCN*, 11 out of 18 *MYC* and 3 out of 3 *MYCL1* were on ecDNA, as were all amplifications of *CCND2*, *GLI2* and *TERT*.

## ecDNA predicts poor prognosis in medulloblastoma

To evaluate ecDNA as a potential prognostic marker in medulloblastoma, we performed survival analyses across patients for whom clinical metadata were available. Patients with ecDNA+ tumors had significantly worse overall and progression-free five-year survival compared to patients with ecDNA-negative (ecDNA−) tumors (log-rank test, $P < 0.005$; Fig. 1c and Extended Data Fig. 1a). Stratified by molecular subgroup, patients with ecDNA+ tumors had worse overall survival in the SHH, Group 3 and Group 4 subgroups ($P < 0.05$; Fig. 1d–f and Extended Data Fig. 1b–d). Survival of patients in the WNT subgroup was not analyzed because no WNT tumors in our patient cohort were ecDNA+. To determine whether patients with ecDNA+ tumors had worse outcomes than patients with tumors harboring other types of focal somatic copy number amplification, we stratified patients by the topology of the amplification(s) present in the tumor genomes[3]. As expected, patients with ecDNA+ tumors had the poorest outcomes, significantly ($P < 0.005$) worse than patients without focal somatic copy number amplification or with linear amplifications (Extended Data Fig. 2).

To further estimate the prognostic value of ecDNA, we conducted Cox proportional hazards regressions, controlling for sex, age and molecular subgroup. Patients with ecDNA+ tumors were at greater estimated risk for progression (hazard ratio, 2.36; $P < 0.005$) and mortality (hazard ratio, 2.99; $P < 0.005$) than patients with ecDNA− tumors (Fig. 1g, Extended Data Fig. 1e and Supplementary Tables 5 and 6).

## *TP53* alterations are associated with ecDNA in SHH medulloblastoma tumors

The tumor suppressor protein p53 (encoded by *TP53*) regulates DNA damage sensing and cell cycle arrest and apoptosis, and is frequently affected by somatic mutations and pathogenic germline variants in SHH medulloblastoma[19,30,31]. Moreover, SHH medulloblastomas with inactivating *TP53* mutations are known to be associated with chromothripsis[17], the catastrophic shattering of a chromosome that precedes ecDNA formation in some cell line models[32,33]. To test whether *TP53* mutations were associated with the presence of ecDNA, we accessed somatic and germline *TP53* mutation status of 92 SHH medulloblastomas. *TP53* alterations were enriched in ecDNA+ SHH subgroup tumors (12 out of 23, 52%) compared to the ecDNA− SHH subgroup (2 out of 69, 3%; Fisher exact test, $P = 1.3 \times 10^{-7}$). We did not find a significant association between *TP53* alterations and ecDNA in the other subgroups or across the entire cohort, suggesting that in medulloblastoma, a possible functional relationship between *TP53* alterations and ecDNA is restricted to the SHH subgroup. We reasoned that the established effect of *TP53* mutation on the survival of patients with medulloblastoma[34] may be mediated, at least partially, by ecDNA (Extended Data Fig. 3). To test this hypothesis, we conducted mediation analysis using the Baron–Kenny approach[35]. Accelerated failure time (AFT) regressions of progression-free survival on *TP53* mutation and ecDNA status suggest that much of the effect of *TP53* mutation on prognosis can be explained by an effect of ecDNA and by the frequent co-occurrence of ecDNA in *TP53*-mutant tumors (Supplementary Tables 7 and 8 and Supplementary Note 1).

To evaluate whether there is a *TP53*-independent effect of ecDNA on survival, we performed Cox regression, including *TP53* alteration as a covariate and controlling for collinearity. The effect of ecDNA on survival remains significant but diminished when we include *TP53* alteration as a covariate in our Cox models (hazard ratio for progression-free survival, 1.87, $P = 0.01$; hazard ratio for overall survival, 2.32, $P < 0.005$; Extended Data Fig. 4 and Supplementary Tables 9 and 10), indicating that there is an effect of ecDNA on survival that cannot be explained by *TP53* mutation alone. Such an effect may be explainable by a *TP53*-independent mechanism of ecDNA formation or by inactivation of the *TP53* pathway by other means, such as *CDKN2A* deletion or *PPM1D*, *CDK6*, *MDM4* or *MDM2* amplification[36]. In our patient cohort, we observe nine such amplifications on ecDNA across all subgroups (Fig. 1b). Although causality cannot be inferred from these data alone, these survival analyses identify *TP53* alteration and ecDNA as clinically relevant biomarkers for a subset of highly aggressive SHH medulloblastoma tumors.

## Multiple ecDNA lineages coexist in some medulloblastomas

Our patient cohort included 16 medulloblastoma tumors with multiple distinct ecDNA sequences (Supplementary Table 11). This set

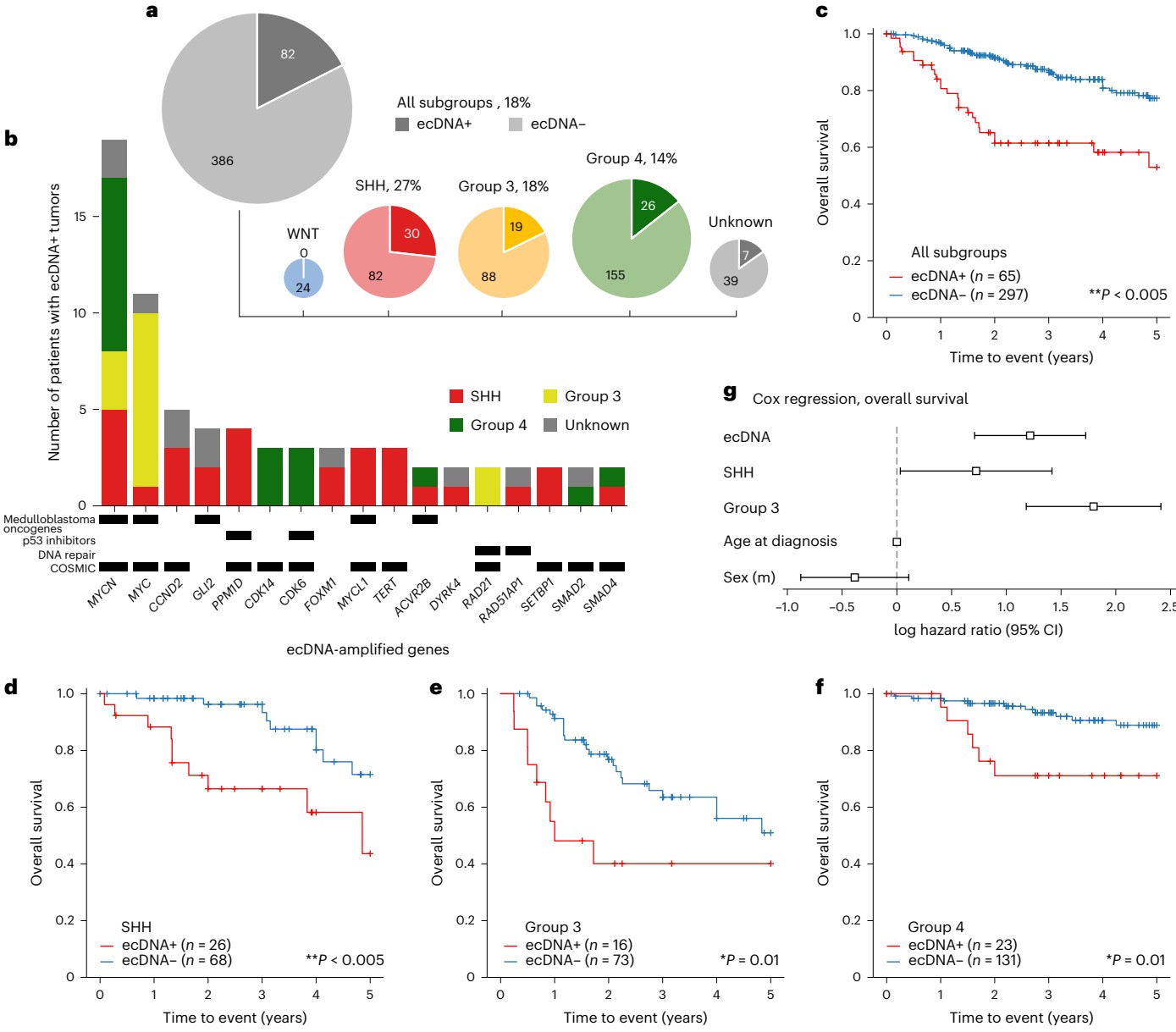

**Fig. 1 | The landscape of ecDNA in medulloblastoma patient tumors.**
**a**, Presence of ecDNA by molecular subgroup across 468 tumors from patients with medulloblastoma. **b**, A subset of recurrently ($n \geq 2$) amplified genes on ecDNA in this patient cohort. p53 inhibitors: negative regulators of p53 pathway activity; COSMIC: genes listed as Tier 1 or Tier 2 of the COSMIC Cancer Gene Census[61]. **c**, Kaplan–Meier curve depicting five-year overall survival in the patient cohort stratified by the presence of ecDNA in tumors. $P = 8.6 \times 10^{-6}$. **d**–**f**, Kaplan–Meier curves indicating overall survival for SHH ($P = 4.8 \times 10^{-3}$) (**d**), Group 3 ($P = 0.01$) (**e**) and Group 4 ($P = 0.01$) (**f**) subgroups, stratified by ecDNA presence. $P$ values for **c**–**f** were derived from two-sided log-rank test without correction for multiple hypotheses. **g**, Log hazard ratios for ecDNA status, medulloblastoma subgroup, age and sex estimated by Cox regression on overall survival. Sample size was $n = 352$ observations. Data are presented as maximum likelihood estimate (MLE) ±95% confidence intervals.

included a SHH medulloblastoma primary tumor with heterozygous somatic *TP53* mutation[37] (RCMB56-ht), which we established as an orthotopic patient-derived xenograft mouse model (RCMB56-pdx). Analysis of WGS data from RCMB56-ht predicted two distinct focal amplifications: a circular ecDNA of length 3.2 Mbp comprising three regions of chromosome 1 (amp1; Supplementary Fig. 1) and a complex, possibly chromothriptic, 4.5 Mbp amplicon comprising 20 segments from chromosome 7 and one segment from chromomsome 17, with ends mapping to pericentromeric and peritelomeric regions (amp2; Supplementary Fig. 2). Similar analysis of RCMB56-pdx confirmed that both focal amplifications were unchanged compared to the original primary human tumor. Sequencing depth of the WGS data also indicated low-copy gain (gain1) of unknown architecture

composed of other segments of chromosome 7 (35 Mbp) and chromosome 17 (800 kbp).

To assemble high-confidence sequences for the two amplicons, we performed optical genome mapping (OGM) of RCMB56-pdx. Genome assembly from deep WGS and OGM validated the circular amp1, composed of three DNA segments from chromosome 1 (Fig. 2a). This analysis also validated the contiguous chromothriptic amp2, comprising 21 segments of chromosome 7 and chromosome 17; however, a circular structure could not be conclusively established from OGM and WGS data (Fig. 2b). Copy number of amp1 and amp2 was estimated from WGS data at 20 and 10, respectively in RCMB56-ht, and 30 and 25, respectively in RCMB56-pdx. DNA fluorescence in situ hybridization (FISH) imaging of metaphase cells for marker gene loci *DNTTIP2* (amp1),

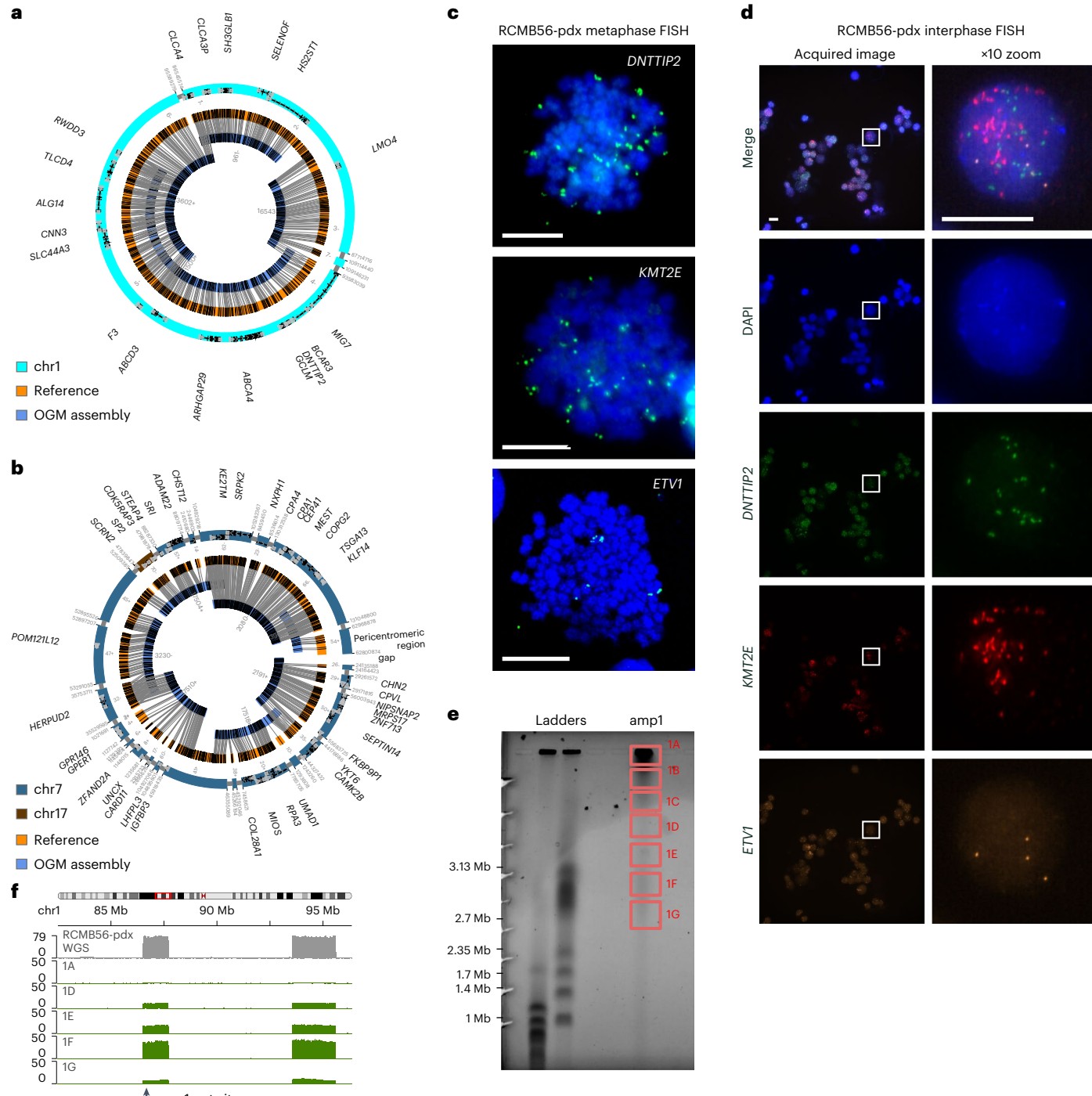

**Fig. 2 | Distinct high-copy extrachromosomal amplifications coexist in a SHH medulloblastoma tumor. a**, Assembly of the high-copy focal amplification amp1 from WGS and OGM of DNA from the RCMB56-pdx tumor. All breakpoints in the assembly are supported by both data types. **b**, Assembly of the high-copy focal amplification amp2 from the same data. All junctions are supported except a peritelomeric region adjacent to the gap, which was inferred from WGS discordant reads only. **c**, Metaphase FISH microscopy targeting amplified *DNTTIP2* (amp1), *KMT2E* (amp2) and *ETV1* (gain1). Images are representative of 12, 18 and 5 stained metaphase cells, respectively. Spots outside the chromosome boundaries (blue) indicate extrachromosomal amplifications. **d**, Interphase FISH microscopy for the same markers confirms co-amplification of all three genes on distinct amplicons. Representative image from a series of 27 images. Boxes (left) are magnified (right). Scale bars in **c**–**d**, 10 μm. **e**, Pulsed-field electrophoresis gel of DNA from RCMB56-pdx, generated by CRISPR-CATCH. DNA was cut using sgRNA targeting amp1, then fractionated by size through the gel. **f**, Sequencing coverage of the fractions indicated in **e** at the amp1 locus. Gray, bulk WGS; green, sgRNA targeting amp1. Amp1 is most enriched in band 1F, consistent with its 3.2 Mbp assembly length.

*KMT2E* (amp2) and *ETV1* (gain1) indicated that amp1 and amp2 are amplified extrachromosomally (Fig. 2c). To confirm co-occurrence in the same cells, we performed multi-channel FISH imaging for the same markers in interphase cells. We observed distinct fluorescence spots for each gene within the same nucleus, indicating that copies of each amplified gene are located on distinct chromatin bodies (Fig. 2d). To further validate the predicted circular amp1 assembly, we used a recent method for targeted profiling of ecDNA, CRISPR-CATCH[38]. As expected,

cutting amp1 in DNA from RCMB56-pdx produced a single fraction of linear DNA matching the length of the amp1 assembly (Fig. 2e). Short read sequencing maps this DNA to the amp1 sequence identified from bulk sequencing, confirming its circular structure (Fig. 2f).

## Medulloblastomas have heterogeneous ecDNA copy number

Substantial intratumoral copy number heterogeneity is expected in ecDNA+ tumors owing to random segregation of ecDNA during mitosis, driving tumor evolution and treatment resistance[39]. To quantify copy number heterogeneity of ecDNA in medulloblastoma, we established an automated image analysis pipeline to estimate the distributions of copy number per cell in interphase FISH microscopy imaging and applied it to four primary medulloblastoma tumors harboring ecDNA: MB036 (*MYCN*), MB177 (*MYCN*), MB268 (*MDM4*) and RCMB56 (*DNTTIP2*, *KMT2E*, *ETV1*). The estimated copy number per cell of all ecDNA-amplified marker genes had significantly greater mean (Wilcoxon test) and variance (Levene's test) than the ecDNA− cell line COLO320-HSR (Fig. 3a,b and Supplementary Fig. 3), which includes the *MYC* locus on a chromosomal amplification[40]. These results from human medulloblastoma tumors are consistent with the high copy number heterogeneity observed in human cancer cell lines with ecDNA[39]. In each primary tumor analyzed, ecDNA was amplified (copy number greater than five) in only a subset of cells (22–41%; Supplementary Tables 12–18).

To determine whether copy number heterogeneity of an ecDNA+ tumor is accompanied by transcriptional heterogeneity, we analyzed 2,762 single nuclei from frozen tissue of RCMB56-ht using a single nuclei multiome RNA (snRNA) and assay for transposase-accessible chromatin with sequencing (snATAC-seq) assay (10x Genomics) to profile transcriptomes and accessible chromatin of the same individual cells. Consistent with previous findings in bulk samples[2,11], RCMB56-ht snATAC-seq coverage was enriched at the amp1 and amp2 loci at the aggregate level and in individual cells (Fig. 3c). To detect focal amplifications in single nuclei, we performed Monte Carlo permutation tests comparing snATAC-seq read density at the amplicon locus to those at random locations elsewhere in the genome. Z-score normalized read density at the amp1 and amp2 loci had greater mean and variance than at gain1 (Fig. 3d), consistent with our observations of the interphase FISH data. We conservatively estimate that at least 224 out of 2,762 (8%, false discovery rate $q < 0.10$) cells contained amp1 or amp2 (ecDNA+ cells). Of these, both amp1 and amp2 were detected together in only a minority of cells (72 out of 224, 32%) (Fig. 3e). Thus, evidence from quantitative FISH microscopy and multiome single-cell sequencing show that only a fraction of tumor cells in ecDNA+ medulloblastoma tumors harbor high-copy ecDNA and that these have highly variable copy numbers of single or multiple different extrachromosomal amplifications.

## ecDNA+ cells have distinct transcriptional profiles

Clustering single cells using the weighted nearest neighbors algorithm[41] placed the majority of ecDNA+ cells in a single cluster with distinct transcriptional and epigenetic features (Fig. 3f and Extended Data Fig. 5a). As expected, cells in the ecDNA+ cluster overexpressed *DNTTIP2* (Wilcoxon rank sum test, $q < 0.001$) and *KMT2E* ($q < 0.001$), the marker genes for amp1 and amp2. Compared with other tumor and normal cells, the ecDNA+ cell cluster also overexpressed *GLI2* ($q < 0.001$), a mediator of SHH-mediated transcription and marker for SHH medulloblastoma, despite *GLI2* not being affected by copy number alteration in this tumor (Fig. 3g and Supplementary Table 19). To further investigate the relationship between ecDNA copy number and transcription, we first estimated ecDNA copy number in single cells ($z$-scores) and then the transcriptional activity of genes amplified on ecDNA in each cell (ssGSEA[42] scores, see Methods). As expected, ssGSEA scores were positively correlated with $z$-scores, indicating greater transcription of ecDNA-amplified genes with increasing ecDNA copy number (Extended

Data Fig. 5b–e). In addition to the ecDNA+ tumor cells, we identified two other clusters of tumor cells that were not enriched for ecDNA and with low expression of the marker genes, one of which strongly expressed mitochondrial genes (labeled 'ecDNA−' and 'ecDNA− MT high'), as well as normal cells such as astrocytes, oligodendrocytes and hematopoietic cells (Fig. 3f,g). Normal cell types were annotated by cluster-specific expression of known marker genes. Genomic copy number estimation from snRNA-seq confirmed that normal cells had stable genomes whereas tumor cell clusters harbored various copy-number alterations (Extended Data Fig. 5f).

## ecDNA places oncogenes in ectopic gene regulatory contexts

It has been shown that some medulloblastoma tumors are driven by 'enhancer hijacking' events, whereby somatic structural variants cause a noncoding regulatory enhancer to be rewired to amplify oncogenic transcription[18,43]. Given the extensive genomic rearrangement associated with some medulloblastoma ecDNA, we investigated whether aberrant DNA interactions emerge on circular ecDNA between co-amplified oncogenes and enhancers. To test this hypothesis, we profiled the accessible chromatin of 25 medulloblastoma tumors (11 ecDNA+, 14 ecDNA−) using ATAC-seq[44], as well as chromatin interactions of 17 medulloblastoma tumors (eight ecDNA+, nine ecDNA−) using chromatin conformation capture (Hi-C)[45]. Consistent with previous reports[11,46], bulk ATAC-seq read density was markedly enriched across entire ecDNA regions, even for ecDNA with only low-level amplification as estimated by bulk WGS. Hi-C sequencing reads exhibited similar patterns of enrichment at ecDNA regions (Fig. 4a).

In half of the analyzed ecDNA+ tumors (D458, MB106, MB268 and RCMB56), we observed clear evidence of aberrant chromatin interactions on ecDNA that spanned structural variant breakpoints to juxtapose accessible loci and co-amplified genes from distal genomic regions. For example, in the ecDNA+ Group 3 primary tumor MB106, DNA interactions occurred between the *MYC* locus and two co-amplified accessible regions 13 Mbp away on the reference genome, but less than 1 Mbp away on the ecDNA (Fig. 4b,c). These chromatin interactions were specific to the MB106 ecDNA compared to the interactome of the ecDNA− Group 3 primary tumor MB288 (Fig. 4c).

In the SHH subgroup primary tumor MB268, we identified a 10.2 Mbp ecDNA amplification including the p53 regulator *MDM4* (ref. 47) (Extended Data Fig. 6). *MDM4* is recurrently amplified on glioblastoma ecDNA[24] and is a putative driver event in MB268. On the same ecDNA, we also observed aberrant DNA interactions with the immune complement system regulator *CFH* promoter. However, the functional significance of these co-amplified genes and DNA interactions remains unclear.

In two instances, the SHH subgroup tumor RCMB56-pdx and the Group 3 cell line D458, we identified rewired interactions between genomic loci originating from different chromosomes but co-amplified on the same ecDNA. As described above, RCMB56 harbored an ecDNA comprising segments of chromosome 1 and a complex extrachromosomal amplification comprising segments of chromosome 7 and chromosome 17. Hi-C data indicated frequent chromatin interaction across breakpoints in each of the two amplicons (Extended Data Fig. 7). Aberrant chromatin interactions mapping to amp1 targeted accessible regions at the *DNTTIP2*, *SH3GLB1* and *SELENOF* gene loci (Extended Data Fig. 7a–c). Aberrant interactions on amp2 included intrachromosomal interactions mapping to *RPA3*, *HERPUD2*, *KLF14* and others; and trans-chromosomal interactions between the *SP2* locus and the brain-specific long noncoding RNA *LINC03013* (ref. 48), and from the *PRR15L* promoter to an intragenic region upstream of *SRI* (Extended Data Fig. 7d–f).

D458 harbored an ecDNA amplification containing oncogenes *MYC* and *OTX2* from chromosomes 8 and 14, respectively. Co-amplification of *MYC* and *OTX2* on the same ecDNA was validated by confocal FISH

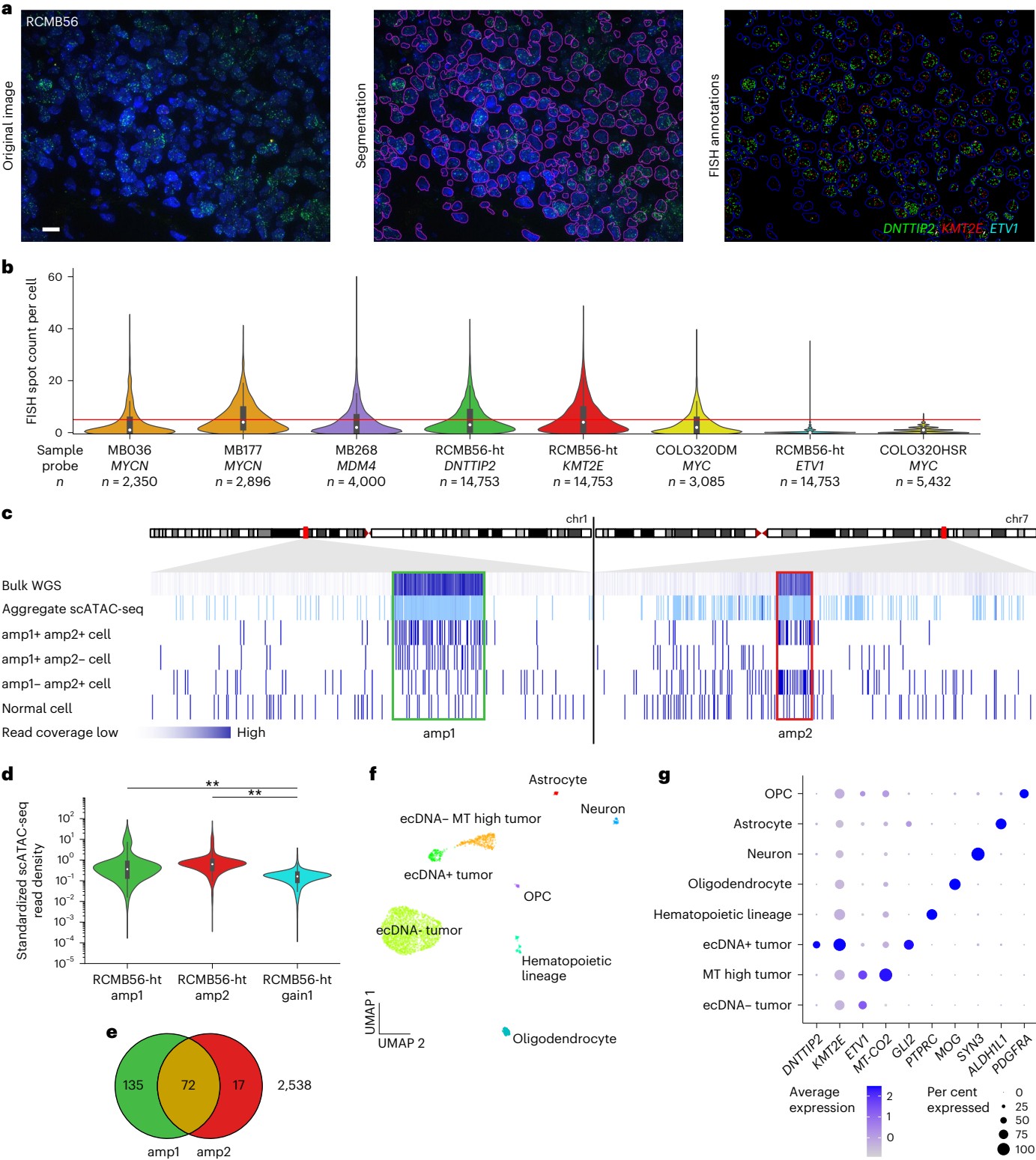

**Fig. 3 | Single-cell analysis reveals a distinct tumor cell population with high-copy ecDNA amplification. a**, Quantitative FISH image analysis of formalin-fixed paraffin-embedded (FFPE) tissue of the SHH medulloblastoma tumor RCMB56-ht. Representative image of 45 regions of one FFPE tissue slide. Scale bar, 10 μm. **b**, Distributions of FISH spot count per cell for amplified marker genes. MB036, MB177 and MB268 are SHH, Group 4 and SHH subgroup primary tumors[23]. COLO320DM and COLO32HSR are positive and negative controls with isogenic extrachromosomal or intrachromosomal *MYC* amplifications, respectively[39]. Red line indicates spot count = 5, the threshold used to classify amplified cells. Bar centers represent medians; bars indicate the interquartile range (IQR); and the whiskers extend to Q3 + 1.5 × IQR and Q1 − 1.5 × IQR. **c**, Read coverage at amp1

and amp2 loci in RCMB56-ht using various sequencing modalities. Each track is scaled independently. **d**, Standardized snATAC-seq read depth (z-scores) at the amp1, amp2 and gain1 regions of n = 2,762 RCMB56-ht cells. Bar centers represent medians; bars indicate the IQR; and the whiskers extend to Q3 + 1.5 × IQR and Q1 − 1.5 × IQR. Two-sided Mann–Whitney test, **P < 0.005. **e**, Number of cells in RCMB56-ht with significantly enriched read depth of amp1, amp2 or both. **f**, Uniform manifold approximation and projection (UMAP) projection of cell clusters detected in RCMB56-ht snRNA + ATAC-seq data using weighted nearest neighbors clustering. Cell clusters have been labeled based on overexpression of cell type-specific genes. **g**, Expression of marker genes across cell clusters of RCMB56-ht. OPC, oligodendrocyte precursor cell.

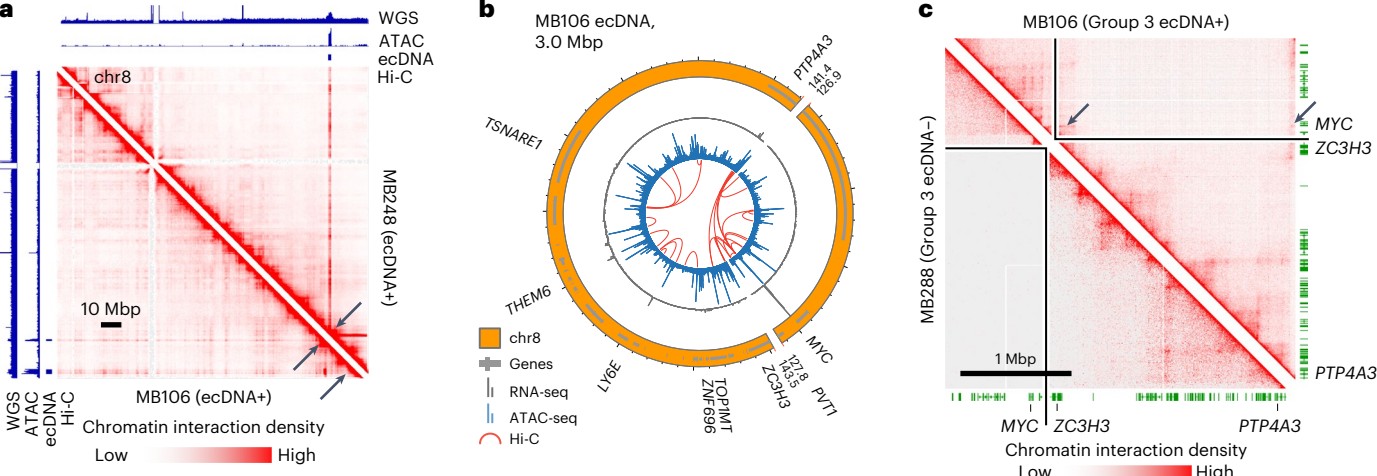

**Fig. 4 | Chromatin interactions with MYC are rewired in a Group 3 medulloblastoma. a**, WGS, ATAC-seq and Hi-C read coverage of chromosome 8 in *MYC*-amplified primary tumors MB248 (top right) and MB106 (bottom left). Arrows indicate low-copy ecDNA amplifications of *MYC* in both samples. Genomic tracks scaled independently. **b**, Reconstruction of the MB106 ecDNA from WGS. Tracks (outer to inner): genome sequence, transcriptome (RNA-seq), chromatin accessibility (ATAC-seq), chromatin interactions (Hi-C). **c**, The Hi-C interactome of MB106 ecDNA (top right) contains enhancer–promoter interactions (arrows) not visible in an unrearranged medulloblastoma genome (bottom left).

(Fig. 5a) and by assembly of the D458 ecDNA from WGS and OGM data (Extended Data Fig. 8a). *OTX2* is a known regulator of *MYC* transcription[49] and both genes are highly expressed in D458 (Fig. 5b). Hi-C data revealed several interactions of the *MYC* promoter with co-amplified regulatory elements of chromosome 8 (Fig. 5c) and chromosome 14 (Extended Data Fig. 8b). In summary, these results show that aberrant enhancer–promoter interactions resulting from structural rearrangements on ecDNA are common in medulloblastoma tumors.

## ecDNA-amplified enhancers modulate oncogene transcription

To test whether co-amplified enhancers on ecDNA have functional roles in tumor cell proliferation, we performed a pooled CRISPRi proliferation screen in the Group 3 medulloblastoma cell line D458, targeting all 645 accessible loci on the ecDNA using 32,530 small guide RNA sequences (sgRNAs). These loci included ten highly accessible regions from chromosome 14, each overlapping ENCODE candidate *cis*-regulatory elements[50]. Given that enhancer usage is highly conserved in Group 3 tumors[51], we performed the same screen in the Group 3 cell line D283, in which *MYC* (but not *OTX2*) is tandem amplified on a 55 Mbp homogeneously staining region of chromosome 8q (Fig. 5d). Although the *MYC* promoter was essential in both cell lines, our screen identified six functional elements that, upon CRISPRi inhibition, specifically reduced D458 proliferation compared to D283 after 21 days (MAGeCK MLE, *q* < 0.05; Fig. 5e)[52]. On chromosome 8, these loci included two accessible regions of a known *MYC* super-enhancer[51] and the *PVT1* promoter. In D458, much of the super-enhancer is duplicated internally on the ecDNA, and *PVT1* is amplified in D458 but not in D283. Conversely, we observed that other accessible regions of the same *MYC* super-enhancer were specifically essential for D283 but not for D458. The D458 interactome included interchromosomal interactions between *MYC* on chromosome 8 and regulatory elements of chromosome 14 co-amplified on the same ecDNA, two of which were essential for D458 proliferation (Extended Data Fig. 8b): a cluster of elements at the *OTX2* locus as well as a distal enhancer[53] located 80 kbp downstream of *OTX2* on the reference genome but inverted on the ecDNA. D283-specific elements on chromosome 14 included peaks at the amino-terminal exon of *OTX2* and another distal enhancer[53] 55 kbp from *OTX2* on the reference but also inverted on the ecDNA.

To further test the influence on transcription of regulatory regions essential in D458 but not in D283, we performed additional CRISPRi inhibition experiments targeting the *PVT1* promoter and an accessible region within the internal duplication of the *MYC* super-enhancer. Consistent with the result of the CRIPSRi proliferation screen, silencing of the *MYC* super-enhancer reduced *MYC* expression for two out of three sgRNAs in D458 but not in D283 (Extended Data Fig. 9a,b). No significant difference was observed in *OTX2* transcription in either cell line (Extended Data Fig. 9c,d). Silencing of the *PVT1* promoter abrogated *PVT1* transcription but not *MYC* or *OTX2*, in D458 but not in D283 (Supplementary Fig. 4). Thus, although proliferation in both Group 3 medulloblastoma cell lines is driven by *MYC* amplification, the relative importance of co-amplified genes and *cis*-regulatory elements is specific to the genomic architecture of the amplicon.

## Discussion

A long-standing problem in the clinical management of medulloblastoma tumors has been the paucity of effective targeted molecular treatments for the disease, especially in relapsed cases. For example, the *SMO* inhibitor vismodegib, one of few targeted drugs approved for SHH medulloblastoma, is ineffective against *TP53*-mutant, *MYCN*-amplified or *GLI2*-amplified tumors[54], each of which were recurrent features of ecDNA+ medulloblastoma in our patient cohort. By retrospective analysis of WGS and clinical outcome data from a large cohort of medulloblastomas, we demonstrate that ecDNA associates with poor outcome across the entire cohort and within individual disease subgroups. Survival analysis indicates that relative to patients with ecDNA−, patients with ecDNA+ medulloblastomas are more than twice as likely to relapse and three times as likely to die during the follow-up interval. Identification of ecDNA in medulloblastoma tumors is therefore crucial to pave the way for precision medicine approaches targeting ecDNA.

As in other cancers[2,3,55], ecDNA frequently amplifies known medulloblastoma oncogenes. ecDNA is a frequent feature of *MYC*-amplified Group 3 and *TP53*-mutant SHH tumors, which share exceptionally poor prognoses[16,17] but few other recurrent driver mutations. Recent longitudinal analysis of Barrett's esophagus suggests that *TP53* alteration is an early event in ecDNA-driven malignant transformation[55]. However, the absence of detectable ecDNA in *TP53*-mutant WNT subgroup tumors and the frequent occurrence of ecDNA in Group 3 tumors with wild-type

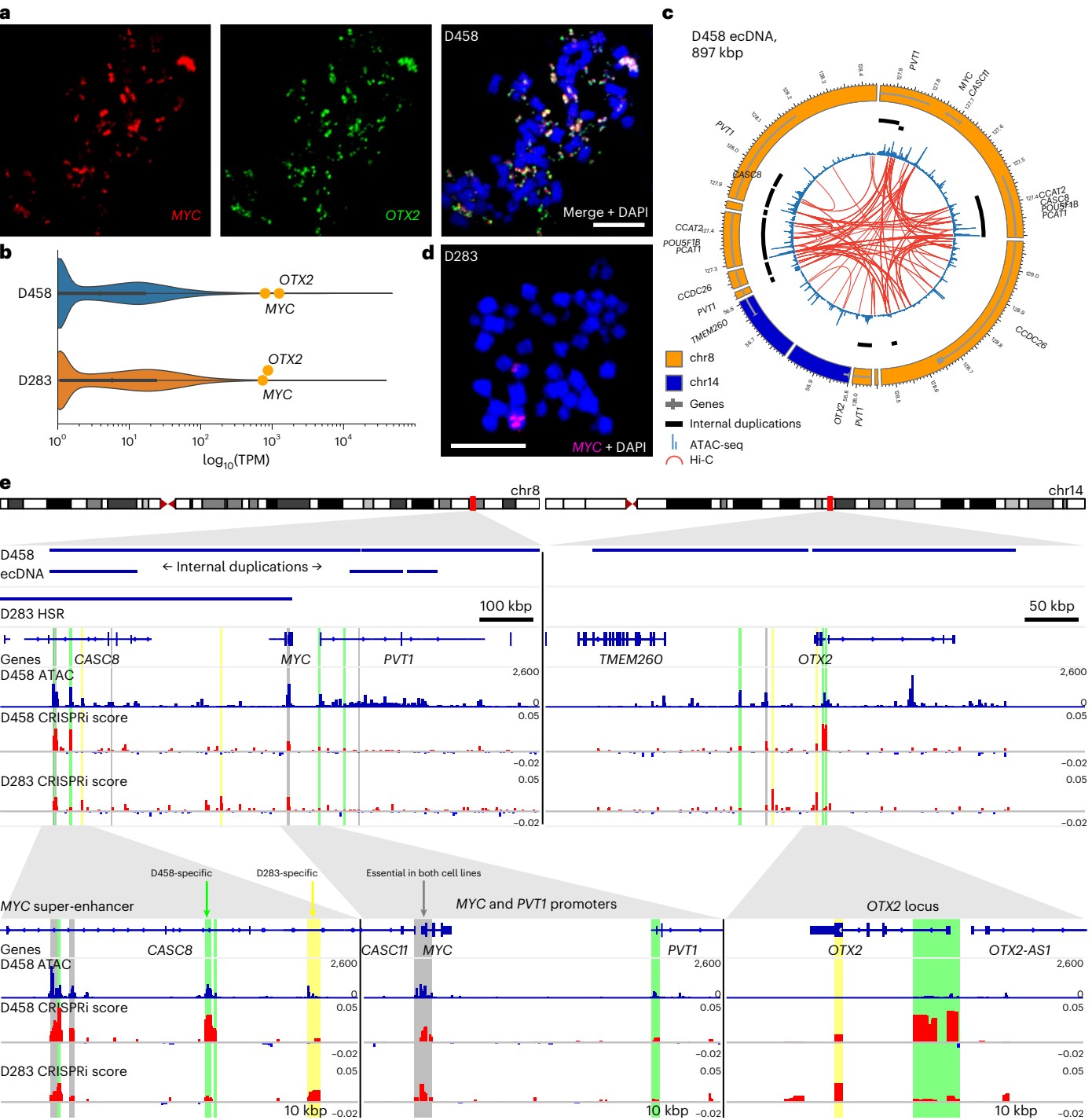

**Fig. 5 | Enhancer rewiring in medulloblastoma ecDNA affects cell proliferation. a**, Confocal FISH microscopy of *MYC* and *OTX2* on a D458 metaphase cell. Representative image of six metaphase cells. Scale bar, 10 μm. **b**, Gene transcription of all protein-coding genes in D458 and D283 from publicly available data in DepMap[62]. Medulloblastoma Group 3 oncogenes *MYC* and *OTX2* are highlighted. TPM, transcripts per million. **c**, Chromatin accessibility and interactions mapped onto the D458 amplicon. Tracks from outer to inner: genome sequence, internally duplicated sequences, chromatin accessibility, chromatin interactions. **d**, FISH in a metaphase spread of a D283 nucleus shows homogeneously staining region (HSR) chromosomal MYC amplification.

Representative image of 11 metaphase cells. Scale bar, 10 μm. **e**, Pooled CRISPRi screen in medulloblastoma cell lines D458 and D283 targeting all accessible loci on the D458 ecDNA. Tracks from top to bottom: D458 ecDNA-amplified loci; D283 HSR-amplified loci; genes; D458 chromatin accessibility; CRISPRi essentiality scores for D458 and D283 generated by CRISPR-SURF[63]. Vertical highlighted bars indicate accessible loci that are significantly depleted at T21 relative to T0 and are colored by cell line specificity. Gray: essential in D458 and D283 with no significant difference; green: essential in D458 relative to D283; yellow: essential in D283 relative to D458. Significance determined by MAGeCK MLE permutation test adjusted for false discovery rate[52] (*q* < 0.05).

*TP53* suggest that the mechanisms for the generation and selection of ecDNA may be modulated by subgroup-specific cellular contexts of medulloblastoma progenitor cells.

Close examinations of medulloblastoma tumors using FISH microscopy and single-cell sequencing reveal broad intratumoral distributions of ecDNA copy number per cell. In the illustrative example

of RCMB56, a pediatric SHH medulloblastoma tumor with somatic *TP53* mutation, we reconstructed two extrachromosomal amplifications and conclusively elucidated the circular structure of amp1. FISH and single-cell sequencing analyses concur that only a minority of RCMB56 primary tumor cells harbored high-copy amplification, and clustering on single-cell data suggests that these cells express a distinct transcriptional and epigenetic profile, including a canonical marker of SHH signaling. Based on these findings, it is imperative to investigate how the heterogeneous cell populations in medulloblastoma tumors respond to therapeutic pressure and contribute to treatment resistance and relapse.

By mapping accessible chromatin and chromosome conformation in medulloblastoma tumors and models, we find frequent gene regulatory rewiring as a consequence of ecDNA sequence rearrangement, suggesting that an altered gene regulatory landscape may contribute to transcriptional activation of ecDNA-amplified oncogenes. Consistent with previous findings in glioblastoma[13], a functional inhibition screen in two Group 3 *MYC*-amplified medulloblastoma cell lines shows that co-amplified enhancer function differs depending on the architecture of the amplification. However, the relative importance to oncogenic gene expression of native co-amplified enhancers versus aberrant regulatory rewiring on ecDNA remains an open question.

Recent studies have revealed intermolecular enhancer–promoter interactions between ecDNA molecules[40] or between the chromosomes and ecDNA[56]. To test for such intermolecular chromatin interactions in medulloblastoma, we computationally identified interchromosomal loops from Hi-C of the SHH medulloblastoma tumor RCMB56-pdx, in which one loop anchor mapped to the circular ecDNA amp1. This analysis revealed a nexus of interactions mapping from the *ARHGAP29* locus on amp1 to loci elsewhere in the genome with plausible tumorigenic roles, including *MECOM*, *RAD51AP2*, *POU4F1* and *IGF1R* (Extended Data Fig. 10). However, the functional significance of these intermolecular chromatin interactions in medulloblastoma remains untested.

ecDNA has been implicated in intratumoral heterogeneity[2,57,58], modulation of oncogene copy number in response to therapy[32,39,59] and evolution of targeted therapy resistance[6,7,58,60]. In this context, we have shown that ecDNA is a strong predictor for the outcome of patients with medulloblastoma tumors and is associated with other known molecular prognostic indicators, oncogene amplification, intratumoral copy number and transcriptional heterogeneity, and transcriptional regulatory rewiring. Further analysis of the mechanistic relationships between DNA repair pathway mutation, ecDNA formation and maintenance, and chemotherapy resistance may uncover new combinatorial therapies for patients with high-risk medulloblastoma who have exceptionally poor prognoses.

## Online content

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

[1]Bioinformatics and Systems Biology Graduate Program, University of California San Diego, San Diego, CA, USA. [2]Department of Medicine, University of California San Diego, San Diego, CA, USA. [3]Sanford Burnham Prebys Medical Discovery Institute, San Diego, CA, USA. [4]Department of Computer Science and Engineering, University of California San Diego, San Diego, CA, USA. [5]Department of Pediatrics, UC San Diego and Rady Children's Hospital, San Diego, CA, USA. [6]Department of Pathology, Stanford University School of Medicine, Stanford, CA, USA. [7]Sarafan ChEM-H, Stanford University, Stanford, CA, USA. [8]Department of Neurology and Herbert Irving Comprehensive Cancer Center, Columbia University, New York, NY, USA. [9]Medical Scientist Training Program, University of California San Diego, San Diego, CA, USA. [10]Biomedical Sciences Graduate Program, University of California San Diego, San Diego, CA, USA. [11]Center for Personal Dynamic Regulomes, Stanford University, Stanford, CA, USA. [12]Salk Institute for Biological Studies, La Jolla, CA, USA. [13]Department of Biological Engineering, Massachusetts Institute of Technology, Cambridge, MA, USA. [14]Rady Children's Institute for Genomic Medicine, Rady Children's Hospital and Healthcare Center, San Diego, CA, USA. [15]Clinical Cooperation Unit Neuropathology (B300), German Cancer Research Center (DKFZ), German Cancer Consortium (DKTK), and National Center for Tumor Diseases (NCT), Im Neuenheimer Feld 280, Heidelberg, Germany. [16]Papé Pediatric Research Institute, Department of Pediatrics and Knight Cancer Insitute, Oregon Health and Sciences University, Portland, OR, USA. [17]Division of Pathology, UC San Diego and Rady Children's Hospital, San Diego, CA, USA. [18]J. Craig Venter Institute, La Jolla, CA, USA. [19]Department of Pathology, University of California San Diego, San Diego, CA, USA. [20]Department of Pediatrics, University of California Irvine and Children's Hospital Orange County, Irvine, CA, USA. [21]Eli and Edythe Broad Institute of MIT and Harvard, Cambridge, MA, USA. [22]Department of Neurology, Boston Children's Hospital, Boston, MA, USA. [23]Harvard Medical School, Boston, MA, USA. [24]UPMC Hillman Cancer Center, Pittsburgh, PA, USA. [25]Department of Neurology, University of Pittsburgh, Pittsburgh, PA, USA. [26]Division of Biostatistics and Bioinformatics, Department of Family Medicine and Public Health, University of California San Diego, San Diego, CA, USA. [27]Department of Genetics, Stanford University, Stanford, CA, USA. [28]Howard Hughes Medical Institute, Stanford University School of Medicine, Stanford, CA, USA. [29]Moores Cancer Center, University of California San Diego, San Diego, CA, USA. [30]These authors jointly supervised this work: Jill P Mesirov, Lukas Chavez. ✉e-mail: lchavez@sbpdiscovery.org

## Methods

### Statistical methods

Statistical tests, test statistics and *P* values are indicated where appropriate in the main text. Categorical associations were established using the chi-squared test of independence if $n > 5$ for all categories and Fisher's exact test otherwise. For both tests, the Python package scipy.stats v1.5.3 implementation was used[64]. Multiple hypothesis corrections were performed using the Benjamini–Hochberg correction[65] implemented in statsmodels v0.12.0 (ref. 66). All statistical tests described herein were two-sided unless otherwise specified.

### Patient consent

Details on informed consent from patients for the collection of samples, and previously published data (Children's Brain Tumor Network (CBTN), St. Jude, International Cancer Genome Consortium (ICGC) and Archer datasets) are described in Supplementary Note 2. Patients that were diagnosed at Rady Children's Hospital–San Diego provided consent under the protocol *Molecular Tumor Profiling Platform for Oncology Patients* (IRB 190055), approved by the University of California San Diego (UCSD) Institutional Review Board (Supplementary Table 1). Patients were not compensated for their participation.

### Medulloblastoma WGS

Paired-end WGS data were acquired from different sources as described in Supplementary Note 2. In total, the WGS cohort comprised 468 patients (161 female, 277 male, 30 N/A; aged 0–36 years; see Supplementary Table 1). Unless otherwise specified, WGS was acquired for one tumor biosample per patient. Details on WGS data processing pipelines are described in Supplementary Note 2.

### ecDNA detection and classification from bulk WGS

To detect ecDNA, all samples in the WGS cohort were analyzed using AmpliconArchitect[24] v1.2 and AmpliconClassifier[3] v0.4.4. In brief, copy number segmentation and estimation were performed using CNVkit v0.9.6 (ref. 67). Segments with copy number ≥ 4 were extracted using AmpliconSuite-pipeline (April 2020 update) as 'seed' regions. For each seed, AmpliconArchitect searches the region and nearby loci for discordant read pairs indicative of genomic structural rearrangement. Genomic segments are defined based on boundaries formed by genomic breakpoint locations and by modulations in genomic copy number. A breakpoint graph of the amplicon region is constructed using the copy-number-aware segments and the genomic breakpoints, and cyclic paths are extracted from the graph. Amplicons are classified as ecDNA, breakage–fusion–bridge, complex, linear or no focal amplification by the heuristic-based companion script, AmpliconClassifier. Biosamples with one or more classifications of 'ecDNA' were considered potentially ecDNA+; all others were considered ecDNA− (Supplementary Table 3). We manually curated all potential ecDNA+ assembly graphs and reclassified those with inconclusive ecDNA status, which we defined as any of the following: low-copy amplification (<5) AND no copy number change at discordant read breakpoints; and/or cycles consisting of the repetitive region at chr5:820000 (GRCh37).

The ecDNA− status of the D283 cell line was not determined computationally by WGS, but by copy number analysis of DNA methylation, FISH (see Methods) and analysis of OGM data.

### Fingerprinting analysis

To uniquely identify WGS from each patient, we counted reference and alternate allele frequencies at 1,000 variable non-pathogenic single-nucleotide polymorphism locations in the human genome according to the 1000 Genomes project[68] and performed pairwise Pearson correlation between all WGS samples. Biospecimens originating from the same patient tumor (for example, primary–relapse or human tumor–PDX pairs) were distinguishable by high correlation across these sites ($r > 0.80$). We identified one case in which two tumor biosamples

had highly correlated fingerprints: MDT-AP-1217.bam and ICGC_MB127.bam. We arbitrarily removed ICGC_MB127 from the patient cohort.

### Patient metadata, survival and subgroup annotations

Where available, patient samples and models were assigned metadata annotations including age, sex, survival and medulloblastoma subgroup based on previously published annotations of the same tumor or model[18,23,31,37,69–71]. Sample metadata are also available in some cases from the respective cloud genomics data platforms: https://dcc.icgc.org (ICGC), https://pedcbioportal.kidsfirstdrc.org and https://portal.kidsfirstdrc.org (CBTN), and https://pecan.stjude.cloud (St. Jude). Patient tumors from CBTN were assigned molecular subgroups based on a consensus of two molecular classifiers, using RSEM-normalized FPKM data: MM2S (ref. 72) and the D3b medulloblastoma classifier at the Children's Hospital of Philadelphia (https://github.com/d3b-center/medullo-classifier-package). Where primary sources disagreed on a metadata value, that value was reassigned to N/A.

### *TP53* mutation annotations

**Somatic mutations.** Somatic *TP53* mutation information for the ICGC and CBTN cohorts was acquired from a previous publication[31] and from the ICGC and CBTN data portals. Somatic *TP53* mutation information for the St. Jude cohort was extracted from the standard internal St. Jude variant calling pipeline[20]. We only considered somatic mutations that were protein-coding and missense, nonsense, insertion or deletion, or that affected a splice site junction.

**Germline variants.** Germline variant GVCF files were downloaded from the ICGC, KidsFirst and St. Jude Pediatric Cancer Genome Project (PCGP) data portals. GVCF files were merged with GLnexus[73] and converted to PLINK format. PCGP genotypes were converted to hg19 coordinates using liftover. Variants from the *TP53* genomic locus (hg19:chromosome 17:7571739–759080) were extracted and annotated with REVEL (https://sites.google.com/site/revelgenomics)[74], CADD v1.6 (https://cadd.gs.washington.edu/info)[75], ClinVar (https://ftp.ncbi.nlm.nih.gov/pub/clinvar/vcf_GRCh37) and Variant Effect Predictor (VEP) r104 (http://grch37.ensembl.org/index.html)[76]. VEP variants that were considered pathogenic included 'frameshift' and 'splice' variants. ClinVar annotations that were considered pathogenic included 'frameshift', 'stop', 'splice' and 'deletion', and for which the clinical significance was 'pathogenic' or 'likely pathogenic'. CADD pathogenic variants had a CADD score of at least ten. REVEL pathogenic variants had a REVEL score of at least 0.5. Only variants with a minor allele frequency of less than 5% according to the gnomAD r2.1.1 database were analyzed[77].

### Survival analyses

Kaplan–Meier, Cox proportional hazards and AFT analyses were performed with Lifelines v0.26.5 (ref. 78). For all analyses, the sample set contained data from all patients annotated with the included covariates; no imputation was performed.

**Kaplan–Meier analysis.** For Kaplan–Meier analysis, the sample size was $n = 362$ (65 ecDNA+; 297 ecDNA−). Differential survival was determined by a log-rank test. For Kaplan–Meier analyses by class of structural variant, samples were assigned a label if at least one amplicon was classified by AmpliconClassifier with that label, in order of priority: ecDNA, breakage–fusion–bridge, complex non-cyclic, linear, no focal somatic copy number amplification[3]. Our sample of tumors with breakage–fusion–bridge amplification but no ecDNA was too small to test ($n = 2$).

**Cox proportional hazards on age, sex, molecular subgroup and ecDNA.** For the Cox proportional hazards analysis, the sample size was $n = 352$ observations. The model was fitted by maximum likelihood estimation.

**Cox proportional hazards on age, sex, molecular subgroup, p53 mutation and ecDNA.** For the proportional hazards analysis that included p53 mutation, the sample size was $n = 322$ observations. Collinearity, which is strong correlation between predictive variables in a regression model, can result in model instability and unreliable estimation of the collinear coefficients[79]. To address collinearity between ecDNA and p53 status in our model, we performed ridge estimation of model coefficients[80,81], determining the ridge penalty parameter $\lambda$ by grid search on fivefold cross-validation of model likelihood on the withheld set.

**AFT models and mediation analysis.** Mediation analysis was performed using the Baron–Kenny framework[35], following recent best practices[82]. Owing to the non-collapsibility of hazard ratios, the proportional hazards assumption and Cox proportional hazards model may not be suitable for mediation analysis in which we need to compare the coefficients with and without the mediator. Therefore, we fitted parametric log-normal AFT regression models as a reasonable alternative to Cox regression. Percentage change values were calculated as:

$$\text{Percentage change} = 100 \left[ e^{\hat{\beta}_k} - 1 \right]$$

where $\hat{\beta}_k$ is the maximum likelihood estimation regression coefficient for random variable $k$.

### OGM data collection and processing
Ultra-high molecular weight (UHMW) DNA was extracted from frozen cells preserved in dimethylsulfoxide (DMSO) following the manufacturer's protocols (Bionano Genomics). Cells were digested with Proteinase K and RNase A. DNA was precipitated with isopropanol and bound with nanobind magnetic disks. Bound UHMW DNA was resuspended in the elution buffer and quantified with Qubit dsDNA assay kits (ThermoFisher Scientific).

DNA labeling was performed following the manufacturer's protocols (Bionano Genomics). Standard Direct Labeling Enzyme 1 reactions were performed using 750 ng of purified UHMW DNA. Fluorescently labeled DNA molecules were imaged sequentially across nanochannels on a Saphyr instrument (Bionano). At least 400× genome coverage was achieved for all samples.

De novo assemblies of the samples were performed with Bionano's De Novo Assembly Pipeline (DNP) using standard haplotype-aware arguments (Bionano Solve v3.6). With the Overlap-Layout-Consensus paradigm, pairwise comparison of DNA molecules was used to create a layout overlap graph, which was then used to generate initial consensus genome maps. By realigning molecules to the genome maps ($P < 10^{-12}$) and using only the best-matched molecules, a refinement step was done to refine the label positions on the genome maps and to remove chimeric joins. Next, an extension step aligned molecules to genome maps ($P < 10^{-12}$) and extended the maps based on molecules aligning past the map ends. Overlapping genome maps were then merged ($P < 10^{-16}$). These extension and merge steps were repeated five times before a final refinement ($P < 10^{-12}$) was applied to 'finish' all genome maps.

### ecDNA reconstruction with OGM data
The ecDNA reconstruction strategy incorporated the copy-number-aware breakpoint graph generated by AmpliconArchitect[24] with OGM contigs generated by the Bionano DNP. For RCMB56 assemblies, we used contigs from the DNP as well as the Rare Variant Pipeline.

We used AmpliconReconstructor[83] v1.01 to scaffold individual breakpoint graph segments from OGM contigs, with the '–noConnect' flag set and otherwise default settings. A subset of informative contigs with alignments to multiple graph segments as well as a breakpoint junction were then selected for subsequent scaffolding, using the '–contig_subset' argument of AmpliconReconstructor's OMPathFinder.py script. For the exploration of unaligned regions of OGM contigs used

in the reconstructions, we used the OGM alignment tool FaNDOM[84] v0.2 (default settings). FaNDOM was used to identify the loose ends of the RCMB56 amp2.

RCMB56 amp1 and D458 were fully reconstructed as described above; however, RCMB56 amp2 required manual intervention. Owing to the fractured nature of the breakpoint graphs in RCMB56 amp2, we searched for copy-number-aware paths in the AmpliconArchitect breakpoint graph, using the plausible_paths.py script from the AmpliconSuite-pipeline, then converted these to in silico OGM sequences and aligned paths to OGM contigs directly using AmpliconReconstructor's SegAligner.

### Animals
NOD-SCID IL2Rγ null (NSG) mice (Jackson Laboratory, strain no. 005557) were housed in an aseptic barrier research animal facility at the Sanford Consortium for Regenerative Medicine, with a 12 h light–dark cycle, ambient temperature of 19–24 °C and 40–60% humidity. All experiments were performed in accordance with national guidelines and regulations, and according to protocols approved by the Animal Care and Use Committees at the Sanford Burnham Prebys Medical Discovery Institute and UCSD (San Diego, CA, USA) and the UCSD Institutional Review Board (Project no. 171361XF). In compliance with humane endpoint protocols, tumor-bearing mice displaying signs of moribundity (dysmorphic head, hunched posture, ataxia, excessive weight loss) were euthanized and processed without exceeding tumor burden limitations.

### Establishment and maintenance of PDX RCMB56
RCMB56-pdx was originally derived with consent from a *TP53*-mutant SHH subgroup medulloblastoma of an eight-year-old male patient who was diagnosed at Rady Children's Hospital–San Diego, under the protocol *Molecular Tumor Profiling Platform for Oncology Patients* (IRB 190055). Primary surgical tumor tissue was disassociated via Liberase (Sigma-Aldrich, 05401020001) and suspended in Neurocult media (Stem Cell Technologies, 05750). Cells ($0.5–1 \times 10^6$) were orthotopically implanted into NSG mouse cerebella for expansion. Initial xenograft tumor latency was six months post-implant, whereupon tumor tissue was dissected from moribund mice, dissociated and reimplanted into new recipient NSG mice or cryopreserved without in vitro passaging. Ex vivo experiments were performed with PDX RCMB56 cells from in vivo passage 1 (x1).

### Metaphase spreads
Cell lines were enriched for metaphases by the addition of KaryoMAX (Gibco) at 0.1 µg ml$^{-1}$ for 2 h to overnight (0.02 µg ml$^{-1}$ overnight for dissociated PDX cells). Single-cell suspensions were then incubated with 75 mM KCl for 8–15 min at 37 °C. Cells were washed in carnoy fixative (3:1 methanol:acetic acid) three times. Cells were then dropped onto humidified slides.

### FISH
Slides containing fixed cells were briefly equilibrated in 2× SSC buffer, followed by dehydration in 70%, 85% and 100% ethyl alcohol for 2 min each. FISH probes (Supplementary Table 20) diluted in hybridization buffer were applied to slides and covered with a coverslip. Slides were denatured at 72 °C for 1–2 min and hybridized overnight at 37 °C. The slide was then washed with 0.4× SSC, then 2× SSC-0.1% Tween 20. DAPI was added before washing again and mounting with Prolong Gold.

### Microscopy
Conventional fluorescence microscopy was performed using either the Olympus BX43 microscope equipped with a QiClick cooled camera, or the Leica DMi8 widefield fluorescence microscope followed by Thunder deconvolution using a ×63 oil objective. Confocal microscopy was performed using a Leica SP8 microscope with lightning deconvolution

and white light laser (UCSD School of Medicine Microscopy Core). Excitation wavelengths for multiple color FISH images were set manually based on the optimal wavelength for the individual probes, with care taken to minimize crosstalk between channels. ImageJ 1.53 was used to uniformly edit and crop images.

## Automated FISH analysis

**Cell segmentation.** We applied NuSeT[85] to perform cell segmentation. The parameters were min_score 0.95, nms threshold of 0.01, a nuclei size threshold of 500 and a scale ratio of 0.3.

**Number of FISH blobs.** To annotate pixels with high local intensity, we convolved the original image with a sampled Gaussian kernel, with a standard deviation of three pixels and a size of seven by seven pixels. After convolving, we applied a threshold of 15 / 255 pixel brightness. Then, to filter out low brightness noise, we set a binary threshold that the brightness of these peaks must exceed one standard deviation above the average FISH brightness and added an additional minimum area requirement.

**Amplification mechanism.** We ran ecSeg-i[86] on each segmented cell to determine the amplification mechanism. ecSeg-i produces three probability scores representing the likelihood of the cell having no amplification, ecDNA amplification or homogeneously staining region amplification. We assigned the amplification mechanism with the highest likelihood.

## Single Cell Multiome ATAC + Gene Expression sequencing

From the RCMB56 primary patient tumor (RCMB56-ht), disassociated cryopreserved cells stored in 10% DMSO/FBS were used. At least 50 mg of tissue (1 M cells) was used for both samples. Disassociated cells were prepared for Single Cell Multiome ATAC + Gene Expression sequencing (10× Genomics) according to the manufacturer's instructions[87]. Sequencing was performed on an Illumina NovaSeq S4 200 to a depth of at least 250 M reads for snATAC-seq and 200 M reads for snRNA-seq.

## Single-cell data processing and clustering

Sequencing data were uniformly processed using CellRanger ARC v2.0.0 with default parameters, followed by Seurat v4.0.4 (ref. 41). Cell barcodes that passed the following quality thresholds were retained: ATAC mitochondrial fraction less than 0.1; ATAC read count between 1,000 and 70,000; and RNA read count between 500 and 25,000. Doublets were identified and removed using DoubletFinder v2.0 (ref. 88) using default parameters. Single-cell transcription data were normalized using regularized negative binomial regression, implemented in the sctransform package[89] (SCT) included with Seurat.

Clustering was performed using the weighted nearest neighbors algorithm[41] with a resolution of 0.1 and the other parameters set at default. To label cell clusters with cell type identities, differentially expressed genes were found for each cluster using Seurat's FindAllMarkers function with default parameters (Supplementary Table 19) and cross-referenced against known cell type marker genes[90].

Copy number estimation from scRNAseq was performed using InferCNV v1.3.3 (ref. 91). Normal reference cells were defined as ecDNA− cells belonging to cell clusters labeled as normal cell types. All parameters were set at default.

Sequencing coverage of single cells (Fig. 4c) were visualized in IGV desktop v2.9.2 (ref. 92). Bulk WGS coverage (bigwig format) was generated from deduplicated sequencing reads using deeptools v3.5.1 (ref. 93) bamCoverage was at 50 bp resolution using default parameters. Single-cell coverage tracks were parsed from CellRanger ARC atac_fragments.txt.gz output format to .bed format using a custom script, then converted to bigwig format using bedtools v2.27.1 (ref. 94) genomecov and UCSC browser tools[95] bedGraphToBigWig v4.

## Identification of ecDNA-containing cells

ecDNA-containing cells were identified by permutation tests comparing snATAC-seq read coverage at the ecDNA regions to read coverage of random regions elsewhere in the genome. In brief, deduplicated snATAC-seq reads from the fragments.tsv output of CellRanger ARC were sorted by barcode. For Monte Carlo permutation testing, 1,000 random contiguous regions of the genome, excluding centromeres, telomeres, known ecDNA and low-mappability regions, were generated using bedtools v2.27.1 (ref. 94). Read coverage was counted using PyRanges v0.0.112 (ref. 96) and scaled to region length. For each cell, empirical $P$ values were estimated as $\hat{p} = (r + 1)/(n + 1)$, where $r$ is the rank of the test value out of $n$ permutations[97]. Multiple hypothesis correction was performed using a Benjamini–Hochberg-corrected $P$ value ($P < 0.10$). $Z$-scores were calculated using the standard formula, comparing the average read coverage at the ecDNA-amplified region to the mean and variance of the Monte Carlo permutations.

## Single-sample gene set enrichment analysis

Single-sample gene set enrichment analysis (ssGSEA) is a variation of gene set enrichment analysis for quantifying the aggregate expression of a gene set across the transcriptome of one sample[42]. To quantify the transcriptional activity of ecDNA in single cells, we performed ssGSEA of two gene sets comprising every gene amplified on RCMB56 amp1 or amp2, treating each cell as a single sample. The population sample consisted of $n = 247$ ecDNA+ cells from the RCMB56-ht sample. Gene expression values were the SCT-normalized transcription matrix, generated as described above using Seurat v4.0.4. ssGSEA was run using ssGSEA v10.0.11 implemented at https://cloud.genepattern.org (ref. 98). Association with z-score ecDNA copy number estimates was performed using Pearson's $R$, implemented in scipy.stats v1.7.3 and visualized using Seaborn v0.9.0 (ref. 99) *histplot*.

## ATAC-seq

ATAC-seq was performed at the Massachusetts Institute of Technology (Cambridge, MA) or ActiveMotif (San Diego, CA). Center-specific detail is included in Supplementary Note 3. Reads were aligned to the hg38 reference, deduplicated and preprocessed according to ENCODE best practices. Accessible chromatin regions were identified using MACS2 v2.1.2 (ref. 100) using a Benjamini–Hochberg-corrected $P$ value threshold ($P < 0.05$).

## Chromosome conformation capture (Hi-C)

Hi-C was performed at the Salk Institute (La Jolla, CA) or Arima Genomics (San Diego, CA). Center-specific details are included in Supplementary Note 4.

## Hi-C data processing

Hi-C reads were trimmed using Trimmomatic 0.39 (ref. 101) and aligned to the hg38 human genome reference using HiC-Pro v2.11.3-beta and bowtie 2.3.5 (ref. 102) with default parameters[103]. Visualization and contact normalization was performed with JuiceBox v1.11.08 (ref. 104) and the Knight–Ruiz algorithm[105]. Intrachromosomal chromatin interactions were called using Juicer Tools GPU HiCCUPS v1.22.01 (ref. 106) using a false discovery rate threshold of 0.2 and default recommended parameters[45]. Visual inspection indicated that HiCCUPS correctly annotated interactions mapping to ecDNA, except for locus pairs mapping within ~50 kb of a structural rearrangement. Owing to these technical challenges, chromatin interactions described herein were manually curated based on HiCCUPS interaction calls. Ectopic chromatin interactions spanning breakpoints on the D458, MB268 and RCMB56 ecDNA, including interchromosomal interactions, could not be accurately called by any software tools known to us because of technical limitations in this emerging field. These interactions were manually annotated from the interaction matrices shown in Extended Data Figs. 6–8.

## Identification of intermolecular chromatin interactions

To screen for putative intermolecular chromatin interactions originating from possible mobile enhancers[56] on ecDNA, we performed loop detection on Hi-C data of RCMB56-pdx using FitHiC v2.0.8 (ref. 107) interchromosomal mode, at a resolution of 50 kbp and setting no bias upper bound, as recommended by the tool's authors for this task. Interactions with corrected $q$-values less than 0.05 were selected and then further filtered for loops with one anchor mapping to RCMB56 amp1. To reduce false-positive loop calls originating from copy number variation, loops mapping to amp2 or to within 100 kbp of a breakpoint on amp1 were also removed. After filtering, 46 high-confidence loops remained that mapped from amp1 to elsewhere in the reference genome. Genes were associated with a loop if the gene locus overlapped the 50 kbp loop anchor. Panel S11a was generated using circos v0.69-8 (ref. 108).

## Pooled CRISPRi proliferation screen

The pooled CRISPRi proliferation screen was designed after a similar screen in glioblastoma cell lines[13]. In brief, this screen targeted all 645 accessible regions of the D458 ecDNA with 32,530 sgRNAs. Cultures of D458 (ecDNA+) and D283 (ecDNA−) cells were grown for 21 days and then sequenced to determine overrepresented and underrepresented sgRNAs. Further details are provided in Supplementary Note 5.

## Targeted CRISPRi experiments

For CRISPRi experiments, D283 and D458 cells were lentivirally transduced with dCas9-KRAB-mCherry plasmid[109] (Addgene, 60954) to express dCas9. Cells stably expressing dCas9 were FACS-sorted based on mCherry expression and transduced with sgRNA vectors. sgRNAs were cloned into the lentiGuide-puro plasmid (Addgene, 52963) (ref. 110). sgRNAs are listed in Supplementary Table 21. All plasmids were verified by Sanger sequencing. HEK293T cells (ATCC, CRL-3216) were used to generate lentiviral particles by cotransfecting the packaging vectors psPAX2 and pMD2.G using LipoD293 transfection reagent (SignaGen, SL100668).

## Quantitative RT–PCR

Five days after sgRNA transduction, total cellular RNA was isolated from cell pellets using a Qiagen RNeasy Kit. iScript cDNA Synthesis Kit (Bio-Rad, 1708890) was used for reverse transcription into cDNA. Quantitative RT–PCR was performed in technical triplicate for two bioreplicates of each experimental condition on a Bio-Rad CFX384 Real-Time System using SYBR Green PCR Master Mix (Bio-Rad, 1725270). qPCR primers are listed in Supplementary Table 22.

Gene transcription was estimated using the delta delta Ct method (Exp, $2^{-\Delta\Delta Ct}$) relative to actin. Testing for change in gene expression was performed using one-sided nested ANOVA with Dunnett's multiple comparisons test, implemented in GraphPad Prism v9.5.2.

## Biological material availability

PDX and cell line materials used in this study are available upon request. Patient tumor material used in this study are depleted and therefore not available.

## Reporting summary

Further information on research design is available in the Nature Portfolio Reporting Summary linked to this article.

## Data availability

WGS data from the ICGC, CBTN and St. Jude datasets are under controlled access as implemented by the respective organizations, but are available from the following sources upon reasonable request. ICGC and Archer patient cohorts: International Cancer Genome Consortium (https://dcc.icgc.org). Inclusion criteria were all medulloblastomas from datasets PEME-CA and PBCA-DE. CBTN patient cohort: Kids First Data Resource Center (https://kidsfirstdrc.org). Inclusion criteria were all medulloblastomas from dataset PBTA-CBTN as of March 2020. St. Jude patient cohort: St. Jude Cloud (https://www.stjude.cloud). Inclusion criteria were all medulloblastomas from the Pediatric Cancer Genome Project (PCGP, SJC-DS-1001) and Real-Time Clinical Genomics (RTCG, SJC-DS-1007) datasets as of March 2020. Rady Children's Hospital patient cohort, medulloblastoma cell line and PDX models: SRA PRJNA1011359. OGM contigs: SRA PRJNA1011359. Other datasets referenced in this work: 1000 Genomes Common SNPs (that is, dbSNP b141; https://ftp.ncbi.nih.gov/snp); DepMap 21Q2 (https://depmap.org/portal/download/all); ENCODE Registry of cCREs v3 (https://screen.encodeproject.org). ATAC-seq, Hi-C, single-cell sequencing and pooled CRISPRi screen data are available at the NCBI Gene Expression Omnibus (GEO) under accession GSE240985. FISH images are available at https://doi.org/10.6084/m9.figshare.c.6759093. Source data are provided with this paper.

## Code availability

Code for the AmpliconArchitect family of software tools is available from the following repositories: PrepareAA (https://github.com/jluebck/PrepareAA); AmpliconArchitect (https://github.com/jluebck/AmpliconArchitect); AmpliconClassifier (https://github.com/jluebck/AmpliconClassifier). Code for the analysis and generation of the figures is available from the following repositories: analyses on clinical and bulk sequencing data (https://github.com/auberginekenobi/medullo-ecdna); and detection and quantification of ecDNA in single-cell ATAC-seq data (https://github.com/auberginekenobi/ecdna-quant). Other single-cell analyses: https://github.com/auberginekenobi/rcmb56-single-cell.

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

## Acknowledgements

This work was delivered as part of the eDyNAmiC team supported by the Cancer Grand Challenges partnership funded by Cancer Research UK (CRUK) (P.S.M. and H.Y.C., CGCATF-2021/100012; V.B. and J.L., CGCATF-2021/100025) and the National Cancer Institute (P.S.M. and H.Y.C., OT2CA278688; V.B. and J.L., OT2CA278635). This work is supported by a generous endowment by the Clayes Foundation to the Research Center for Neuro-Oncology and Genomics within the Rady Children's Institute for Genomic Medicine, a Hannah's Heroes St. Baldrick's Scholar Award (L.C.), a grant from The National Brain Tumor Society (P.S.M.), funding from the National Institutes of Health (NIH) National Institute of Neurological Disorders and Stroke Institute R35 NS122339 (R.J.W.R.), R01 NS132780 (L.C.), R21 NS130137 (L.C.), R21 NS116455 (L.C.) and R21 NS120075 (L.C.), the NIH National Cancer Institute R01 CA159859 (R.J.W.R.), R01 CA238249 (P.S.M.), R01-CA238379 (P.S.M.), U24 CA258406 (J.P.M. and J.T.R.), U24 CA210004 (J.P.M. and J.T.R.), U24 CA220341 (J.P.M.), U24 CA264379 (V.B., P.S.M. and J.P.M.), U01 CA184898 (J.P.M., E.F., S.L.P.), U01 CA253547 (J.P.M. and E.F.), F99 CA274692 (K.L.H.) and F31 CA271777 (O.S.C.), the NIH National Institute of General Medical Sciences R01 GM074024 (J.P.M.) and R01 GM114362 (V.B.), the NIH National Library of Medicine T15 LM011271 (O.S.C.), a Moores Cancer Center Pilot Grant (L.C., V.B., J.P.M. and P.S.M.), Hyundai Hope on Wheels (J.E.A.M., Y.J.C.) and a Stanford Graduate Fellowship (K.L.H.). Microscopy work was supported by funding from the NIH National Institute of Neurological Disorders and Stroke P30 NS047101 (University of California San Diego Microscopy Core). This work used the Extreme Science and Engineering Discovery Environment (XSEDE), which is supported by National Science Foundation grant number ACI-1548562. DNA methylation array analysis was conducted at the IGM Genomics Center, University of California, San Diego, La Jolla, CA (P30 CA023100). This research was conducted using data made available by The Children's Brain Tumor Network (formerly the Children's Brain Tumor Tissue Consortium). H.Y.C. is an Investigator of the Howard Hughes Medical Institute. In addition, we thank I. A. Reyes, J. H. Zhang, C. McLeod and A. Resnick for facilitating data access; M. Reich and

M. Tatineni for computational support; J.Olson, J. Weissman, R. Vibhakar and X.-N. Li for biosamples and materials; A. Pang (Bionano Genomics) for assistance running the Bionano Assembly pipeline; C.-C. Yang, C.-T. Huang, R. Murad, A. Morton and P. Scacheri for CRISPRi services and guidance; M. Kazachkova for exploratory analyses on the data; and A. Wenzel and M. Chapman for helpful scientific discussions.

## Author contributions

O.S.C., J.L., J.P.M. and L.C. prepared the manuscript and figures. S. Wani, A.T., S.C., M.A., L.M., D.D., C.H., J.R.C., N.G.C., M.L.L., D.M.M., A.K., S.L.P., J.R.D., A.B., A.J.D., R.H.S. and E.F. performed sample preparation and experimental analysis. J.D.L., Y.Y.L. and R.J.W.R. contributed PDX mouse models. J.T.L., I.T.L.W. and P.S.M. performed all microscopy. K.L.H., B.J.H. and H.Y.C. performed CRISPR-CATCH. S. Wani., A.T., D.D., L.M., J.N.R., J.E.A.M., Y.J.C, A.B. and A.J.D. performed CRISPRi sample preparation, experiments and analysis. O.S.C., J.L., M.S.P., S. Wang, Y.L., A.D., C.G., S.S., J.T.R., S.R.D., G.P., U.R., E.J., X.Z., H.C. and V.B. contributed computational data analyses. O.S.C., J.P.M. and L.C. designed the study and J.P.M. and L.C. co-supervised the project.

## Competing interests

The authors declare the following competing interests: H.Y.C. is a co-founder of Accent Therapeutics, Boundless Bio, Cartography Biosciences, Orbital Therapeutics and is an advisor of 10x Genomics, Arsenal Biosciences, Chroma Medicine and Spring Discovery. P.S.M. is a co-founder, chairs the scientific advisory board (SAB) and has equity interest in Boundless Bio. P.S.M. is also an advisor with equity for Asteroid Therapeutics and is an advisor to Sage Therapeutics. V.B. is a co-founder, consultant, SAB member and has equity interest in Boundless Bio and Abterra. The terms of this arrangement have been reviewed and approved by the University of California, San Diego in accordance with its conflict of interest policies. J.L. is a part-time consultant for Boundless Bio. The terms of this arrangement have been reviewed and approved by the University of California, San Diego in accordance with its conflict of interest policies. J.T.L. is an employee of Boundless Bio. His employment began after his contributions to the manuscript. The remaining authors declare no competing interests.

## Additional information

**Extended data** is available for this paper at https://doi.org/10.1038/s41588-023-01551-3.

**Correspondence and requests for materials** should be addressed to Lukas Chavez.

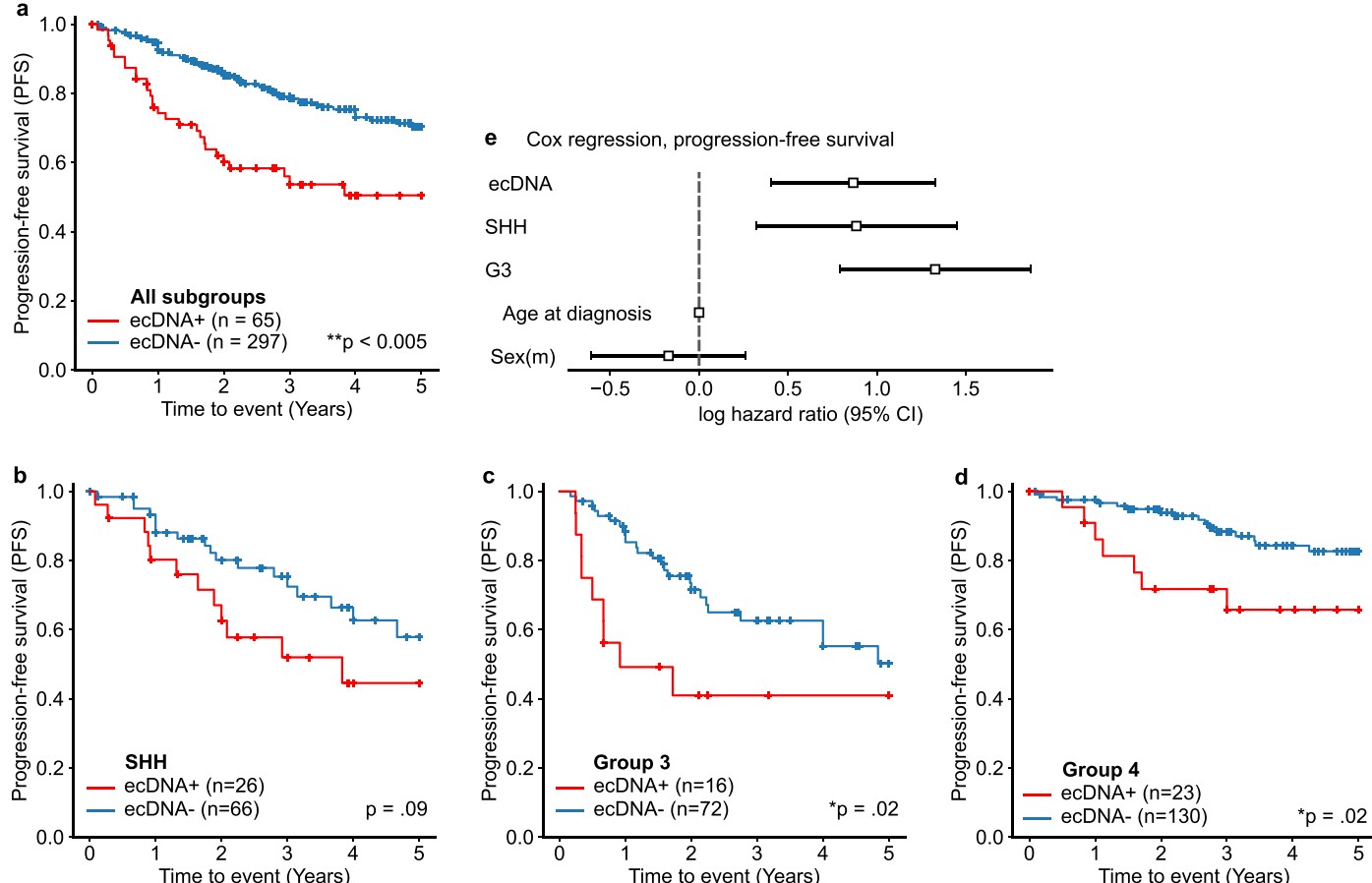

**Extended Data Fig. 1 | Progression-free survival of the medulloblastoma patient cohort.** (**a**) Kaplan-Meier curve depicting 5-year progression-free survival (PFS) in the patient cohort stratified by presence of extrachromosomal DNA (ecDNA) in patient tumors. *p* = 1.2e-4. (**b-d**) Kaplan-Meier curves indicating PFS for SHH (b, *p* = 0.09), Group 3 (c, *p* = 0.02), and Group 4 (d, *p* = 0.02) subgroups, stratified by ecDNA presence. All *p*-values derived from two-sided log-rank test; no adjustment was performed for multiple hypotheses. (**e**) Log hazard ratios for ecDNA status, medulloblatoma subgroup, age and sex estimated by Cox proportional hazards regression on PFS. Sample was *n* = 322 observations. Bars indicate 95% confidence intervals.

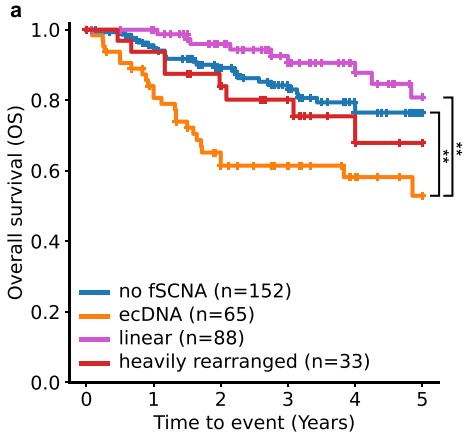
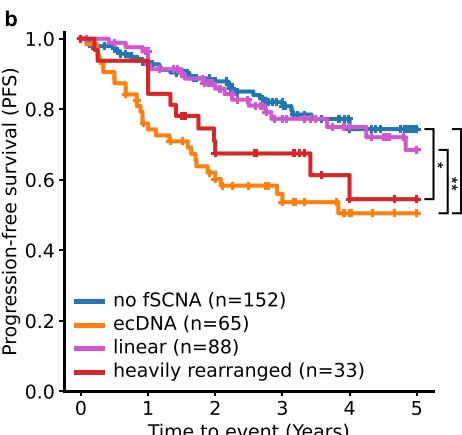

**Extended Data Fig. 2 | Patient cohort survival by genomic amplification class.** Kaplan-Meier curves indicating (**a**) overall survival and (**b**) progression-free survival for n = 338 medulloblastoma patients, stratified by the amplifications present in the patient tumors. Patients with ecDNA amplification had significantly worse overall and progression-free survival compared to those with linear amplifications and compared to those without focal somatic copy number amplification (fSCNA). * *p* < 0.05; ** *p* < 0.005.

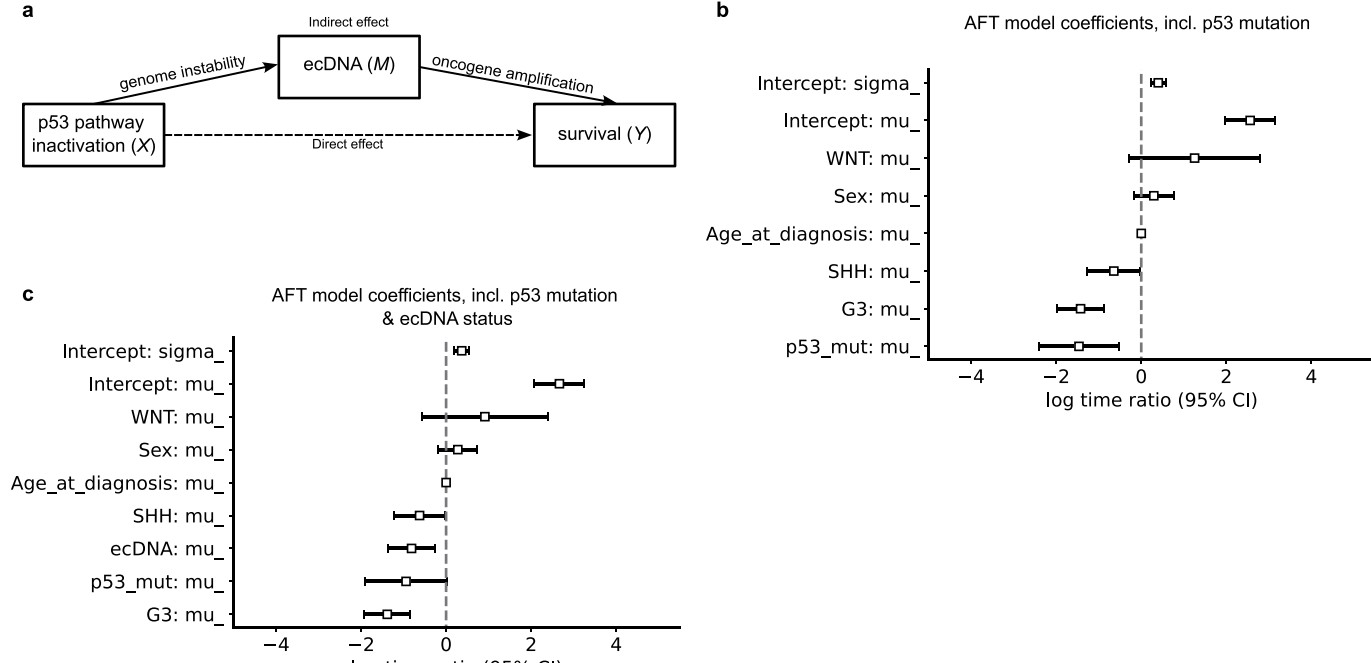

**Extended Data Fig. 3 | Mediation links ecDNA to known prognostic markers.** Log-normal accelerated failure time (AFT) regression models estimating relative time to progression or death of n = 340 patients. (**a**) Model diagram of proposed mediation by ecDNA of the effect of *TP53* mutation on survival. According to this model, *TP53* inactivation generates genome instability, facilitating the formation of ecDNA, which then affects survival. (**b**) Forest plot of $\mu$ coefficients (log time ratios) of AFT model including age, sex, subgroup and *TP53* mutation status as covariates. The estimate $\mu_{p53\_mut} = -1.5$ indicates $1 - exp(\mu_{p53\_mut}) = 77\%$ reduction in

expected survival time for medulloblastoma patients with *TP53*-mutant tumors. (**c**) $\mu$ coefficients of AFT model including ecDNA as an additional covariate estimates 56% reduction in survival time for patients with ecDNA-positive (ecDNA+) tumors and an insignificant and reduced coefficient *TP53* for mutation, indicating partial mediation by ecDNA of the effect of *TP53* mutation on survival. Data in (b) and (c) are presented as maximum likelihood estimate (MLE) +/− 95% confidence intervals.

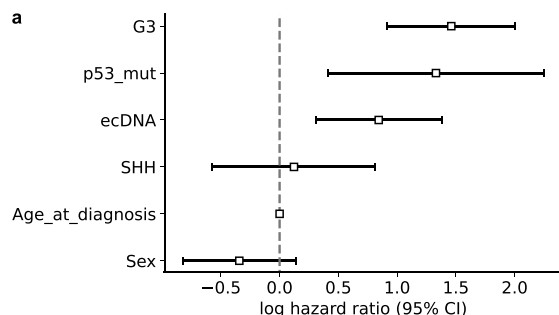
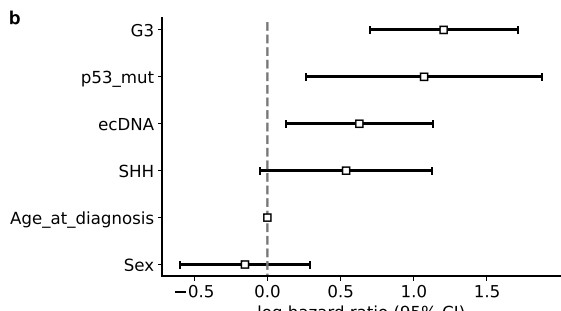

**Extended Data Fig. 4 | Estimated hazards of clinical and molecular features on medulloblastoma patient survival.** Forest plots of $\beta$ coefficients (log hazard ratios) of Cox Proportional Hazards models fitted on n = 322 patients using L2 ridge regression to control instability due to collinearity. WNT subgroup patients were excluded due to perfect separation. Log hazard estimates are relative to Group 4 and female patients. Log hazard ratios for (**a**) OS and (**b**) PFS of Cox models including age, sex, subgroup, ecDNA and *TP53* mutation as covariates. Data are presented as maximum *a posteriori* (MAP) estimate with a Gaussian prior (L2 regularization) +/− 95% confidence intervals.

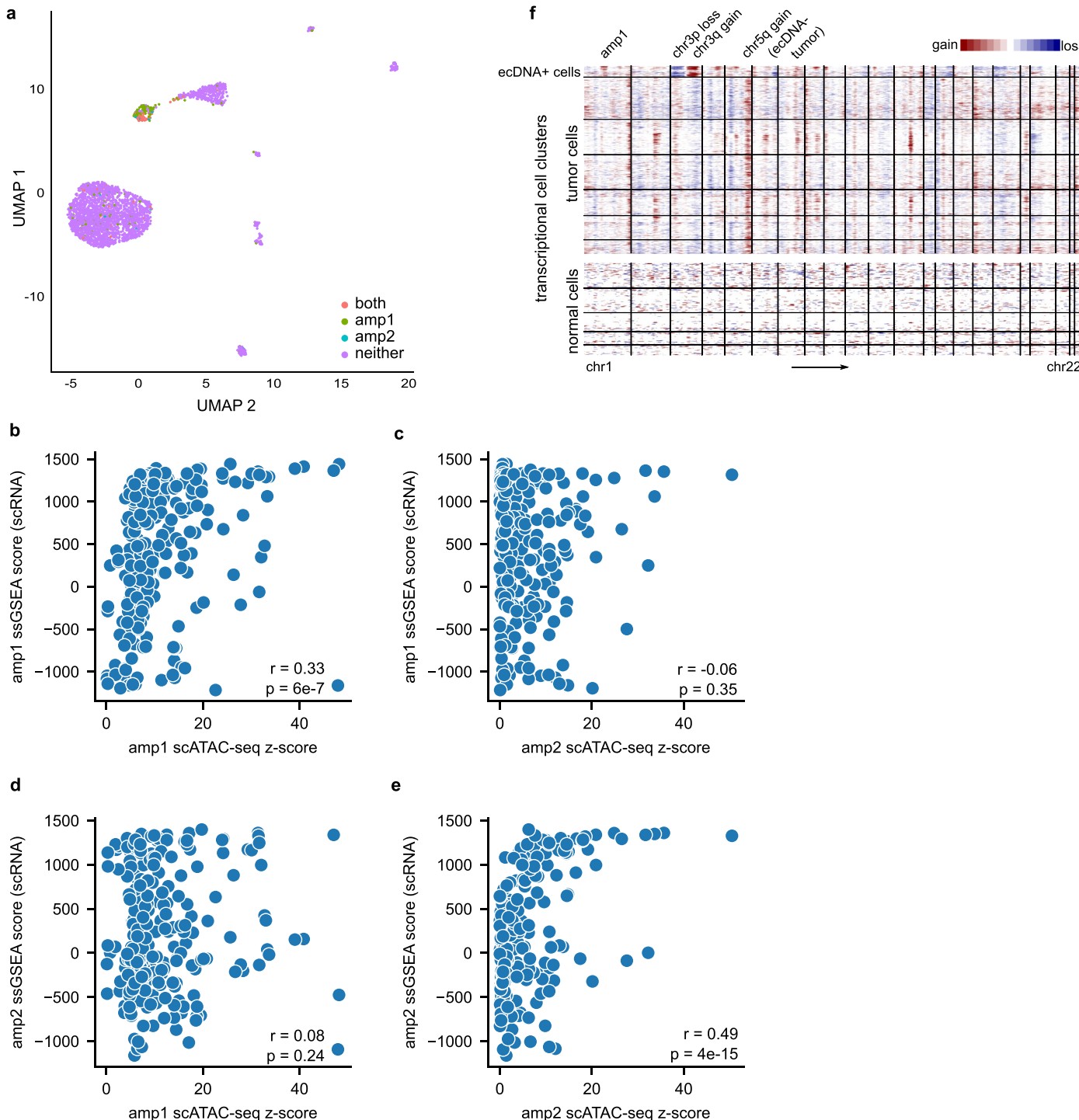

**Extended Data Fig. 5 | Transcriptional and accessible chromatin features of ecDNA-containing cells from the RCMB56 primary tumor.** (**a**) UMAP projection of RCMB56-p0 cells by transcriptional and accessible chromatin similarity. Cells are colored to indicate whether high-copy amplification was detected in snATAC-seq data at the amp1 or amp2 loci. Cells carrying one or both amplifications are enriched in a small transcriptionally distinct cluster of cells. (**b-e**) Correlations between copy number at the amplified locus (z-scores) and transcriptional activity of amplified genes (ssGSEA scores). Copy number of amp1 is associated with transcription of amp1-amplified genes, but not with transcription for amp2-amplified genes. Conversely, copy number of amp2 is associated with amp2-amplified, but not amp1-amplified, gene expression. p-values are derived from two-sided Student's *t*-test; no adjustment was performed for multiple hypotheses. (**f**) Genome-wide copy number estimation of normal and tumor single cells in RCMB56-ht. The ecDNA+ tumor cell cluster is distinguished by gain of chr1 at the amp1 locus, gain of chr3q, loss of chr3p, and no copy number change to chr5q. Amp2 (chr7 and chr17) is not readily visible at the resolution afforded by CNV estimation from single-cell transcription at this sequencing depth.

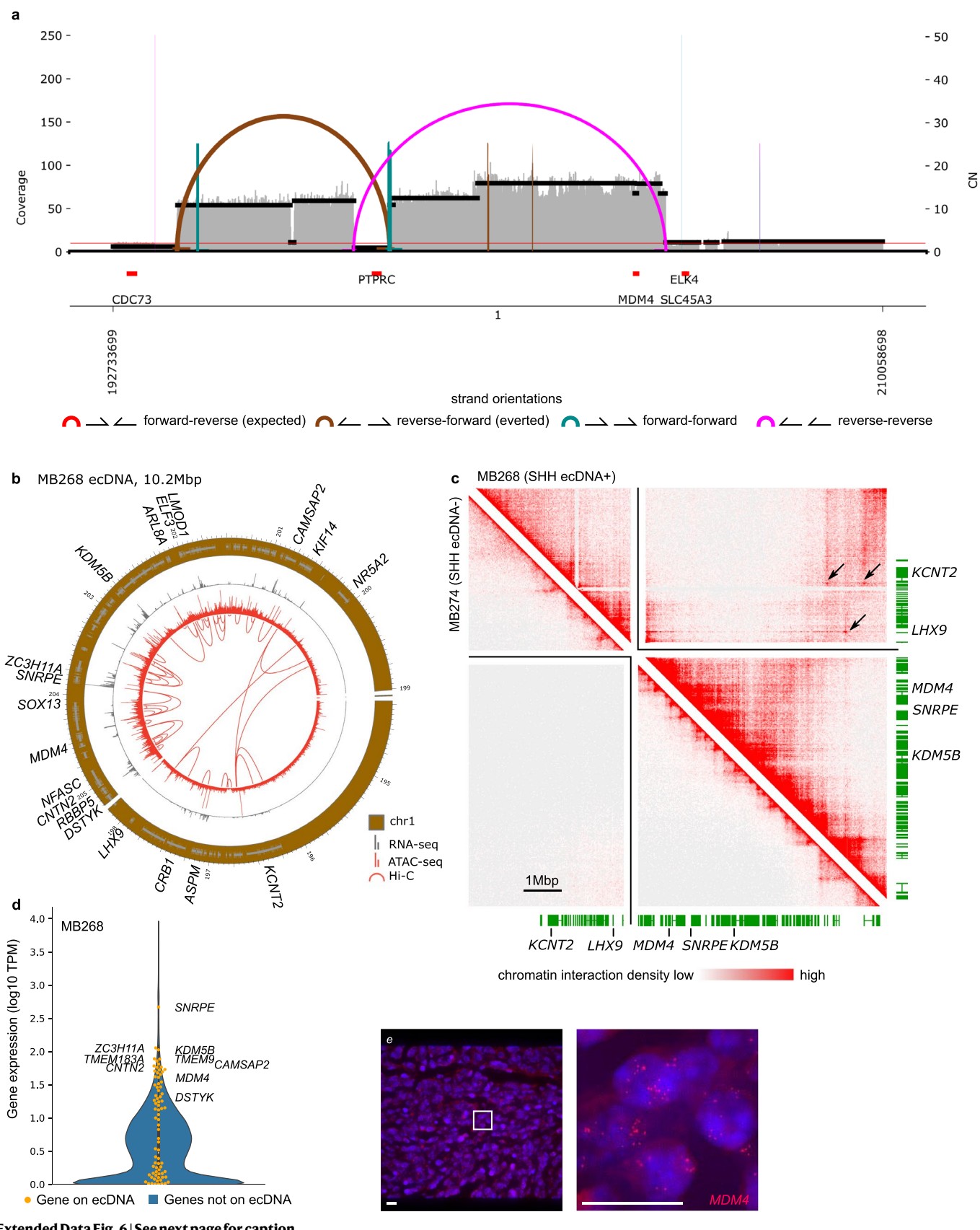

Extended Data Fig. 6 | See next page for caption.

**Extended Data Fig. 6 | Sequence and conformation of the MB268 ecDNA.**
(**a**) AmpliconArchitect resolves a circular structure composed of 3 segments of chr1 from short paired-end reads. (**b**) RNA-seq, ATAC-seq and Hi-C interactions mapped onto the ecDNA sequence. Amplified oncogenes include *MDM4*, a *TP53* pathway inhibitor frequently amplified on ecDNA of cancers of various types. Chromatin interactions spanning breakpoints target accessible regions at the *LHX9* and *KCNT2* loci, but neither gene is expressed. (**c**) Hi-C interaction density mapped onto the ecDNA sequence. Long-range chromatin interactions spanning breakpoint junctions are indicated by arrows. (**d**) Gene expression in the MB268 primary tumor. All ecDNA-amplified genes are indicated by the orange swarmplot; highly expressed genes are labelled. The violin plot indicates a kernel density estimate of the distribution of expression of all genes in MB268. (**e**) FISH of *MDM4* in the MB268 primary tumor confirms extrachromosomal amplification of *MDM4*. Representative image of 18 regions of 1 FFPE tissue slide. Scale bar is 10 μm.

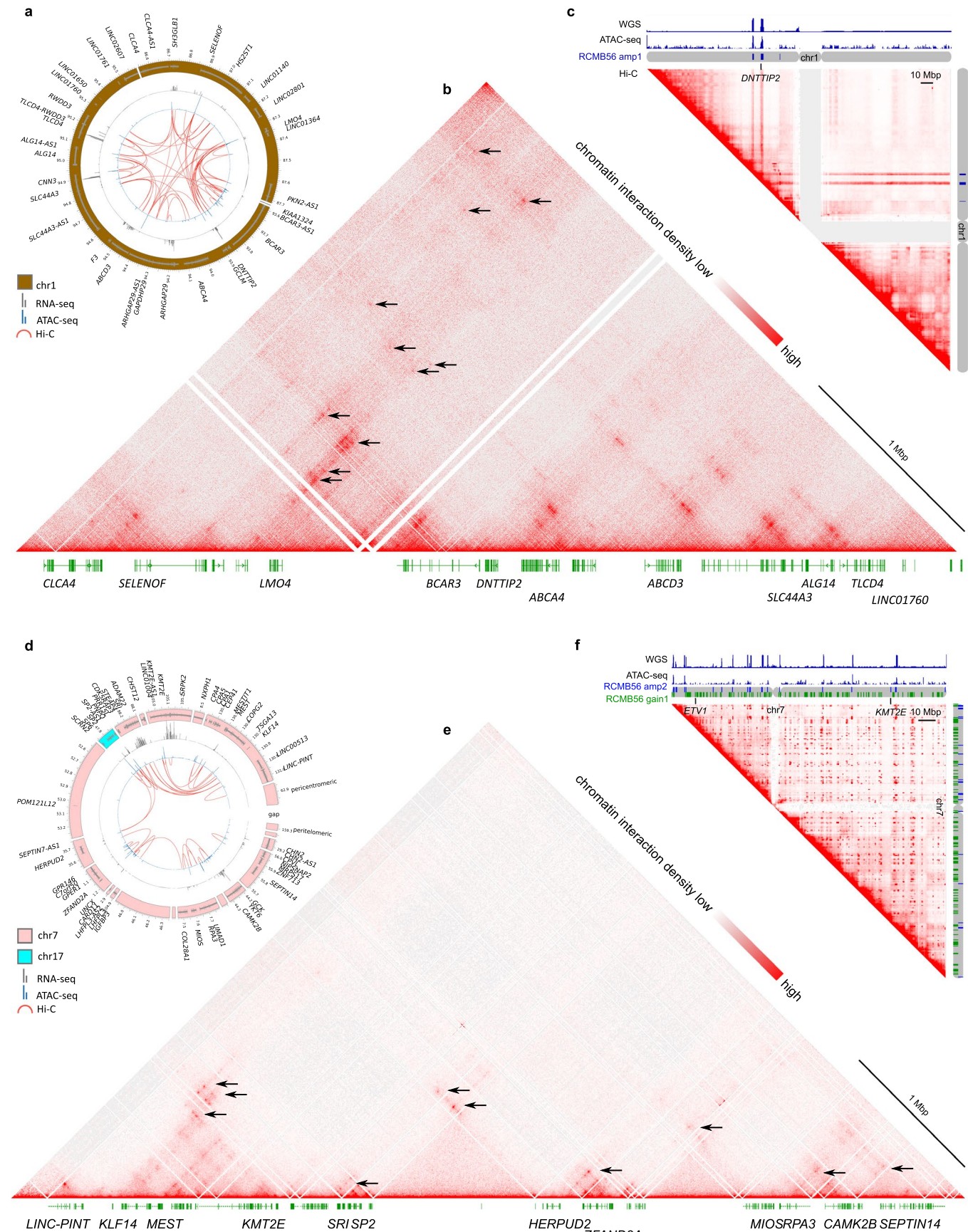

**Extended Data Fig. 7 | See next page for caption.**

**Extended Data Fig. 7 | Sequence and conformation of the amp1 and amp2 high-copy amplifications in RCMB56-pdx cells.** (**a**) Transcription (RNA-seq, grey), accessible chromatin (ATAC-seq, blue), and chromatin interactions (Hi-C, red arcs) mapped onto the amp1 assembly (outer track, brown). Chromatin interactions occur across a structural breakpoint between accessible loci near highly-expressed genes such as *DNTTIP2*. (**b**) Chromatin interaction density map of the amp1 assembly. Arrows indicate putative enhancer rewiring events, or chromatin loops which span a breakpoint on the amp1 assembly. (**c**) Chromatin interaction density map of chr1. Dark stripes indicate that the ecDNA locus more frequently interacts with the rest of the genome, an indicator of high-copy focal amplification. (**d**) Transcription, accessible chromatin, and chromatin interactions mapped onto the amp2 assembly. The gap in the assembly is adjacent to pericentromeric and peritelomeric loci. (**e**) Chromatin interaction density map of the amp2 assembly. Putative enhancer hijacking events are again indicated by arrows. The two ends of this assembly do not interact (top of the triangle), suggesting that they are spatially distant in the cell. (**f**) Chromatin interaction map of chr7. The 'checkerboard' pattern reflects copy number amplification of 2 mutually exclusive structural variants amp2 and gain1, which may have originated from the same chromothriptic event.

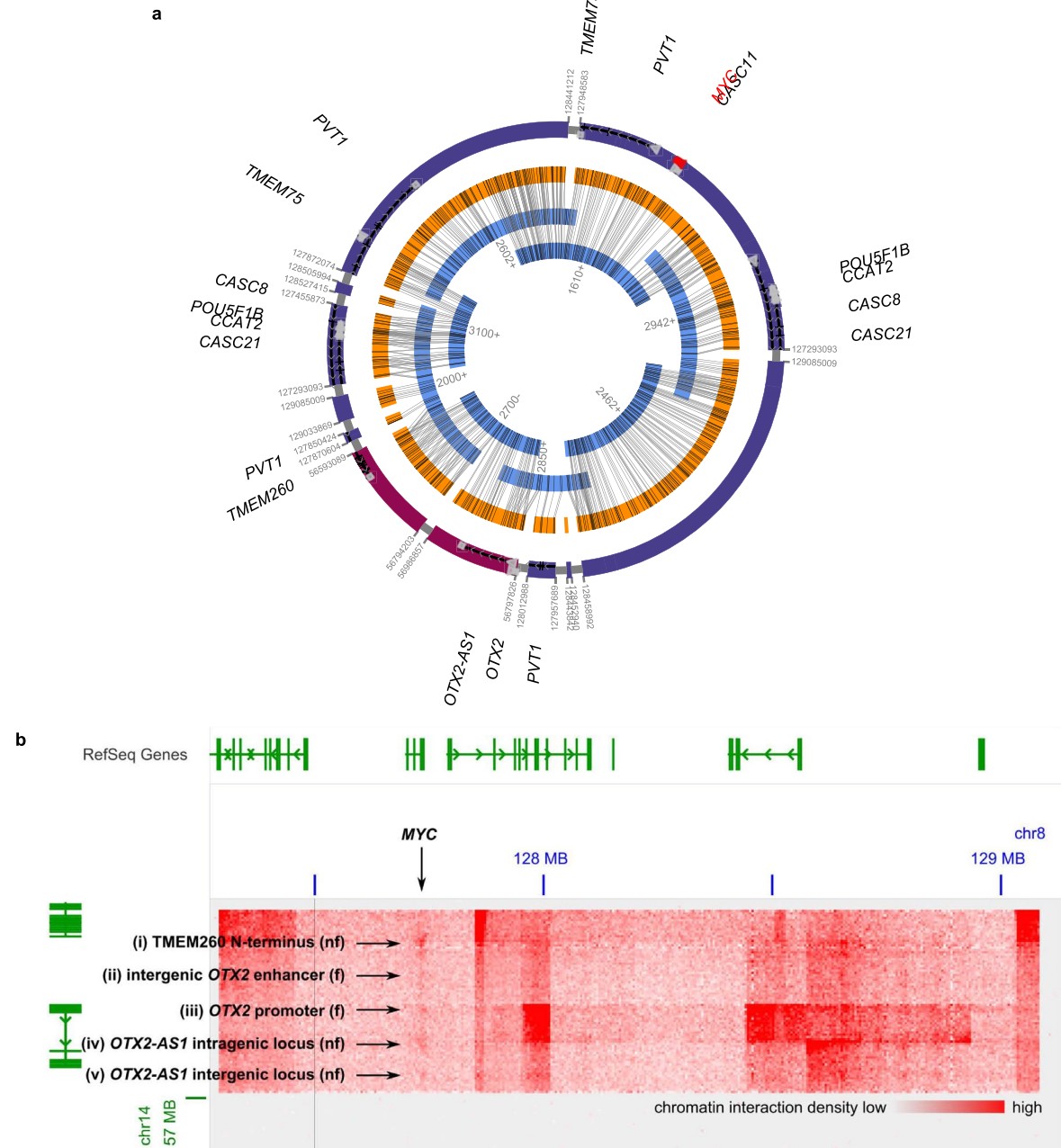

**Extended Data Fig. 8 | Sequence and conformation of the D458 ecDNA.**
(**a**) Reconstruction of the D458 ecDNA from OGM and WGS data. All junctions are supported by WGS discordant and optical genome mapping reads. (**b**) Chromatin interaction heatmap between co-amplified segments of chr8 and chr14 on the D458 ecDNA. Notable ectopic interchromosomal interactions are indicated here and in Fig. 4f. f: functional, as determined by a significant and D458-specific effect on cell proliferation upon CRISPRi inhibition (Fig. 4g); nf: not identified as functional in the same pooled CRISPRi screen.

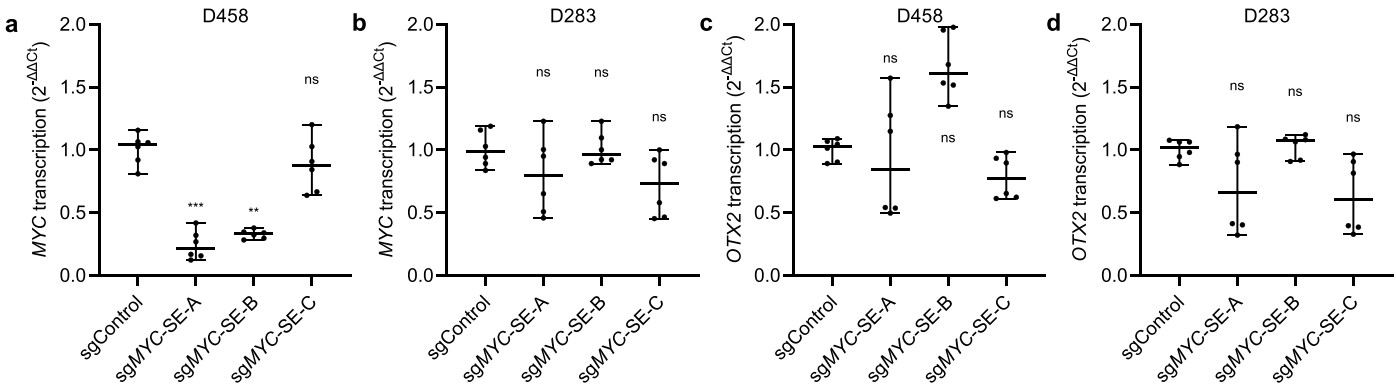

**Extended Data Fig. 9 | A co-amplified enhancer on the D458 ecDNA promotes MYC expression.** Relative expression of *MYC* (**a-b**), and *OTX2* (**c-d**), measured by qPCR ($2^{-\Delta\Delta Ct}$), in D458 and D283 cell lines upon CRISPRi targeting of an accessible locus within a known *MYC* superenhancer which promotes D458 proliferation (see also Fig. 4h). sgNT: nontargeting control; sg*MYC*-SE-A-C: sgRNAs targeting the *MYC* enhancer at D458_peak_30782, positions chr8:127330655, chr8:127330840, and chr8:127330927 (hg38) respectively. qPCR was performed on all guides in triplicate; each technical replicate is shown. Bars represent median +/− 95% CI. ** adjusted *p* = 0.002; *** adjusted *p* = 0.0008; one-sided nested ANOVA with Dunnett's correction.

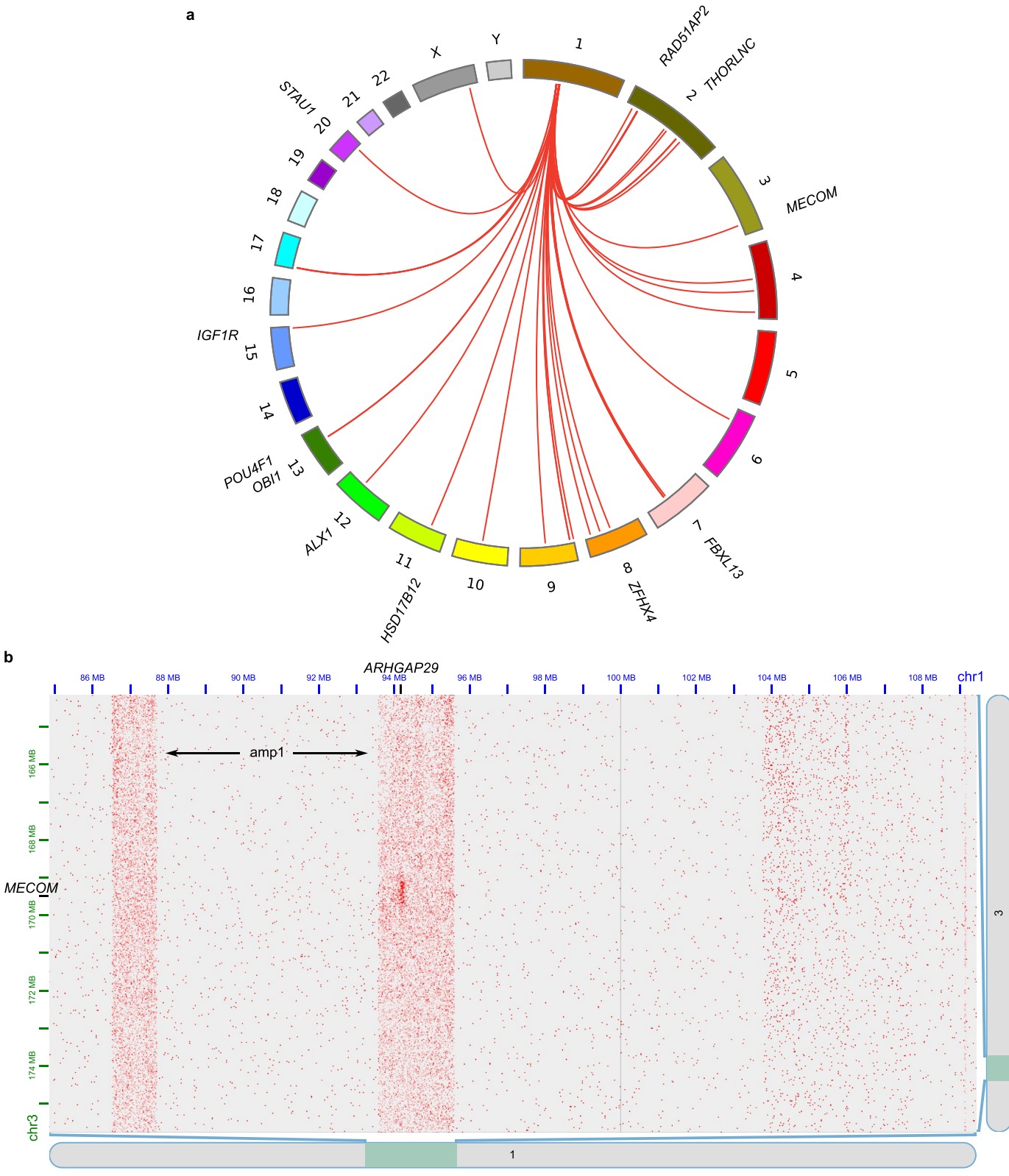

**Extended Data Fig. 10 | The RCMB56 amp1 ecDNA interacts with chromosomal gene loci.** (**a**) Interchromosomal chromatin interactions detected by FitHiC2 between amp1 and other chromosomes. Interactions mapping to a gene locus are labelled. (**b**) Interaction density map between segments of chr1 and chr3, rendered in Juicebox. Vertical stripes indicate increased contact density between the ecDNA and chr3 over background interactions between chr1 and chr3.

# Reporting Summary

## Statistics

For all statistical analyses, confirm that the following items are present in the figure legend, table legend, main text, or Methods section.

| n/a | Confirmed | |
|---|---|---|
| ☐ | ☒ | The exact sample size ($n$) for each experimental group/condition, given as a discrete number and unit of measurement |
| ☐ | ☒ | A statement on whether measurements were taken from distinct samples or whether the same sample was measured repeatedly |
| ☐ | ☒ | The statistical test(s) used AND whether they are one- or two-sided<br>*Only common tests should be described solely by name; describe more complex techniques in the Methods section.* |
| ☐ | ☒ | A description of all covariates tested |
| ☐ | ☒ | A description of any assumptions or corrections, such as tests of normality and adjustment for multiple comparisons |
| ☐ | ☒ | A full description of the statistical parameters including central tendency (e.g. means) or other basic estimates (e.g. regression coefficient) AND variation (e.g. standard deviation) or associated estimates of uncertainty (e.g. confidence intervals) |
| ☐ | ☒ | For null hypothesis testing, the test statistic (e.g. $F$, $t$, $r$) with confidence intervals, effect sizes, degrees of freedom and $P$ value noted<br>*Give P values as exact values whenever suitable.* |
| ☒ | ☐ | For Bayesian analysis, information on the choice of priors and Markov chain Monte Carlo settings |
| ☒ | ☐ | For hierarchical and complex designs, identification of the appropriate level for tests and full reporting of outcomes |
| ☐ | ☒ | Estimates of effect sizes (e.g. Cohen's $d$, Pearson's $r$), indicating how they were calculated |

*Our web collection on statistics for biologists contains articles on many of the points above.*

## Software and code

Policy information about availability of computer code

| Data collection | ICGC WGS data were downloaded using Score Client 5.0.0. sgRNAs targeting the D458 ecDNA regulome were designed using CHOPCHOP v3. |
|---|---|
| Data analysis | WGS for samples from RCH was processed using BWA v0.7.17-r118810, samtools v0.1.1911, Picard Tools v2.12.3, and GATK v3.8-1-012-14. ecDNA was identified and classified using AmpliconArchitect 1.2, CNVkit 0.9.6, AmpliconClassifier 0.4.4, and AmpliconReconstructor 1.01. Optical mapping assembly was performed using Bionano Solve 3.6. Survival analysis was performed using Lifelines 0.21.0. All other statistical tests were performed using scipy.stats 1.5.3. Multiple-hypothesis correction was performed using statsmodels 0.12.0. Visualizations were generated using circos 0.69-9, IGV desktop 2.9.2, Juicebox 1.11.08, and Seaborn 0.9.0. Genomic tracks were generated using bedtools v2.27.1, bedGraphToBigWig v4, and deeptools v3.5.1. FISH data were processed using NuSeT commit 37bcb9c and ecSeg-i commit 901ca79. Single-cell data were processed using CellRanger ARC 2.0.0, Seurat 4.0.4, DoubletFinder 2.0, PyRanges 0.0.112, ssGSEA 10.0.11, and InferCNV 1.3.3. ATAC-seq reads were trimmed using trimmomatic 0.36; quality-checked using fastqc 0.11.7; aligned using bowtie 2.3.4.3; indexed using samtools 1.10; and deduplicated using Picard Tools 2.20.8. ATAC-seq peaks were called using MACS2 2.1.2. Hi-C reads were trimmed using trimmomatic 0.39; aligned and processed using HiC-Pro 2.11.3-beta and bowtie 2.3.5; and normalized using Juicebox 1.11.08. Hi-C interactions were called using HiCCUPS 1.22.01 or FitHiC 2.0.8. qPCR data were analyzed using GraphPad Prism 9.5.2. Fingerprint analysis v1.1 is available at https://github.com/chavez-lab/fingerprint. Analysis code specific to this manuscript is available for review at https://github.com/auberginekenobi/medullo-ecdna, and https://github.com/auberginekenobi/ecdna-quant, https://github.com/auberginekenobi/rcmb56-single-cell. |

For manuscripts utilizing custom algorithms or software that are central to the research but not yet described in published literature, software must be made available to editors and reviewers. We strongly encourage code deposition in a community repository (e.g. GitHub). See the Nature Portfolio guidelines for submitting code & software for further information.

## Data

Policy information about availability of data

All manuscripts must include a data availability statement. This statement should provide the following information, where applicable:
- Accession codes, unique identifiers, or web links for publicly available datasets
- A description of any restrictions on data availability
- For clinical datasets or third party data, please ensure that the statement adheres to our policy

Whole genome sequencing data analyzed in this work are under controlled access, but are available from the following sources upon request:
- ICGC and Archer patient cohorts: International Cancer Genome Consortium (https://dcc.icgc.org/)
- CBTN patient cohort: Kids First Data Resource Center (https://kidsfirstdrc.org/)
- St Jude patient cohort: St Jude Cloud (https://www.stjude.cloud/)
- MB cell line and PDX models: requests for materials and manuscript correspondence should be directed to the corresponding author.
ATAC-seq, Hi-C, single cell sequencing, and pooled CRISPRi screen data will be available from NCBI Gene Expression Omnibus (GEO). ATAC-seq: [GEO accession here]. Hi-C: [GEO accession here]. scRNA+ATAC-seq: [GEO accession here]. CRISPRi: [GEO accession here]. FISH images are available at 10.6084/m9.figshare.c.6759093.

# Field-specific reporting

Please select the one below that is the best fit for your research. If you are not sure, read the appropriate sections before making your selection.

☒ Life sciences ☐ Behavioural & social sciences ☐ Ecological, evolutionary & environmental sciences

For a reference copy of the document with all sections, see nature.com/documents/nr-reporting-summary-flat.pdf

# Life sciences study design

All studies must disclose on these points even when the disclosure is negative.

| | |
|---|---|
| Sample size | No method was undertaken to predetermine sample size. Because medulloblastoma is a rare disease, we accessed all data and samples available to us. |
| Data exclusions | Sample ICGC_MB127 was predicted to be duplicate by fingerprinting analysis and was removed. Exclusion criteria were preestablished. |
| Replication | No replication experiments were performed. Two biological replicates of each cell line were grown for pooled and targeted CRISPRi experiments, and variance between replicates was addressed in subsequent linear models. |
| Randomization | Random allocation was not relevant to our patient data because no treatment/control experiments were performed. |
| Blinding | Blinding was not relevant in our study since experimental validation was focused on specific tumor cell lines with limited variance e.g. only 2 cell lines available per group. |

# Reporting for specific materials, systems and methods

We require information from authors about some types of materials, experimental systems and methods used in many studies. Here, indicate whether each material, system or method listed is relevant to your study. If you are not sure if a list item applies to your research, read the appropriate section before selecting a response.

## Materials & experimental systems

| n/a | Involved in the study |
|---|---|
| ☒ | ☐ Antibodies |
| ☐ | ☒ Eukaryotic cell lines |
| ☒ | ☐ Palaeontology and archaeology |
| ☐ | ☒ Animals and other organisms |
| ☐ | ☒ Human research participants |
| ☒ | ☐ Clinical data |
| ☒ | ☐ Dual use research of concern |

## Methods

| n/a | Involved in the study |
|---|---|
| ☒ | ☐ ChIP-seq |
| ☒ | ☐ Flow cytometry |
| ☒ | ☐ MRI-based neuroimaging |

## Eukaryotic cell lines

Policy information about cell lines

| | |
|---|---|
| Cell line source(s) | Cell lines D458 and D283 were a gift from the lab of Jae Cho (OHSU). 293T cells were purchased from ATCC (Cat# CRL-3216). |

| Authentication | Data obtained from all cell lines were consistent with previously published knowledge of these cell lines. STR testing was performed for all samples received from external labs and matched to public STR profiles for those cells. |
| --- | --- |
| Mycoplasma contamination | All cell lines tested negative for mycoplasma contamination. |
| Commonly misidentified lines (See ICLAC register) | No commonly misidentified cell lines were used. |

# Animals and other organisms

Policy information about studies involving animals; ARRIVE guidelines recommended for reporting animal research

| Laboratory animals | We used immunodeficient NSG mice (NOD.Cg-Prkdcscidll2rgtm1Wjl/SzJ, The Jackson Laboratory #005557) for RCMB56 PDX intracranial implants and tumor harvests. 5 male mice between 6 and 12 weeks old were used for each experiment. |
| --- | --- |
| Wild animals | This study did not involve wild animals. |
| Field-collected samples | This study did not involve samples collected in the field. |
| Ethics oversight | All experiments were performed in accordance with national guidelines and regulations, and with the approval of the the Institutional Animal Care and Use Committees (IACUC) at the Sanford Burnham Prebys Medical Discovery Institute and University of California San Diego (AUF19-055 and S12123, respectively) and the UCSD Institutional Review Board (Project #171361XF). |

Note that full information on the approval of the study protocol must also be provided in the manuscript.

# Human research participants

Policy information about studies involving human research participants

| Population characteristics | In total the WGS cohort comprised 481 medulloblastoma tumors from 468 patients (161 female, 277 male, 30 N/A; ages 0-36; see Supplementary Table 1). |
| --- | --- |
| Recruitment | No participants were recruited directly for this study. All human data were accessed or generated according to patient consents for general research use. Because sample metadata were compiled in part from peer-reviewed manuscripts including one specifically addressing SHH MB, the set of samples with unknown subgroup may be modestly enriched for WNT, G3, and G4 subgroups. We do not anticipate this will affect results. |
| Ethics oversight | Protocols were approved by Institutional Review Boards (IRB) affiliated with the University of California San Diego and Sanford Burnham Prebys Medical Discovery Institute, and Data Access Committees (DAC) from the International Cancer Genome Consortium (ICGC), St. Jude Children's Hospital, and the Children's Brain Tumor Network (CBTN). |

Note that full information on the approval of the study protocol must also be provided in the manuscript.

