## [Peer Review File · Nature Genetics]

Peer Review Information

Manuscript Title: Sequencing data deposits are currently in progress at GEO and SRA. We will furnish accession IDs when they are generated.

Corresponding author name(s): Dr Lukas Chavez

Reviewer Comments & Decisions:

Decision Letter, initial version:
--

19th Jul 2022

Dear Dr Chavez,

Your Article entitled "EcDNA promotes intratumoral heterogeneity and tumorigenicity in medulloblastoma" has now been seen by 3 referees, whose comments are attached. While they find your work of potential interest, they have raised serious concerns which in our view are sufficiently important that they preclude publication of the work in Nature Genetics, at least in its present form.

While the referees find your work of some interest, they raise concerns about the strength of the novel conclusions that can be drawn at this stage.

Should further experimental data allow you to fully address these criticisms we would be willing to consider an appeal of our decision (unless, of course, something similar has by then been accepted at Nature Genetics or appeared elsewhere). This includes submission or publication of a portion of this work someplace else.

The required new experiments and data include, but are not limited to those detailed here. We hope you understand that until we have read the revised manuscript in its entirety we cannot promise that it will be sent back for peer review.

If you are interested in attempting to revise this manuscript for submission to Nature Genetics in the future, please contact me to discuss a potential appeal. Otherwise, we hope that you find our referees' comments helpful when preparing your manuscript for resubmission elsewhere.

Sincerely,

Safia Danovi
Editor
Nature Genetics

Referee expertise:

Referee #1: ecDNA

Referee #2: medulloblastoma genomics

Referee #3: paediatric cancers, computational biology

Reviewers' Comments:

Reviewer #1:

Remarks to the Author:

In their manuscript „ecDNA promotes intratumoral heterogeneity and tumorigenicity in medulloblastoma” Chapman et al describe the frequency, structure and inter-cellular differences in ecDNA abundance in medulloblastomas from different clinical subgroups. To do so, the authors analyzed a previously published, well-curated cohort of medulloblastoma genomes and applied state-of-the-art ecDNA detection algorithms. Analyses of the regulatory landscape and chromatin interactions on ecDNA combined with CRISPRi screening for dependencies in MB cell lines identified regulatory elements required for sustained oncogene expression on ecDNA and their rewiring. Notably, multiple distinct ecDNA structures were detected to co-exist in a subset of tumors. Of particular novelty and interest to the field, the authors describe inter-cellular differences in ecDNA frequency in one medulloblastoma using single cell RNA and ATAC sequencing and show that ecDNA copy number correlates with genes' expression from ecDNA.

In general, this manuscript is very timely and describes important features of ecDNA in medulloblastoma. The amount of data presented here is impressive and I applaud the authors for their efforts. However, the number of different datasets presented here is also somewhat overwhelming to the reader, especially because some of the datapoints are not well interconnected throughout the manuscript. The novelty is reduced by the fact that most of the results presented here reproduce observations about ecDNA, which were previously reported in other cancer entities. Eg. the Verhaak laboratory nicely showed that ecDNA+ cancers have dismal clinical outcomes, the Mischel laboratory and Scacheri laboratory nicely showed that regulatory elements on ecDNA can be re-wired and are required for oncogene expression. Even though the reproduction of these observations in medulloblastoma is an important achievement and deserves to be published, many of the findings remain descriptive and are listed in this manuscript without being logically/biologically connected to each other. Additionally, some of the analyses were performed in small cohorts (eg. single cell sequencing or optical mapping), which reduces the robustness of some conclusions. Even though this manuscript is of considerable interest to the field, there are additional limitations listed below that need to be addressed, before this manuscript is suitable for publication.

Major points:

1. As mentioned above, the manuscript lacks focus. Many ecDNA features were analyzed, but most of them are described with little detail. For example, the CRISPRi screen alone is a major datapoint, but is only very briefly mentioned in the text and figures. In my very personal view, the manuscript would greatly benefit from excluding some of the datapoints and focusing on describing some of the discoveries in more detail.

2. The authors analyzed the presence of ecDNA using AmpliconArchitect (AA). If I understand correctly, AA can also report on other amplicon structures, incl. linear forms such as for example breakage fusion bridge cycles (BFB). The authors do not report the frequency of these linear amplicons in their manuscript (or at least I was not able to find that information) and do not compare their impact on clinical outcome with that of ecDNA. This significantly reduces the support of their conclusion that ecDNA impacts medulloblastomas' clinical outcome, as this could also just be due to the mere presence of any type of oncogene amplification, regardless of whether it is linear/intra-chromosomal or extrachromosomal. It has repeatedly been reported that MYC and MYCN amplifications are associated with poor outcome in MBs, which is why these features are used to stratify MBs into their risk groups. Without the above comparison of linear vs. circular amplification, similar as done in a publication by Kim et al (Nature Genetics 2020, Figure 4 a and b) one of the main conclusions by the authors is not sufficiently supported.
3. As mentioned above, one of the authors' main conclusions is that ecDNA+ medulloblastomas have a particularly poor prognosis. Even though part of their data support this conclusion, the analysis lacks important controls. Eg. their Cox proportional hazards model is parametrized on sex, age and the molecular subgroups, but the authors did not perform the analysis for other known molecular parameters associated with poor prognosis in MBs, eg. TP53 status, MYC amplification and others. This is especially important because the authors note that TP53 mutations are associated with ecDNA in SHH MBs (something that was previously known).
4. The findings on enhancer rewiring are nice and are in line with previous observation by Morton et al and Helmsauer et al. The CRISPRi screen is very elegant and the results support the authors conclusions. Helmsauer et al (Nature Communications 2020) analyzed the distribution of amplicon boundaries in MB subgroups (see supplementary Fig. 3 in their paper) and found enhancers to be included on these amplicons. This paper did not have information about the status (ecDNA vs. linear) of these amplicons in medulloblastomas. The authors of the current manuscript may want to compare ecDNA+ vs. linear amplifications for the presence of these enhancers to see if they can confirm these previously published results. Are the enhancers that the authors found to be required for oncogene expression using CRISPRi the same as those observed to be included in medulloblastoma amplicons?
5. Even though the analysis of ecDNA's structural evolution is interesting, it is quite preliminary. Shoshani et al (Nature 2021) and Rosswog et al (NatGen 2021) amongst others have nicely shown how chromothripsis can contribute to ecDNA evolution. The authors here show three examples for ecDNA evolution, all of which show different changes. Due to the limited number of samples the authors cannot derive any biological conclusions from their observations. I would suggest either expanding the cohort of serial biopsied samples or omitting/shortening this part from this manuscript.
6. The authors nicely reproduce the previously reported association of chromothripsis, ecDNA with TP53 loss in SHH MBs (Rausch et al., Cell 2012), but mention that no association of ecDNA with TP53 loss was observed in other risk groups. Are there any other molecular factors associated with ecDNA presence in the other risk groups? One might expect to find other means of TP53 inactivation such as MDM2 amplification etc.
7. The use of optical mapping with NGS is elegant, but it remains unclear why the authors did not use other means of validating the presence and structure of ecDNAs in their cohort, eg. FISH, CRISPR-CATCH (<https://www.biorxiv.org/content/10.1101/2021.11.28.470285v1>) or exonuclease-based enrichment protocols.
8. The single cell results are fascinating, but unfortunately remains rather preliminary. The authors only analyzed one tumor, raising the question of how generalizable their findings are. Relatively few analyses are done to look at the structural heterogeneity in single cells, which is something the authors had observed in bulk sequencing. As with the WGS data, the authors did not validate the presence and heterogeneity of ecDNA in their sample using orthogonal methods, eg. FISH or others.

Minor comments:

1. The title strongly focusses on intratumoral heterogeneity, but the results presented here are mainly based on bulk sequencing. There is also no direct evidence presented here that would support that ecDNA "promotes tumorigenicity". I would suggest refining the title to better reflect the manuscript's main conclusions.
2. The abbreviation ecDNA is usually used to refer to extrachromosomal DNA and not extrachromosomal circular DNA, which is usually abbreviated eccDNA. I would suggest changing this as it risks to cause some confusion amongst readers.
3. In Fig 1b there is a row that indicates "p53 inhibitors". It is unclear what this refers to.
4. Chang and Verhaak and others proposed the ecDNA could interact with the linear genome, and termed these ecDNA "mobile enhancers". Did the authors observe any contacts of ecDNA with the linear chromosomes in trans in their HiC data? I understand that this may be outside the scope of this manuscript, but it would be of great interest to the field.
5. The description of ecDNA in patient derived models is placed in the same figure describing single cell sequencing data, which is quite confusing to the reader and not easy to follow.

Reviewer #2:

Remarks to the Author:

The manuscript 'EcDNA promotes intratumoral heterogeneity and tumorigenicity in medulloblastoma' by Chapman et al, focuses on the detection of extrachromosomal circular DNA (or double minutes) across a large number of human medulloblastomas and a smaller number of medulloblastoma cell lines and PDXs, using predominantly previously published datasets. While the overall theme of ecDNA is interesting, I am concerned that the analyses presented in this manuscript superficially span several different areas/themes (for example spanning analyses in primary tumors, cell lines and then focusing on evolution of PDXs) rather than going 'deep' into answering any one specific question. As such, the findings overall don't significantly improve our understanding of the pathogenesis of medulloblastoma and, in my opinion, the analyses unfortunately do not reveal insights about the mechanisms through which ecDNA are generated and contribute to tumorigenesis.

Below are some comments and suggestions that I hope the authors will find helpful.

1. Line 81. The authors claim that the absence of ecDNA may account for their good prognosis. The observation that there is low ecDNA in WNT is an association with the WNT subgroup and doesn't necessarily mean that the presence of ecDNA has prognostic significance. If the authors do not prove causality between presence of ecDNA and poor prognosis.
2. Similarly, the cox proportional hazards model described in line 88 only take into account sex, age and molecular subgroup and find an association with ecDNA and prognosis. However, other important variables that may track with ecDNA (and thus may confound the analysis) are not taken into account. In particular, TP53 mutations come to mind as these are found more commonly in SHH subgroups (which the authors report have higher amounts of ecDNA) – and TP53 mutations are associated with a dismal prognosis. Multivariate analyses should take into account known variables that have prognostic significance in medulloblastoma. This is particularly important as the authors later show that TP53 mutations are also associated with ecDNA in SHH medulloblastomas.

3. Line 83. The authors highlight several genes that are involved in ecDNA, many of which have been shown to be amplified in medulloblastomas. What proportion of tumors do these amplicons involve ecDNA vs amplification. Are there any associations with other driver genes that may determine suggest different mechanisms through which these genes are amplified vs incorporated into ecDNA? Are there any other biological differences between these groups? Are there regions in the genome that are more likely to be involved in ecDNA more frequently than one would expect by chance?

4. Line 92 on: the authors describe three primary -relapsed tumors and in each case, characterize the differences in ecDNA between the primary and recurrent tumor. However, this analysis is performed in isolation of other genetic features (somatic nucleotide variants, copy-number, other structural variants etc) and as such it is difficult to ascertain the significance in changes in ecDNA as being driver vs passenger events. Moreover, based on this limited analysis the authors conclude that sequence rearrangement or de novo formation of ecDNA may be common in medulloblastoma. This is a very strong conclusion based on limited and descriptive analyses of three tumors.

5. Line 107. The authors show an association between TP53 mutations and ecDNA. Did the authors perform a broader multivariate analysis to identify other molecular features that are associated with ecDNA in medulloblastoma?

6. Lines 117 on. The authors perform 'optical mapping' of ecDNA in two medulloblastoma cell lines generated from the same patient (primary and metastasis) many decades ago, and four PDX models generated more recently. The authors first show genetic evolution of D425 and D458 based on ecDNA analyses. Technically this is interesting, but biologically not surprising, given how old these models are and the many years that they have been in culture. Description of the analyses of ecDNA in the PDXs is very limited and descriptive and it is hard to determine what conclusion(s) the authors derive from this analysis.

7. The authors perform analyses to show that DNA breakpoints connect gene and enhancer loci co-amplified in ecDNA in at least half of ecDNA positive tumors. This is not surprising as structural variants in general have been well documented to commonly involve gene regulatory elements. If the authors were to compare the frequency of these interactions in non-ecDNA structural variants vs ecDNA, is there an enrichment in these interactions in ecDNA?

8. Line 171 onwards. The authors leverage CRISPRi to show that the enhancers associated with ecDNA in the D458 cell line exert proliferative advantage. While this is an important experiment, in its current form, it doesn't show specificity of the ecDNA enhancers as being essential for cell proliferation. For example, what if the guides targeted other enhancers on D458 that were not part of the ecDNA? These enhancers are not necessary for D283 however the structural variant landscape of this cell line is different to D458. Even in the absence of ecDNA of D283, I suspect that if the authors had designed a library to target enhancers involved in other structural variants present in D283 that they would see a similar phenotype. As such, I am concerned that this experiment shows that enhancers are necessary in D458 rather than enhancers specifically involved in ecDNAs present in D458.

9. Line 230 onwards. This section compares ecDNA in a PDX to a human tumor. This section doesn't really add anything to the role of ecDNA in generating medulloblastomas and shows fidelity of one PDX to its primary tumor. Similarly, the final section of the results show that ecDNA is more enriched in PDXs than primary tumors. Again, this doesn't provide insights into the role of ecDNA in tumor formation. If the authors want to examine the relevance of PDXs to human tumors, I suggest that this

is presented in a separate manuscript.

Reviewer #3:

Remarks to the Author:

The manuscript entitled "EcDNA promotes intratumoral heterogeneity and tumorigenicity in medulloblastoma" by Dr. Chavez and colleagues presents a comprehensive analysis on the landscape of ecDNA based on the analysis 481 medulloblastoma whole-genome sequencing data. The authors discovered a prevalence of 18% ecDNA+ cases in their cohort and found ecDNA+ status is significantly associated with relapse and mortality of medulloblastoma, the most common pediatric brain tumor. The complex structure of ecDNA was delineated by computational modeling (AmpliconArchitect) and integration of long-read optical mapping. Based on the ecDNA structure defined by these approaches, the authors explored the significance of co-amplified enhancers by performing CRISPRi screening on open chromatin regions in ecDNA and discovered 6 enhancers enriched in the ecDNA tumor sample. Finally, they examined the heterogeneity of ecDNA in primary tumor and PDX models and analyzed ecDNA evolution from patient sample to PDX model by performing dual single cell (sc) assay of scRNA-seq and scATAC-seq and inferring ecDNA status from scATAC-seq.

The study is very interesting in highlighting the role of ecDNA in the pathogenesis of medulloblastoma. Furthermore, the high-quality data presented in this study, in particular optical mapping, unveiled new architecture of ecDNA which is lacking in current literature. Given that ecDNA research is a relatively new field, further analysis on the extensive data sets generated from this study can potential help elucidating some of the recently reported features of ecDNA, which can broaden the impact of the current work.

Specific comments are listed below.

Major comments:

1) In the section "Co-amplified enhancers shape the transcriptional regulatory circuitry of ecDNA in D458 cells", the authors presented CRISPRi screen of 32,530 guides targeting all 645 accessible chromatin regions on the D458 ecDNA. The same CRISPRi screen was performed on D283 cell line as a control because D283 has MYC HSR but no amplification of OTX2. The authors reported 6 D458-specific functional elements which were highlighted in Figure 2h. There are several points that will require clarification: a) Is MYC or OTX2 suppressed in the CRISPRi cells derived from D458 but not in those of D283?; b) do these loci interact with MYC/OTX2 promoter specifically in D458 based on Hi-C data? c) Was OTX2 also over-expressed in D283? If not, what the molecular basis for reduced growth of D283 cells when the yellow highlighted site in OTX2 (last exon) was affected by CRISPRi? d) Does the green site at PVT1 promoter affect the expression of both cMYC and PVT1? Without addressing these questions, the mechanism leading to reduced proliferation of D458 remains unclear. These questions can be addressed using existing data or performing assays on cells harvested after 21 days in D458 are CRISPRi inhibition.

2) scATAC-seq read-depth was used to estimate the ecDNA+ cells. Additional verification (e.g. FISH) is needed to validate the estimated prevalence of ecDNA+ cells patient tumor or PDX of RCMB56. Are there any somatic mutations outside the ecDNA amplicon that co-segregate with the two ecDNAs identified in this case?

3) Koche et al (Nature Genetics volume 52, pages29–34 (2020)) reported that ecDNA re-integration is common in neuroblastoma. Is this phenomenon (ecDNA re-integration) also found in

medulloblastoma? Duplication of genomic fragments can occur on ecDNA itself as shown in Figure 2f. How does a pattern of within-episome duplication differ from ecDNA re-integration?

4) Given that Hi-C was performed for several samples, are there any data supporting the role of ecDNA as a mobile enhancer as described by Zhu et al in Cancer Cell. 2021 May 10;39(5):694-707? Minor comments:

1) Survival analysis on ecDNA+ versus ecDNA- patients described in line 88-89. Would the authors comment whether the significance can be achieved when considering molecular subtype? Supplementary Figure 1 presented this information, but it will be good to make an explicit statement in the main text.

2) Was the association of TP53 mutation with ecDNA reported in other tumor types? If so, please add a reference in the Discussion.

3) Figure 2a: what is the genomic location? What does the arrow mean in this panel? Do they represent the circularized loci described in lines 147-148?

4) Figure 2b & f: the RNA-seq & ATAC-seq tracks in the circular view are barely visible and occupy a big ring with extensive white space. It would be good to convert the scale to a heatmap so that the footprint of gene expression/ATA-seq can be clearer and Hi-C interaction can be presented with more clarity. There is also a thin gray track in the DNA ring with no annotation which will be good to describe in figure legend. Does the gaps in the ring represent unknown sequence fragments? It will be good to clarify these in figure legend. Panel f appears to contain duplicated regions which need to be labeled or described in figure legend.

5) Lines 151-152: "DNA interactions occurred between the MYC locus and two co amplified enhancer regions located 13Mbp away on the linear genome". Does the authors referring distance based on the reference genome? If so, it will also be important to describe their physical distance on the re-arranged episome. Presumably the two enhancer regions are those marked as ZC3H3 and PTP4A3 on figure 2c; however, the It is hard to recognize these regions as enhancers given the difficult representation in Figure 2b.

6) Line 167-168: "The Hi-C data revealed a myriad of intra- and inter- chromosomal interactions of the MYC promoter with co-amplified regulatory elements of chr8 and chr14 (Fig. 2f)." Given the duplications of PVT1-related regions on the ecDNA, would it be possible that some of the interactions were caused by the ambiguity in determining the contact sites amongst the duplicated regions?

7) Line 168-169: "Thus, we present 4 MB samples in which 169 aberrant enhancer-promoter interactions occur across structural breakpoints on ecDNA (Fig. 2g)." Figure 2g does not show 4 MB samples. It is a FISH image of D283 showing HSR MYC amplification. Similarly, Fig. 2h on lines 176 and 177 should be changed to Fig. 2g

8) Figure 2h: the enrichment on OTX2 last exon in D283 is perplexing as chr14 is not involved in D283 HSR. The authors should provide an explanation.

9) Supplementary Fig. 3h: in the description of line 196-197, the gap on ecDNA2 was referred as peritelomeric while the region is labeled as pericentromeric. Please clarify whether this is a centromeric or telomeric region.

10) Line 202-203: "These data also revealed low-level amplification of other segments of chr7 and chr17, of 203 total length 35.2Mbp, which we hypothesized may be a third low-copy ecDNA (ecDNA3 (Supplementary fig. 6b).)" —it is not clear how a 3rd ecDNA can be inferred from Sup. Fig. 6b. The authors need to provide more description on how this was inferred in figure legend.

11) Figure 4: ecDNA2-only ATAC-seq are not shown in Figure 4a. Is this due to the low number of this population? This should be mentioned in figure legend as well as in the description at line 218-219 as the current narrative would have left the misconception that ecDNA1-edDNA2+ is no a minor population.

12) Figure 4: what does MT-high tumor stands for? Mitochondria?

13) Line 328-329 "WGS was preprocessed and aligned according to internal pipelines at St Jude (hg38)". The analysis pipeline was described in the Method section of McLeod et al, 2021 (ref 28). Please revise the statement as "WGS was preprocessed and aligned by bwa-mem to hg38 as described in McLeod et al".

14) Line 396 and 397. St. Jude Pediatric Cancer Genome Project (PCPG)—abbreviation should be PCGP

15) The presentation of intra-tumor heterogeneity of ecDNA should mention some earlier work such as those published by Ke et al (Structure and evolution of double minutes in diagnosis and relapse brain tumors, PMID 30267146)

Decision Letter, Appeal:

24th Apr 2023

Dear Dr. Chavez,

Thank you for your message of 24th Apr 2023, asking us to reconsider our decision on your manuscript "Circular extrachromosomal DNA promotes inter- and intratumoral heterogeneity in high-risk medulloblastoma".

Please accept my apologies for the delay in getting back to you. I have now discussed the points of your letter with my colleagues, and we think that you have some valid points. We therefore invite you to revise your manuscript along the lines that you propose.

When preparing a revision, please ensure that it fully complies with our editorial requirements for format and style; details can be found in the Guide to Authors on our website (<http://www.nature.com/ng/>).

Please be sure that your manuscript is accompanied by a separate letter detailing the changes you have made and your response to the points raised. At this stage we will need you to upload:

- 1) a copy of the manuscript in MS Word .docx format.
- 2) The Editorial Policy Checklist:

<https://www.nature.com/documents/nr-editorial-policy-checklist.pdf>

3) The Reporting Summary:

(Here you can read about the role of the Reporting Summary in reproducible science:

<https://www.nature.com/news/announcement-towards-greater-reproducibility-for-life-sciences-research-in-nature-1.22062>)

Please use the link below to be taken directly to the site and view and revise your manuscript:

[redacted]

With kind wishes,

Safia Danovi

Editor

Nature Genetics

Author Rebuttal to Initial comments

Reviewer #1

Remarks to the Author:

In their manuscript „ecDNA promotes intratumoral heterogeneity and tumorigenicity in medulloblastoma” Chapman et al describe the frequency, structure and inter-cellular differences in ecDNA abundance in medulloblastomas from different clinical subgroups. To do so, the authors analyzed a previously published, well-curated cohort of medulloblastoma genomes and applied state-of-the-art ecDNA detection algorithms. Analyses of the regulatory landscape and chromatin interactions on ecDNA combined with CRIPSRi screening for dependencies in MB cell lines identified regulatory elements required for sustained oncogene expression on ecDNA and their rewiring. Notably, multiple distinct ecDNA structures were detected to co-exist in a subset of tumors. Of particular novelty and interest to the field, the authors describe inter-cellular differences in ecDNA frequency in one medulloblastoma using single cells RNA and ATAC sequencing and show that ecDNA copy number correlates with genes' expression from ecDNA.

In general, this manuscript is very timely and describes important features of ecDNA in medulloblastoma. The amount of data presented here is impressive and I applaud the authors for their efforts. However, the number of different datasets presented here is also somewhat overwhelming to the reader, especially because some of the datapoints are not well interconnected throughout the manuscript. The novelty is a reduced by the fact that most of the results presented here reproduce observations about ecDNA, which were previously reported

in other cancer entities. Eg. the Verhaak laboratory nicely showed that ecDNA+ cancers have dismal clinical outcomes, the Mischel laboratory and Scacheri laboratory nicely showed that regulatory elements on ecDNA can be re-wired and are required for oncogene expression. Even though the reproduction of these observations in medulloblastoma is an important achievement and deserves to be published, many of the findings remain descriptive and are listed in this manuscript without being logically/biologically connected to each other. Additionally, some of the analyses were performed in small cohorts (eg. single cell sequencing or optical mapping), which reduces the robustness of some conclusions. Even though this manuscript is of considerable interest to the field, there are additional limitations listed below that need to be addressed, before this manuscript is suitable for publication.

Major points:

1. As mentioned above, the manuscript lacks focus. Many ecDNA features were analyzed, but most of them are described with little detail. For example, the CRISPRi screen alone is a major datapoint, but is only very briefly mentioned in the text and figures. In my very personal view, the manuscript would greatly benefit from excluding some of the datapoints and focusing on describing some of the discoveries in more detail.

We would like to thank the reviewer for recognizing the strengths of our manuscript. Moreover, it was of great value that the reviewer pointed out some lack of focus, which motivated us to deepen the manuscript on several topics with additional experiments and analyses and to exclude some other, tangential points. In particular, we have narrowed the scope of this manuscript by:

- Excluding the detailed comparison of ecDNA found in cell lines D425 and D458 (M.P. 2.6);
- Excluding the analysis of ecDNA evolution in patient tumors that was based on a limited sample of 4 primary/relapse sample pairs (M.P. 1.5) and one paired primary/pdx sample (M.P. 2.9);
- Excluding the retrospective analysis of ecDNA in PDX and cell line models of MB. We now only describe RCMB56-pdx and D458, which were used extensively for characterizations of intratumoral heterogeneity and enhancer rewiring.

In addition, we have expanded upon the following topics with additional analyses, based on feedback from reviewers:

- Comparison of survival and distribution of gene amplifications of ecDNA+ tumors, relative to other classes of focal somatic copy number amplification (M.P. 1.2);
- Survival analysis of ecDNA+ tumors, accounting for known prognostic indicators in MB (M.P. 1.3, M.P. 2.2);

- Orthogonal validations of intra-tumoral heterogeneity of ecDNA copy number by an automated FISH image analysis workflow applied to a series of patient tissues (M.P. 1.7 and 1.8);
- Established a new collaboration with Dr. Howard Y. Chang at Stanford University to further validate circular amplifications in MB using CRISPR-CATCH, a recently published method for targeted profiling of ecDNA (Hung *et al.*, Nature Genetics 2022).

We think that the revised manuscript has a much clearer focus on the crucial new findings and future implications for medulloblastoma research and clinical management.

2. The authors analyzed the presence of ecDNA using AmpliconArchitect (AA). If I understand correctly, AA can also report on other amplicon structures, incl. linear forms such as for example breakage fusion bridge cycles (BFB). The authors do not report the frequency of these linear amplicons in their manuscript (or at least I was not able to find that information) and do not compare their impact on clinical outcome with that of ecDNA. This significantly reduces the support of their conclusion that ecDNA impacts medulloblastomas' clinical outcome, as this could also just be due to the mere presence of any type of oncogene amplification, regardless of whether it is linear/intra-chromosomal or extrachromosomal. It has repeatedly been reported that MYC and MYCN amplifications are associated with poor outcome in MBs, which is why these features are used to stratify MBs into their risk groups. Without the above comparison of linear vs. circular amplification, similar as done in a publication by Kim *et al* (Nature Genetics 2020, Figure 4 a and b) one of the main conclusions by the authors is not sufficiently supported.

We agree that the stratification of patients with different types of amplifications is necessary and was missing in the original manuscript. Therefore, we have now re-analyzed our cohort survival data with respect to the amplification classes returned by the AmpliconClassifier tool, including

(1) ecDNA, (2) breakage-fusion-bridge (BFB), (3) linear, (4) complex non-cyclic ('heavily rearranged'), and (5) no focal somatic copy number amplification ('no fSCNA'). Similarly to the results of Kim *et al.* across the PCAWG cohort (Nature Genetics 2021), we find that ecDNA+ MB tumors have the worst survival of all classes. These results are now reported in the text of our manuscript (see section "EcDNA predicts poor prognosis in medulloblastoma") and in **Supplementary Fig. 2**.

3. As mentioned above, one of the authors' main conclusions is that ecDNA+ medulloblastomas have a particularly poor prognosis. Even though part of their data support this conclusion, the analysis lacks important controls. Eg. their Cox proportional hazards model is parametrized on sex, age and the molecular subgroups, but the authors did not perform the analysis for other known molecular

parameters associated with poor prognosis in MBs, eg. TP53 status, MYC amplification and others. This is especially important because the authors note that TP53 mutations are associated with ecDNA in SHH MBs (something that was previously known).

We would like to thank the reviewer for this comment as the revised manuscript, through our additional analyses, more clearly illustrates the relationship of ecDNA and other genomic alterations common in MB (Results subsections “ecDNA predicts poor prognosis in medulloblastoma” and “TP53 alterations are associated with ecDNA in SHH MB tumors”, lines 104-142).

Most regression models, including Cox Proportional Hazards (CPH) analysis, assume independence between model parameters, and cannot reliably estimate hazards of highly correlated variables. In our dataset, *TP53* mutation status is (nearly) a linear combination of ecDNA status and molecular subgroup, because WNT tumors with *TP53* mutation universally do not have ecDNA and SHH tumors with *TP53* mutation (nearly) universally have ecDNA. To address this issue comprehensively, we have consulted with a biostatistician at the Moores Cancer Center of UCSD and included 2 new analyses in our revision. Firstly, we perform accelerated failure time (AFT) regressions describing a classical mediation model in which part of the effect of *TP53* mutation on survival is indirect via ecDNA (new **Supplementary Fig. 3**). Consistently with this proposed mechanism, we find that much of the total effect of *TP53* mutation on survival may be explained by an effect of ecDNA on survival and frequent co-occurrence of *TP53* mutation and ecDNA in the same tumor, Secondly, we now report additional CPH models of survival which include *TP53* mutation as a covariate, controlling for model instability due to collinearity using ridge regression (new **Supplementary Fig. 4**). The effect of ecDNA on survival remains significant, suggesting that there is a contribution of ecDNA to patient outcome that is independent of *TP53* mutation. These analyses offer a reasonable explanation for the associations we observe between *TP53*, ecDNA and patient outcomes.

In the case of *MYC* family amplification, the problem of collinearity again arises when estimating independent hazards for ecDNA amplification and *MYC* family amplification. To illustrate, we have separately performed Kaplan-Meier survival analysis on the subsets of our cohort with *MYC* amplification, with respect to the class of focal amplification (**Response Fig. 1** below). In neither case did we have a large enough sample to meaningfully distinguish between the survival outcomes of the different amplification classes, in part because most *MYC* family amplifications were amplified on ecDNA. Although we do not have the power to distinguish between possible independent effects of ecDNA and *MYC* family amplifications, we emphasize here and in the

revised manuscript that ecDNA is a frequent vehicle for oncogenic amplification, including of

oncogenes such as *MYC* and *MYCN* with known associations with poor patient prognosis in MB.

Response Figure 1: Kaplan-Meier survival analysis of *MYC*-amplified MB for which clinical survival data were available. The *other* class comprises linear and highly rearranged amplifications.

In summary, these results show that there is a strong association between ecDNA and known genomic alterations in MB. Most importantly, our results reveal for the first time that known MB oncogenes are frequently amplified in the form of ecDNA in MB patients, potentially providing future therapeutic opportunities for high-risk MB patients targeting ecDNA-specific properties, such as its generation, inheritance, micronucleation, or aggregation.

4. The findings on enhancer rewiring are nice and are in line with previous observations by Morton et al and Helmsauer et al. The CRISPRi screen is very elegant and the results support the authors conclusions. Helmsauer et al (Nature Communications 2020) analyzed the distribution of amplicon boundaries in MB subgroups (see supplementary Fig. 3 in their paper) and found enhancers to be included on these amplicons. This paper did not have information about the status (ecDNA vs. linear) of these amplicons in medulloblastomas. The authors of the current manuscript may want to compare ecDNA+ vs. linear amplification for the presence of these enhancers to see if they can confirm these previously published results. Are the enhancers that the authors found to be required for oncogene expression using CRISPRi the same as those observed to be included in medulloblastoma amplicons?

We thank the reviewer for raising this interesting question. To determine whether there is selective pressure for including regulatory elements near the *MYCN* locus on ecDNA in MB, we have now compared the relative frequency of ecDNA vs. non-ecDNA amplifications at the extended *MYCN* locus. In this patient cohort, we identified 19 circular ecDNA and 4 non-ecDNA amplifications of *MYCN*, and compared the genomic co-ordinates of amplicons in these two classes (**Response**

Fig. 2a below). We observed that the distribution of non-ecDNA amplification appeared to be shifted downstream of *MYCN* relative to the locations of ecDNA amplifications. However, the majority of these non-ecDNA *MYCN* amplifications overlapped a gap in the hg19 assembly downstream of *MYCN*, and corresponded to WGS sequencing data aligned to hg19. Thus, although we observe asymmetric amplification of the regulatory regions surrounding *MYCN*, we have reason to believe that our observation may reflect a technical limitation of our approach rather than a true biological difference.

To evaluate whether the enhancers of *MYCN* described by Helmsauer *et al.* (*Nature Communications*, 2020) in neuroblastoma are also active in medulloblastoma, we reexamined our ATAC-seq data of ecDNA amplifications of the *MYCN* locus. (**Response Fig. 2b** below). Our small sample of n=2 precludes any generalization to the MB subgroups as observed by Helmsauer *et al.* Nevertheless, we can observe that enhancers e2 and e4 were accessible in the ATAC-seq data of the SHH MB tumor MB275, and enhancers e1, e2, and e4 were accessible in the ATAC-seq data of the Group 4 MB tumor MB281. Overall, our ATAC-seq data at the *MYCN* amplification are concordant with the H3K27ac profiles of MB subgroups shown by Helmsauer *et al.*

At the *MYC* locus, the known superenhancer region 5' upstream of *MYC* (Lin *et al.*, *Nature* 2016) is enriched on *MYC* amplifications in MB, both ecDNA and non-ecDNA (**Response Fig. 2c** below). We find that accessible regions within this superenhancer are co-amplified on ecDNA, affect cell proliferation, and drive *MYC* expression in D458. Co-amplification of *MYC* and this *MYC* enhancer has previously been described in lung cancer (Hnisz *et al.*, *Cell* 2013), chronic lymphocytic leukemia (Edelmann *et al.*, *Blood* 2012), and a B-cell lymphoma cell line (Ryan *et al.*, *Cancer Discovery* 2015).

In summary, our data indicate that co-selection of enhancers with oncogenes on focal amplifications may occur in MB regardless of the topology of the amplification. While interesting, we believe that this analysis faces some technical challenges and should be extended across other frequently co-amplified oncogenes and enhancers, and would best be addressed in subsequent studies.

Response Figure 2: Distribution of ecDNA amplification at the *MYCN* and *MYC* locus in MB.

(a) Histograms of ecDNA-amplified ($n = 19$) and non-ecDNA-amplified ($n = 4$) regions of the genome in our patient cohort. Pictured is a 4Mbp window centered on MB oncogene *MYCN*. Amplifications identified as ecDNA are enriched downstream, and non-ecDNA amplifications are enriched upstream, of *MYCN*. However, this result appears to be driven by a gap in the hg19 assembly upstream of *MYCN*, pictured, which may disrupt algorithmic detection of circular ecDNA.

(b) H3K27ac and chromatin accessibility profiles of *MYCN*-amplified MB. Anecdotally, ATAC-seq of MB ecDNA at the *MYCN* locus appears to covary with the H3K27ac profiles reported by Helmsauer *et al.*, but we do not have sufficient sample to perform a meaningful statistical comparison.

(c) Amplifications of *MYC*, both ecDNA and non-ecDNA, are enriched to include an upstream superenhancer of *MYC*, which

we find is co-amplified and functional on the D458 ecDNA. The full interactive figure spanning the full human genome may be found at <https://tinyurl.com/2eozxte3>.

5. Even though the analysis of ecDNA's structural evolution is interesting, it is quite preliminary. Shoshani et al (Nature 2021) and Rosswog et al (NatGen 2021) amongst others have nicely shown how chromothripsis can contribute to ecDNA evolution. The authors here show three examples for ecDNA evolution, all of which show different changes. Due to the limited number of samples the authors cannot derive any biological conclusions from their observations. I would suggest either expanding the cohort of serial biopsied samples or omitting/shortening this part from this manuscript.

We agree with the reviewer on the limitations imposed by the low number of MB patient-matched primary/ relapse sample pairs available. As the reviewer suggests, we have removed the paragraph on ecDNA structural evolution from this manuscript.

6. The authors nicely reproduce the previously reported association of chromothripsis, ecDNA with TP53 loss in SHH MBs (Rausch et al., Cell 2012), but mention that no association of ecDNA with TP53 loss was observed in other risk groups. Are there any other molecular factors associated with ecDNA presence in the other risk groups? One might expect to find other means of TP53 inactivation such as MDM2 amplification etc.

TP53 loss is observed recurrently in SHH and WNT tumors, but we did not observe any association between TP53 loss and ecDNA in WNT subgroup medulloblastoma. TP53 is rarely altered in Group 3 or Group 4, and no association was found with ecDNA in these subgroups. It remains unclear to us why the association between TP53 and ecDNA is specific to the SHH subgroup.

We do observe p53 inactivation by means other than TP53 mutation in SHH MB, including by MDM4, PPM1D, CDK6 and MDM2 amplification. These are depicted in **Figure 1b**. In response to this and other reviewer comments, we have revised the text as shown here to clarify this observation (ll. 134-142):

To evaluate whether there is a TP53-independent effect of ecDNA on survival, we performed a Cox regression including TP53 alteration as a covariate and controlling for collinearity. The effect of ecDNA on survival remains significant but diminished when we include p53 alteration as a covariate in our Cox models ($HR_{PFS} = 1.87$, $p = 0.01$; $HR_{OS} = 2.32$, $p < 0.005$; **Supplementary Fig. 4**), indicating that there is an effect of ecDNA on survival that cannot be explained by p53 mutation alone. Such an effect may be explainable by a p53-independent mechanism of ecDNA formation, or by inactivation of the p53 pathway by other means such as CDKN2A deletion or

PPM1D, *CDK6*, *MDM4* or *MDM2* amplification⁴⁸. In our patient cohort, we observe 9 such amplifications on ecDNA across all subgroups (**Fig. 1b**, **Supplementary Table 1**). Although causality cannot be inferred from these data alone, our survival analyses identify *TP53* alteration and ecDNA as clinically relevant biomarkers for a subset of highly aggressive SHH MB tumors.

We have now performed association testing of SNVs and small indels in all protein coding genes genome wide with respect to ecDNA status across the entire cohort and in the four individual MB subgroups. Besides the *TP53* mutation in SHH tumors, no other genes had significant associations with ecDNA after multiple hypothesis correction.

7. The use of optical mapping with NGS is elegant, but it remains unclear why the authors did not use other means of validating the presence and structure of ecDNAs in their cohort, eg. FISH, CRISPR-CATCH or exonuclease-based enrichment protocols.

We would like to thank the reviewer for acknowledging our use of optical mapping with NGS to validate the presence of ecDNA amplifications. We agree that additional validation of ecDNA using complementary methods will further strengthen our results. Therefore, we have now performed FISH on archival FFPE tissue for additional ecDNA+ samples that were available, bringing the total number of ecDNA+ samples validated by FISH to 5: RCMB56 (**Fig. 2**), D458 (**Fig. 4d**), MB268 (**Suppl. Fig. 9**), MB036 and MB177 (**Suppl. Fig. 7**). We have also developed and applied a novel FISH microscopy image processing pipeline ecSeg-i, based on a previously published method ecSeg (Rajkumar *et al.*, 2019), for the automated quantification of ecDNA. This analysis provides further evidence for increased intratumoral heterogeneity of ecDNA copy number in human MB tumors (**Fig. 3a**).

Motivated by the reviewer's comment and taking advantage of the available RCMB56-pdx tumor, we collaborated with the laboratory of Dr. Howard Y. Chang at Stanford University to additionally validate the predicted circular ecDNA amp1 using their recently developed CRISPR-CATCH method. As expected, cutting this ecDNA in HMW DNA isolated from the RCMB56-pdx tumor produced a single fraction of DNA with the predicted length of ecDNA amp1 and standard short read sequencing confirmed that this DNA fraction maps to the sequence of the ecDNA amp1 (**Fig. 2g-h**).

8. The single cell results are fascinating, but unfortunately remain rather preliminary. The authors only analyzed one tumor, raising the question of how generalizable their findings are. Relatively few analyses are done to look at the structural heterogeneity in single cells, which is something the authors had observed in bulk sequencing. As with the WGS data, the authors did not validate the presence and heterogeneity of ecDNA in their sample using orthogonal methods, eg. FISH or others.

We have now performed FISH microscopy of interphase cells in archival FFPE tissue from 4

primary MB tumors, including RCMB56-ht, the tumor we analyze by single-cell sequencing. To estimate copy number heterogeneity, we established an automated image analysis pipeline which segments nuclei using NuSeg and estimates FISH spots by gaussian kernel estimation. These data allow us to validate our estimate of heterogeneity from RCMB56-ht single cell sequencing, and to report more general observations in several patient tumors.

Our estimates of copy number heterogeneity from RCMB56-ht FISH imaging were consistent with our estimate from single-cell data, where both modalities indicated that the mean and variance of copy number was greater for ecDNA amplifications RCMB56 amp1 and RCMB56 amp2 than the control non-ecDNA amplifications COLO320HSR and RCMB56 gain1 (**Fig. 4b,d**). Furthermore, the estimated distributions of ecDNA copy number in cells of other primary tumors (MB036, MB177, MB268) are similar to RCMB56-ht extrachromosomal amplifications, but distinct from the estimates for COLO320HSR and RCMB56 gain1, suggesting that the high mean and variance of copy number we observe in RCMB56-ht is specific to extrachromosomal DNA relative to other forms of copy gain.

Minor comments:

1. The title strongly focuses on intratumoral heterogeneity, but the results presented here are mainly based on bulk sequencing. There is also no direct evidence presented here that would support that ecDNA “promotes tumorigenicity”. I would suggest refining the title to better reflect the manuscript’s main conclusions.

We agree with the reviewer and have removed any references to tumorigenicity from the title. With the additional results on imaging ecDNA in single cells in a series of MB tumors, we feel that we have examined intratumoral heterogeneity of ecDNA in MB sufficiently to refer to it in the revised title:

Circular extrachromosomal DNA promotes inter- and intratumoral heterogeneity in high- risk medulloblastoma

2. The abbreviation ecDNA is usually used to refer to extrachromosomal DNA and not extrachromosomal circular DNA, which is usually abbreviated eccDNA. I would suggest changing this as it risks to cause some confusion amongst readers.

We thank the reviewer for bringing this potential confusion to our attention. We agree that consensus on terminology in this field is lacking, and we wish to adhere to the still-emerging consensus. The term “eccDNA” has been used to refer to small circular sequences less than

10kbp, containing no genes and present in cancerous and healthy cells alike. Our study specifically examines the large circular extrachromosomal DNA sequences containing oncogenes and found exclusively in cancer, which have been widely discussed under the monikers “double minutes” (dm) or “circular ecDNA”. We have amended our language to clarify our focus on circular ecDNA.

3. In Fig 1b there is a row that indicates “p53 inhibitors”. It is unclear what this refers to.

As noted above in M.P. 1.6, we observe that the p53 pathway may be disrupted by means other than mutation of the *TP53* gene, namely, ecDNA amplification of the p53 inhibitor genes *PPM1D* (4), *CDK6* (3), *MDM2* (1) and *MDM4* (1). We have updated the text to clarify this relationship (**Figure 1** legend):

(b) A subset of recurrently ($n \geq 2$) amplified genes on ecDNAs in this patient cohort. p53 inhibitors: negative regulators of p53 pathway activity; COSMIC: genes listed as tier 1 or 2 of the COSMIC Cancer Gene Census³.

4. Chang and Verhaak and others proposed the ecDNA could interact with the linear genome, and termed these ecDNA “mobile enhancers”. Did the authors observe any contacts of ecDNA with the linear chromosomes in trans in their HiC data? I understand that this may be outside the scope of this manuscript, but it would be of great interest to the field.

Although outside the focus of this manuscript, we share the interest of Reviewers 1 and 3 in the possible functional roles of intermolecular chromatin interactions with ecDNA. To address this interest, we performed interchromosomal chromatin contact mapping of the confirmed circular extrachromosomal amplicon RCMB56-pdx amp1, using FitHiC2 and following best practices from the developer. This computational analysis points to a number of chromatin interactions between a putative enhancer locus on the ecDNA and various genes including the known oncogenes *MECOM* and *IGF1R*. Although we did not perform functional validation of these interactions, we felt that the presence of clear chromatin interactions between ecDNA and chromosomal oncogenes merited a short mention in the Discussion (ll. 305-311):

Recent studies have revealed intermolecular enhancer-promoter interactions between ecDNA molecules (‘ecDNA hubs’)⁴² or between the chromosomes and ecDNA (‘mobile enhancers’)⁶⁵. To test for such intermolecular chromatin interactions in medulloblastoma, we computationally identified interchromosomal loops from Hi-C of the SHH MB tumor RCMB56-pdx, where one loop anchor mapped to the circular ecDNA amp1. This analysis revealed a nexus of interactions mapping from the *ARHGAP29* locus on amp1 to loci elsewhere in the genome with plausible tumorigenic roles, including *MECOM*, *RAD51AP2*, *POU4F1*, and *IGF1R*

(Supplementary Fig. 14). However, the functional significance of these intermolecular chromatin interactions in MB remains untested.

5. The description of ecDNA in patient derived models is placed in the same figure describing single cell sequencing data, which is quite confusing to the reader and not easy to follow.

We thank the reviewer for the suggestion. Based on this comment and feedback from another reviewer (see also M.P. 2.9), we have removed the analysis of ecDNA across patient-derived models from the manuscript.

Reviewer #2

Remarks to the Author:

The manuscript 'EcDNA promotes intratumoral heterogeneity and tumorigenicity in medulloblastoma' by Chapman et al, focuses on the detection of extrachromosomal circular DNA (or double minutes) across a large number of human medulloblastomas and a smaller number of medulloblastoma cell lines and PDXs, using predominantly previously published datasets. While the overall theme of ecDNA is interesting, I am concerned that the analyses presented in this manuscript superficially span several different areas/themes (for example spanning analyses in primary tumors, cell lines and then focusing on evolution of PDXs) rather than going 'deep' into answering any one specific question. As such, the findings overall don't significantly improve our understanding of the pathogenesis of medulloblastoma and, in my opinion, the analyses unfortunately do not reveal insights about the mechanisms through which ecDNA are generated and contribute to tumorigenesis.

Below are some comments and suggestions that I hope the authors will find helpful.

We thank the reviewer for the critical evaluation of our study. We found that many of these comments focused on understanding the roles of circular ecDNA in medulloblastoma etiology and treatment. Therefore, we have excluded some of the points the reviewer found least supportive (see M.P.s 2.4, 2.6, 2.9) and strengthened each of this manuscript's core themes with additional analyses. Among other additions, this revision presents further statistical analyses clarifying the relationships between ecDNA and other clinically relevant biomarkers (M.P.s 2.1, 2.2, 2.3, and 2.5), and additional CRISPRi-based experiments showing the functional import of co-amplified enhancers (M.P.s 2.7, 2.8).

1. Line 81. The authors claim that the absence of ecDNA may account for their good prognosis. The observation that there is low ecDNA in WNT is an association with the WNT subgroup and doesn't necessarily mean that the presence of ecDNA has prognostic significance. If the authors do not prove

causality between presence of ecDNA and poor prognosis.

We agree with the reviewer that this was an overstatement not justified by the available data, and have updated the text accordingly (ll. 283-289):

As in other cancers^{11,12,63}, ecDNA frequently amplifies known oncogenic MB driver genes. Our results identify ecDNA as a frequent feature of *MYC*-amplified Group 3 and p53-mutant SHH tumors, which share exceptionally poor prognoses^{27,28} but few other recurrent driver mutations. Recent longitudinal analysis of Barrett's esophagus suggests that *TP53* alteration is an early event in ecDNA-driven malignant transformation⁶³. However, the absence of detectable ecDNA in *TP53*-mutant WNT subgroup tumors, and the frequent occurrence of ecDNA in Group 3 tumors with wild type *TP53*, suggest that the mechanisms for the generation and selection of tumor cells with ecDNA are MB subgroup-specific and thus may be modulated by the different cellular contexts of MB progenitor cells.

2. Similarly, the cox proportional hazards model described in line 88 only take into account sex, age and molecular subgroup and find an association with ecDNA and prognosis. However, other important variables that may track with ecDNA (and thus may confound the analysis) are not taken into account. In particular, TP53 mutations come to mind as these are found more commonly in SHH subgroups (which the authors report have higher amounts of ecDNA) – and TP53 mutations are associated with a dismal prognosis. Multivariate analyses should take into account known variables that have prognostic significance in medulloblastoma. This is particularly important as the authors later show that TP53 mutations are also associated with ecDNA in SHH medulloblastomas.

We would like to thank the reviewer for this comment as the revised study, through our additional analyses, more clearly illustrates the relationship of ecDNA and other genomic alterations common in MB.

Most regression models, including Cox Proportional Hazards (CPH) analysis, assume independence between model parameters, and cannot reliably estimate hazards of highly correlated variables. In our dataset, *TP53* mutation status is (nearly) a linear combination of ecDNA status and molecular subgroup, because WNT tumors with *TP53* mutation universally do

not have ecDNA and SHH tumors with *TP53* mutation (nearly) universally have ecDNA. To address this issue comprehensively, we have consulted with a biostatistician at the Moores Cancer Center of UCSD and included 2 new analyses in our revision. Firstly, we perform accelerated failure time (AFT) regressions describing a classical mediation model in which part of the effect of *TP53* mutation on survival is indirect via ecDNA (**Suppl. Fig. 3**). Consistently with this proposed mechanism, we find that much of the total effect of *TP53* mutation on survival

may be explained by an effect of ecDNA on survival and frequent co-occurrence of *TP53* mutation and ecDNA in the same tumor, Secondly, we now report additional CPH models of survival which include *TP53* mutation as a covariate, controlling for model instability due to collinearity using ridge regression (**Suppl. Fig. 4**). The effect of ecDNA on survival remains significant, suggesting that there is a contribution of ecDNA to patient outcome that is independent of *TP53* mutation. These analyses offer a reasonable explanation for the associations we observe between *TP53*, ecDNA and patient outcomes.

3. Line 83. The authors highlight several genes that are involved in ecDNA, many of which have been shown to be amplified in medulloblastomas. What proportion of tumors do these amplicons involve ecDNA vs amplification. Are there any associations with other driver genes that may determine suggest different mechanisms through which these genes are amplified vs incorporated into ecDNA? Are there any other biological differences between these groups? Are there regions in the genome that are more likely to be involved in ecDNA more frequently than one would expect by chance?

Similarly to Kim *et al.* (*Nature Genetics*, 2020, Figure 2a), we do observe that some loci are more likely to be involved in ecDNA than one would expect by chance. At the reviewer's suggestion, we examined the genomic distribution of ecDNA amplifications in MB to that of linear, highly rearranged, and breakage-fusion-bridge (BFB) amplifications. Comparing the fraction of ecDNA vs. non-ecDNA amplifications, we found that ecDNA accounts for most of the total amplifications of many of the most frequently amplified genes in MB: *MYCN* (19/23), *MYC* (11/18), *MYCL1* (3/3), *CCND2* (5/5), *GLI2* (4/4), and *TERT* (3/3). This observation is concordant with the known genomic distribution of amplifications of any kind in MB (Northcott *et al.*, *Nature* 2012) and establishes ecDNA as a very common means of oncogenic amplification in MB.

We have now also performed association testing of SNVs and small indels in all protein coding genes genome wide with respect to ecDNA status across the entire cohort and in the four individual MB subgroups. Besides the *TP53* mutation in SHH tumors, no other genes had significant associations with ecDNA after multiple hypothesis correction.

4. Line 92 on: the authors describe three primary -relapsed tumors and in each case, characterize the differences in ecDNA between the primary and recurrent tumor. However, this analysis is performed in isolation of other genetic features (somatic nucleotide variants, copy-number, other structural variants etc) and as such it is difficult to ascertain the significance in changes in ecDNA as being driver vs passenger events. Moreover, based on this limited analysis the authors conclude that sequence rearrangement or de novo formation of ecDNA may be common in medulloblastoma. This is a very strong conclusion based on limited and descriptive analyses of three tumors.

We agree with the reviewer on the limitations imposed by the low number of patient-matched

primary/ relapse sample pairs available. As also suggested by reviewer 1 (M.P. 1.5), we have now removed some weaker and more tangential points, including this paragraph on structural evolution of ecDNA, and strengthened several findings with additional experiments and analyses. As a result, the overall revised study better focuses on the key issues of survival analysis in the context of ecDNA, intra- and inter-tumor heterogeneity of ecDNA copy number, and regulatory DNA co-amplified on ecDNA.

5. Line 107. The authors show an association between TP53 mutations and ecDNA. Did the authors perform a broader multivariate analysis to identify other molecular features that are associated with ecDNA in medulloblastoma?

We have performed association testing of somatic SNVs and small indels in all protein coding genes with respect to ecDNA status across the entire cohort and in the four individual MB subgroups. Besides the *TP53* mutation in SHH tumors, no other genes had significant associations with ecDNA after multiple hypothesis correction.

6. Lines 117 on. The authors perform 'optical mapping' of ecDNA in two medulloblastoma cell lines generated from the same patient (primary and metastasis) many decades ago, and four PDX models generated more recently. The authors first show genetic evolution of D425 and D458 based on ecDNA analyses. Technically this is interesting, but biologically not surprising, given how old these models are and the many years that they have been in culture. Description of the analyses of ecDNA in the PDXs is very limited and descriptive and it is hard to determine what conclusion(s) the authors derive from this analysis.

We agree with the reviewer and have omitted comparison of D425 and D458 from the revised manuscript.

7. The authors perform analyses to show that DNA breakpoints connect gene and enhancer loci co-amplified in ecDNA in at least half of ecDNA positive tumors. This is not surprising as structural variants in general have been well documented to commonly involve gene regulatory elements. If the authors were to compare the frequency of these interactions in non-ecDNA structural variants vs ecDNA, is there an enrichment in these interactions in ecDNA?

We agree with the reviewer that structural variants frequently lead to enhancer hijacking in tumors. For MB, two cases of enhancer hijacking due to chromosomal SVs were described to date: that of *GFI* family oncogenes (Northcott *et al.*, *Nature* 2014) and of *PRDM6* (Northcott *et al.*, *Nature* 2017). We do not have any reasons to believe that enhancer rewiring is more frequent across breakpoints on linear or on ecDNA. Systematic analysis of enhancer rewiring

in linear vs ecDNA amplification is challenging due to limitations of available tools to accurately reconstruct re-arranged tumor genomes and DNA loops that span breakpoints, within and across different chromosomes. However, our focused analysis of enhancer rewiring on ecDNA+ cases has revealed frequent enhancer rewiring on ecDNA that were not previously described. Those include

cases of MB cell lines and tumors where several pieces of DNA from different chromosomes are co-amplified on the same ecDNA that show DNA loops in the Hi-C data (**Fig. 4f, Suppl. Fig. 10**).

8. Line 171 onwards. The authors leverage CRISPRi to show that the enhancers associated with ecDNA in the D458 cell line exert proliferative advantage. While this is an important experiment, in its current form, it doesn't show specificity of the ecDNA enhancers as being essential for cell proliferation. For example, what if the guides targeted other enhancers on D458 that were not part of the ecDNA? These enhancers are not necessary for D283 however the structural variant landscape of this cell line is different to D458. Even in the absence of ecDNA of D283, I suspect that if the authors had designed a library to target enhancers involved in other structural variants present in D283 that they would see a similar phenotype. As such, I am concerned that this experiment shows that enhancers are necessary in D458 rather than enhancers specifically involved in ecDNAs present in D458.

We agree with the reviewer that targeting any enhancer essential for the D458 cell line will decrease proliferation, regardless of whether it's located on ecDNA or on the linear chromosome. To control for ecDNA-specificity, we relied on the experimental design of Morton *et al.* (*Cell* 2019) who used an ecDNA- glioblastoma cell line as a control for their CRISPRi screen in a ecDNA+ glioblastoma cell line. While Morton *et al.* focused on the *EGFR* locus that is frequently amplified in glioblastoma, we designed a sgRNA library with 32,530 guides targeted against 645 accessible loci amplified together with the *MYC* oncogene on ecDNA in the Group 3 MB cell line D458. These sgRNA sequences have no homology elsewhere in the human genome, as determined by the CRISPR guide design software CHOPCHOP. To control for ecDNA-specificity in the best possible way, we selected another Group 3 cell line with chromosomal (HSR) *MYC* amplification, D283, as a control line for D458. By mapping enhancers across 28 medulloblastoma tumors and three MB cell lines, we had previously shown that the enhancer landscapes are highly conserved within the MB subgroups WNT, SHH, Group 3, and Group 4, and divergent across MB subgroups (Lin *et al.*, *Nature* 2016). We have performed the same CRISPRi screen, using the same guide library, in the two Group 3 MB cell lines D458 and D283. Given that both cell lines are derived from Group 3 MB tumors and both have *MYC* amplifications, we conclude that any significant differences of enhancer essentiality is a result of the different architecture of the *MYC* amplifications in these two cell lines (*MYC* ecDNA in D458 vs. *MYC* HSR in D283). In response to reviewer 3, we have now performed

additional CRISPRi validation experiments in D458 and D283 targeted against selected regulatory regions followed by transcriptional analysis of *MYC*, *PVT1*, and *OTX2*. As expected, we find significant changes in transcription upon CRISPR inhibition in D458, consistent with the D458-specific effects on proliferation (see also M.P. 3.1).

9. Line 230 onwards. This section compares ecDNA in a PDX to a human tumor. This section doesn't really add anything to the role of ecDNA in generating medulloblastomas and shows fidelity of one PDX to its primary tumor. Similarly, the final section of the results show that ecDNA is more enriched in PDXs than primary tumors. Again, this doesn't provide insights into the role of ecDNA in tumor formation. If the authors want to examine the relevance of PDXs to human tumors, I suggest that this is presented in a separate manuscript.

We concur that the analysis of ecDNA in MB PDX models may be better addressed in a separate study (see also M.P. 1.1). Therefore, we have removed this part from the revised manuscript.

Reviewer #3

Remarks to the Author:

The manuscript entitled "EcDNA promotes intratumoral heterogeneity and tumorigenicity in medulloblastoma" by Dr. Chavez and colleagues presents a comprehensive analysis on the landscape of ecDNA based on the analysis 481 medulloblastoma whole-genome sequencing data. The authors discovered a prevalence of 18% ecDNA+ cases in their cohort and found ecDNA+ status is significantly associated with relapse and mortality of medulloblastoma, the most common pediatric brain tumor. The complex structure of ecDNA was delineated by computational modeling (AmpliconArchitect) and integration of long-read optical mapping. Based on the ecDNA structure defined by these approaches, the authors explored the significance of co-amplified enhancers by performing CRISPRi screening on open chromatin regions in ecDNA and discovered 6 enhancers enriched in the ecDNA tumor sample. Finally, they examined the heterogeneity of ecDNA in primary tumor and PDX models and analyzed ecDNA evolution from patient sample to PDX model by performing dual single cell (sc) assay of scRNA-seq and scATAC-seq and inferring ecDNA status from scATAC-seq.

The study is very interesting in highlighting the role of ecDNA in the pathogenesis of medulloblastoma. Furthermore, the high-quality data presented in this study, in particular optical mapping, unveiled new architecture of ecDNA which is lacking in current literature. Given that ecDNA research is a relatively new field, further analysis on the extensive data sets generated from this study can potential help elucidating some of the recently reported features of ecDNA, which can broaden the impact of the current work.

Specific comments are listed below.

We would like to thank the reviewer for acknowledging the high quality of the data included in this study and the importance of ecDNA in the pathogenesis of medulloblastoma.

Major comments:

1) In the section “Co-amplified enhancers shape the transcriptional regulatory circuitry of ecDNA in D458 cells”, the authors presented CRISPRi screen of 32,530 guides targeting all 645 accessible chromatin regions on the D458 ecDNA. The same CRISPRi screen was performed on D283 cell line as a control because D283 has MYC HSR but no amplification of OTX2. The authors reported 6 D458-specific functional elements which were highlighted in Figure 2h. There are several points that will require clarification: a) Is MYC or OTX2 suppressed in the CRISPRi cells derived from D458 but not in those of D283?; b) do these loci interact with MYC/OTX2 promoter specifically in D458 based on Hi-C data? c) Was OTX2 also over-expressed in D283? If not, what is the molecular basis for reduced growth of D283 cells when the yellow highlighted site in OTX2 (last exon) was affected by CRISPRi? d) Does the green site at PVT1 promoter affect the expression of both cMYC and PVT1? Without addressing these questions, the mechanism leading to reduced proliferation of D458 remains unclear. These questions can be

addressed using existing data or performing assays on cells harvested after 21 days in D458 are CRISPRi inhibition.

We agree with the reviewer that gene expression analysis of MYC, PVT1, and OTX2 will provide additional important information on the transcriptional dependencies of enhancers found to be essential for proliferation of D458 and D283, respectively. Due to the nature of the pooled dropout CRISPRi screen used for this experiment, cells harvested after 21 days are depleted of guides targeted against enhancers essential for proliferation. Thus, cells harvested after 21 days cannot be used for analyzing transcriptional dependencies on the essential enhancers. Motivated by the reviewer, we have now performed additional CRISPRi experiments targeted against two of the accessible regions that were found by the screen to be essential for proliferation in D458 but not in the control line D283: the *MYC* superenhancer at chr8:127,330,000 and the *PVT1* promoter at chr8:127,795,000. Consistently with the results of the pooled proliferation screen, we find that inhibition of the *MYC* enhancer reduces *MYC* expression, and inhibition of the *PVT1* promoter reduces *PVT1* expression, specifically in D458.

(a) Is MYC or OTX2 suppressed in the CRISPRi cells derived from D458 but not in those of D283?

Consistent with the result of the CRISPRi proliferation screen, silencing of the *MYC* superenhancer reduced *MYC* expression for 2 out of 3 tested sgRNAs in D458 but not in D283 (new **Suppl. Fig. 12a-b**, reproduced in **Response Fig. 3a-b**). No significant difference was observed in *OTX2* transcription in either cell line (new **Suppl. Fig. 12c-d**, reproduced in

Response Fig. 3c-d). Silencing of the PVT1 promoter abrogated PVT1 transcription but not MYC or OTX2, in D458 but not in D283 (new **Suppl. Fig. 13**, reproduced in **Response Fig. 4**). Thus, although proliferation in both Group 3 MB cell lines is driven by MYC amplification, the relative importance of co-amplified genes and cis-regulatory elements is specific to the genomic architecture of the amplicon.

(b) Do these loci interact with MYC/OTX2 promoter specifically in D458 based on Hi-C data?

Of the 6 functional elements which, upon CRISPRi inhibition, specifically reduced D458 proliferation compared to D283, the *PVT1* promoter and both *MYC* enhancers show significant DNA interactions with *MYC* in the D458 Hi-C data. By visual inspection of the D458 Hi-C data, the *OTX2* enhancer and promoter appear to interact with *MYC* in *trans* (new **Suppl. Fig. 11b**, reproduced in **Response Fig. 5a**). However, due to current technological limitations in automated calling of inter-chromosomal interactions near breakpoints of structural variants that involve different chromosomes (such as chromosomes 8 and 14 fused together on the D458 ecDNA amplification), we were unable to use existing software tools (HiCCUPS, FitHiC2, NeoLoopFinder) to reproducibly identify high-quality interchromosomal loops. In the revised manuscript, we include figure panels showing interaction maps of ecDNAs that suggest inter-chromosomal interactions between MYC on chr8 and several regulatory regions on chromosome 14 co-amplified on ecDNA (RCMB56 amp2, new **Suppl. Fig. 10b**; D458, new **Suppl. Fig. 11b**) and stress in the Methods and the figure legends that these loops were manually annotated (ll. 512-515):

Ectopic chromatin interactions spanning breakpoints on the D458 ecDNA and RCMB56 amp2, including interchromosomal interactions, could not be accurately called by software tools known to us due to technical limitations in this emerging field. These interactions were manually annotated from the interaction matrices shown in **Supplementary Figures 10b, 11b**.

(c) Was OTX2 also over-expressed in D283? If not, what the molecular basis for reduced growth of D283 cells when the yellow highlighted site in OTX2 (last exon) was affected by CRISPRi?

OTX2 is a known marker for Group 3 medulloblastoma and is highly expressed in both D458 and D283 (new **Fig. 4e** in the revised manuscript and **Response Fig. 5b**). Thus, the molecular basis for reduced growth upon CRISPR inhibition of the last *OTX2* exon in D283 may be explained by D283-specific transcriptional regulation of *OTX2*.

(d) Does the green site at PVT1 promoter affect the expression of both cMYC and PVT1?

Motivated by the reviewer, we have performed an additional CRISPR inhibition experiment targeted against the PVT1 promoter with subsequent transcriptional analysis of MYC, PVT1

and OTX2 (see also above in a). In line with the results of the CRISPRi proliferation screen silencing of the PVT1 promoter abrogated PVT1 transcription, but not MYC or OTX2, in D458 but not in D283 (new **Suppl. Fig. 13**, reproduced in **Response Fig. 4** below).

Response figure 3: A co-amplified enhancer on the D458 ecDNA promotes MYC expression. Relative expression of MYC (a-b), and OTX2 (c-d), measured by qPCR ($2^{-\Delta\Delta Ct}$), in D458 and D283 cell lines upon CRISPRi targeting of an accessible locus within a known MYC superenhancer. The targeted region promotes proliferation in D458 but not D283 (see also **Fig. 4h**). sgNT: nontargeting control; sgMYC-SE-A-C: sgRNAs targeting the MYC enhancer at D458_peak_30782, positions chr8:127330655, chr8:127330840, and chr8:127330927 respectively. qPCR was performed on all guides in technical triplicate. All co-ordinates refer to the hg38 assembly. ** $p < 0.01$; *** $p < 0.001$; nested ANOVA with Dunnett's correction.

Response Figure 4: A co-amplified enhancer on the D458 ecDNA promotes MYC expression.

Relative expression of *PVT1* (a-b), *MYC* (c-d), and *OTX2* (e-f) measured by qPCR ($2^{-\Delta\Delta Ct}$), in D458 and D283 cell lines upon CRISPRi targeting of the *PVT1* promoter which promotes D458 proliferation (see also Fig. 4h). sgNT: nontargeting control;

sg*PVT1*-D-F: sgRNAs targeting the *PVT1* promoter at D458_peak_30920, positions chr8:127794266, chr8:127794773, and chr8:127794945 respectively. qPCR was performed on all guides in triplicate. All co-ordinates refer to the hg38 assembly. *** $p < 0.001$; **** $p < 0.0001$; nested ANOVA with Dunnett's correction.

Response Figure 5. (a) Hi-C interchromosomal interaction map of D458 at the

amplified segments of chr8 and chr14. Notable loci of interest are indicated by arrows. f: detected as functional in D458 in our CRISPRi screen illustrated in Figure 4g; nf: not detected as functional. **(b)** Transcription of *MYC* and *OTX2* in Group 3 medulloblastomas. *OTX2* is highly transcribed in Group 3 tumors, including a subset for which *MYC* is not activated (Archer *et al.*, *Cancer Cell* 2018).

2) scATAC-seq read-depth was used to estimate the ecDNA+ cells. Additional verification (e.g. FISH) is needed to validate the estimated prevalence of ecDNA+ cells patient tumor or PDX of RCMB56. Are there any somatic mutations outside the ecDNA amplicon that co-segregate with the two ecDNAs identified in this case?

As suggested, we performed FISH to estimate heterogeneity of ecDNA+ cells by an orthogonal method (see also M.P. 1.8).

While we have not identified somatic SNV or indel mutations that co-segregate with each ecDNA lineage, there are additional somatic mutations outside the ecDNA amplicon that co-segregate with the ecDNA+ cell cluster, notably chr3p loss and chr3q gain (**Suppl. Fig 8f**).

3) Koche *et al* (Nature Genetics, 2020) reported that ecDNA re-integration is common in neuroblastoma. Is this phenomenon (ecDNA re-integration) also found in medulloblastoma? Duplication of genomic fragments can occur on ecDNA itself as shown in Figure 2f. How does a pattern of within-episome duplication differ from ecDNA re-integration?

Signatures of duplications inside an ecDNA episome differ significantly from the signatures of HSR structures themselves, which are composed of multiple appearances of full ecDNA elements. To illustrate, we refer to the Song *et al.* study (*Cancer Discovery* 2022), which presented a candidate structure for an ecDNA-derived HSR using similar methodology to the reconstruction of ecDNA in this study. The candidate HSR structure presented in Song *et al.* (Figure 4H in Song *et al.*, *Cancer Discovery* 2022) suggests that HSR formation can involve multiple different breakpoints to connect complete (or nearly complete) copies of ecDNA together, some breakpoints being conserved between repeat units, and some which are not. This occurs in stark contrast to the simpler duplication events we present in **Figure 4f**, which are still concordant with an uninterrupted ecDNA cycle, but themselves do not explain chaining of entire ecDNA copies into HSRs. It is also possible that a small fraction of cells do harbor HSR forms of the complete ecDNA, but overall we do not find evidence of sequence re-integration in our optical mapping data. We also do not observe evidence of sequence re-integration among the MB cells for which FISH imaging was performed (**Fig. 2c-d, Suppl. Fig. 7**). While ecDNA re-integration may plausibly occur in medulloblastoma, we do not feel we have sufficient evidence to comment on the question in this manuscript.

4) Given that Hi-C was performed for several samples, are there any data supporting the role of ecDNA as a mobile enhancer as described by Zhu et al in Cancer Cell. 2021 May 10;39(5):694- 707.?

We would like to thank the reviewer for raising this question as we share the interest in the possible functional roles of intermolecular chromatin interactions with ecDNA. To address this question, we performed interchromosomal chromatin contact mapping of the confirmed circular extrachromosomal amplicon RCMB56-pdx amp1, using FitHiC2 and following best practices from the developer (personal communication). This computational analysis points to a number of chromatin interactions between a putative enhancer locus on the ecDNA and various genes including the known oncogenes *MECOM* and *IGF1R*. However, because we do not have data on the functionality of these interactions, we only briefly describe these observations in the Discussion of the revised manuscript and point out that the functional significance of intermolecular chromatin interactions in MB should be tested in future studies.

Minor comments:

1) Survival analysis on ecDNA+ versus ecDNA- patients described in line 88-89. Would the authors comment whether the significance can be achieved when considering molecular subtype? Supplementary Figure 1 presented this information, but it will be good to make an explicit statement in the main text.

We have updated the text according to the reviewer's suggestion (ll. 108-110):

Stratified by molecular subgroup, ecDNA+ patients had worse overall survival in the SHH, Group 3 and Group 4 MB subgroups ($p < 0.05$ for all subgroups; **Fig. 1d-f, Supplementary Fig. 1b-d**). Survival of WNT subgroup patients was not analyzed because no WNT tumors in our patient cohort were ecDNA+.

2) Was the association of TP53 mutation with ecDNA reported in other tumor types? If so, please add a reference in the Discussion.

TP53 was not associated with ecDNA in the other tumor types. For Group 3 and Group 4 tumors, this is unsurprising because these tumors rarely present with *TP53* mutation. We might expect an association between *TP53* mutation and ecDNA in WNT tumors, given the strong association in SHH tumors, but we do not observe ecDNA in WNT tumors regardless of *TP53* mutation status. We have clarified the Results and commented in the Discussion, as suggested (ll. 283-289):

As in other cancers^{11,12,63}, ecDNA frequently amplifies known oncogenic MB driver genes. Our

results identify ecDNA as a frequent feature of *MYC*-amplified Group 3 and p53-mutant SHH tumors, which share exceptionally poor prognoses^{27,28} but few other recurrent driver mutations. Recent longitudinal analysis of Barrett's esophagus suggests that *TP53* alteration is an early event in ecDNA-driven malignant transformation⁶³. However, the absence of detectable ecDNA in *TP53*-mutant WNT subgroup tumors, and the frequent occurrence of ecDNA in Group 3 tumors with wild type *TP53*, suggest that the mechanisms for the generation and selection of tumor cells with ecDNA are MB subgroup-specific and thus may be modulated by the different cellular contexts of MB progenitor cells.

See also M.P. 1.6, reproduced below:

TP53 loss is observed recurrently in SHH and WNT tumors, but we did not observe any association between *TP53* loss and ecDNA in WNT subgroup medulloblastoma. *TP53* is rarely altered in Group 3 or Group 4, and no association was found with ecDNA in these subgroups. It remains unclear to us why the association between *TP53* and ecDNA is specific to the SHH subgroup.

We do observe p53 inactivation by means other than *TP53* mutation in SHH MB, including by *MDM4*, *PPM1D*, *CDK6* and *MDM2* amplification. These are depicted in **Figure 1b**. In response to this and other reviewer comments, we have revised the text to clarify this observation (ll. 134- 142):

To evaluate whether there is a *TP53*-independent effect of ecDNA on survival, we performed a Cox regression including *TP53* alteration as a covariate and controlling for collinearity. The effect of ecDNA on survival remains significant but diminished when we include p53 alteration as a covariate in our Cox models ($HR_{PFS} = 1.87$, $p = 0.01$; $HR_{OS} = 2.32$, $p < 0.005$; **Supplementary Fig. 4**), indicating that there is an effect of ecDNA on survival that cannot be explained by p53 mutation alone. Such an effect may be explainable by a p53-independent mechanism of ecDNA formation, or by inactivation of the p53 pathway by other means such as *CDKN2A* deletion or *PPM1D*, *CDK6*, *MDM4* or *MDM2* amplification⁴⁸. In our patient cohort, we observe 9 such amplifications on ecDNA across all subgroups (**Fig. 1b**, **Supplementary Table 1**). Although causality cannot be inferred from these data alone, our survival analyses identify *TP53* alteration and ecDNA as clinically relevant biomarkers for a subset of highly aggressive SHH MB tumors.

We have now performed association testing of SNVs and small indels in all protein coding genes genome wide with respect to ecDNA status across the entire cohort and in the four individual MB subgroups. Besides the *TP53* mutation in SHH tumors, no other genes had significant associations with ecDNA after multiple hypothesis correction.

3) Figure 2a: what is the genomic location? What does the arrow mean in this panel? Do they represent

the circularized loci described in lines 147-148?

The genomic location is chr8; the arrows indeed represent the circularized regions. We have revised the figure legend to clarify the figure (**Figure 4** legend):

Figure 4: Enhancer rewiring in medulloblastoma ecDNA affects cell proliferation. (a) ATAC-seq and Hi-C read coverage of chr8 in MYC-amplified primary tumors MB248 (top right) and MB106 (bottom left). Arrows indicate genomic regions corresponding to low-copy ecDNA amplification of *MYC* in both samples.

4) Figure 2b & f: the RNA-seq & ATAC-seq tracks in the circular view are barely visible and occupy a big ring with extensive white space. It would be good to convert the scale to a heatmap so that the footprint of gene expression/ATA-seq can be clearer and Hi-C interaction can be presented with more clarity. There is also a thin gray track in the DNA ring with no annotation which will be good to describe in figure legend. Does the gaps in the ring represent unknown sequence fragments? It will be good to clarify these in figure legend. Panel f appears to contain duplicated regions which need to be labeled or described in figure legend.

We thank the reviewer for these suggestions and have improved the visualizations accordingly. Upon comparison, we have preferred histogram over heatmap representation of sequence tracks, but have added thickness to the lines to make them more visible. Both circles are completely assembled; gaps in the circle indicate adjacent sequences on the circle which originate from distal parts of the genome. We have added a new track in panel f (now **Figure 4f**) to indicate duplicated regions. The grey track in the DNA ring indicates gene loci and has now been annotated in the figure legend.

5) Lines 151-152: "DNA interactions occurred between the MYC locus and two co amplified enhancer regions located 13Mbp away on the linear genome". Does the authors referring distance based on the reference genome? If so, it will also be important to describe their physical distance on the re-arranged episome. Presumably the two enhancer regions are those marked as ZC3H3 and PTP4A3 on figure 2c; however, the It is hard to recognize these regions as enhancers given the difficult representation in Figure 2b.

Yes, we mean 13Mbp on the reference, and have added text noting the distance on the ecDNA (75 and 800kbp). We hope that these text revisions and the figure revision (see also M.P. 3.4) have improved clarity.

6) Line 167-168: "The Hi-C data revealed a myriad of intra- and inter- chromosomal interactions of the

MYC promoter with co-amplified regulatory elements of chr8 and chr14 (Fig. 2f).” Given the duplications of PVT1-related regions on the ecDNA, would it be possible that some of the interactions were caused by the ambiguity in determining the contact sites amongst the duplicated regions?

Indeed, because bulk Hi-C cannot distinguish between interactions involving duplicated regions, we have annotated all such interactions to both duplicate copies in **Fig. 4f**. These duplicate regions have been annotated in the figure panel as suggested in M.P. 3.4. Motivated by the reviewer's comment, we add the following clarification in the main text (ll. 252-254):

On chr8, these loci included two accessible regions of a known *MYC* superenhancer⁶⁰ and the *PVT1* promoter. In D458, much of the superenhancer is duplicated internally on the ecDNA, and *PVT1* is amplified in D458 but not in D283.

7) Line 168-169: “Thus, we present 4 MB samples in which aberrant enhancer-promoter interactions occur across structural breakpoints on ecDNA (Fig. 2g).” Figure 2g does not show 4 MB samples. It is a FISH image of D283 showing HSR MYC amplification. Similarly, Fig. 2h on lines 176 and 177 should be changed to Fig. 2g

We apologize for this inaccuracy and have corrected the figure and legends.

8) Figure 2h: the enrichment on *OTX2* last exon in D283 is perplexing as chr14 is not involved in D283 HSR. The authors should provide an explanation.

OTX2 is a known marker for Group 3 medulloblastoma and is highly expressed in both D458 and D283 (new **Fig. 4e** in the revised manuscript and **Response Fig. 5b** above). Thus, the molecular basis for reduced growth upon CRISPR inhibition of the last *OTX2* exon in D283 may likely be explained by D283-specific transcriptional regulation of *OTX2*. We have added new panel **Fig. 4e** showing *MYC* and *OTX2* transcription in both lines.

9) Supplementary Fig. 3h: in the description of line 196-197, the gap on ecDNA2 was referred as peritelomeric while the region is labeled as pericentromeric. Please clarify whether this is a centromeric or telomeric region.

The ends of the ecDNA2 assembly (now named amp2 in the revised manuscript) map to a pericentromeric and a peritelomeric region of chr7 at either end. However, amp2 appears extrachromosomal in metaphase FISH imaging (**Fig. 2c**), and the copy number distribution per cell resembles ecDNAs (**Fig. 3a-b**). We have now updated the text to show that one end maps to a pericentromeric and the other end to a peritelomeric region of chr7.

10) Line 202-203: “These data also revealed low-level amplification of other segments of chr7 and

chr17, of 203 total length 35.2Mbp, which we hypothesized may be a third low-copy ecDNA (ecDNA3 (Supplementary fig. 6b).)”—it is not clear how a 3rd ecDNA can be inferred from Sup. Fig. 6b. The authors need to provide more description on how this was inferred in figure legend.

We agree that the assertion was speculative, given that we were not able to reconstruct the sequence of this amplification. Rather, its relative length (35Mbp) and low copy (<10 per cell) distinguish it from the ecDNA amplifications we describe. We have adjusted our language to describe “a low-copy gain (gain1) of unknown architecture composed of other segments of chr7 (35Mbp) and chr17 (800kbp),” which we feel is more appropriate.

11) Figure 4: ecDNA2-only ATAC-seq are not shown in Figure 4a. Is this due to the low number of this population? This should be mentioned in figure legend as well as in the description at line 218-219 as the current narrative would have left the misconception that edDNA1-edDNA2+ is no a minor population.

We have updated the Figure (now **Fig. 3c**) to include an example of an amp1- amp2+ cell. Although there are relatively few of these (**Fig. 3e**), we agree that an example is illustrative of the cells classified as such.

12) Figure 4: what does MT-high tumor stands for? Mitochondria?

MT indeed stands for mitochondria. We observed high differential expression of mitochondrial genes in this cell cluster, even after removing low-quality cells according to standard best practices. These cells may represent apoptotic cells, cells under oxidative stress, or heterogeneity in sequencing quality even after QC preprocessing. We have updated the text to describe these cells in more detail.

13) Line 328-329 “WGS was preprocessed and aligned according to internal pipelines at St Jude (hg38)”. The analysis pipeline was described in the Method section of McLeod et al, 2021 (ref 28). Please revise the statement as “WGS was preprocessed and aligned by bwa-mem to hg38 as described in McLeod et al”.

We appreciate this reference and have updated the text in the revised manuscript accordingly.

14) Line 396 and 397. St. Jude Pediatric Cancer Genome Project (PCPG)—abbreviation should be PCGP

We thank the reviewer for catching the typo and have corrected it in the revision.

15) The presentation of intra-tumor heterogeneity of ecDNA should mention some earlier work such

as those published by Ke et al (Structure and evolution of double minutes in diagnosis and relapse brain tumors, PMID 30267146)

The study by Ke *et al.* is of high relevance to this study, and we now cite it in our revised manuscript (ll. 312-313).

Final Decision Letter:

22nd Sep 2023

Dear Dr Chavez,

I am delighted to say that your manuscript "Circular extrachromosomal DNA promotes tumor heterogeneity in high-risk medulloblastoma" has been accepted for publication in an upcoming issue of Nature Genetics.

Your paper will be published online after we receive your corrections and will appear in print in the next available issue. You can find out your date of online publication by contacting the Nature Press Office (press@nature.com) after sending your e-proof corrections. Now is the time to inform your Public Relations or Press Office about your paper, as they might be interested in promoting its publication. This will allow them time to prepare an accurate and satisfactory press release. Include your manuscript tracking number (NG-A60243R2) and the name of the journal, which they will need when they contact our Press Office.

Please note that *Nature Genetics* is a Transformative Journal (TJ). Authors may publish their research with us through the traditional subscription access route or make their paper immediately open access through payment of an article-processing charge (APC). Authors will not be required to make a final decision about access to their article until it has been accepted. [Find out more about Transformative Journals](https://www.springernature.com/gp/open-research/transformative-journals)

Authors may need to take specific actions to achieve [compliance](https://www.springernature.com/gp/open-research/funding/policy-compliance-faqs) with funder and institutional open access mandates. If your research is supported by a funder that requires immediate open access (e.g. according to [Plan S principles](https://www.springernature.com/gp/open-research/plan-s-compliance)) then you should select the gold OA route, and we will direct you to the compliant route where possible. For authors selecting the subscription publication route, the journal's standard licensing terms will need to be accepted, including [self-archiving-and-license-to-publish](https://www.nature.com/nature-portfolio/editorial-policies/self-archiving-and-license-to-publish). Those licensing terms will supersede any other terms that the author or any third party may assert apply to any version of the manuscript.

If you have not already done so, we invite you to upload the step-by-step protocols used in this manuscript to the Protocols Exchange, part of our on-line web resource, natureprotocols.com. If you complete the upload by the time you receive your manuscript proofs, we can insert links in your article that lead directly to the protocol details. Your protocol will be made freely available upon publication of your paper. By participating in natureprotocols.com, you are enabling researchers to more readily reproduce or adapt the methodology you use. [Natureprotocols.com](http://natureprotocols.com) is fully searchable, providing your protocols and paper with increased utility and visibility. Please submit your protocol to <https://protocolexchange.researchsquare.com/>. After entering your nature.com username and password you will need to enter your manuscript number (NG-A60243R2). Further information can be found at <https://www.nature.com/nature-portfolio/editorial-policies/reporting-standards#protocols>

Sincerely,

Safia Danovi
Editor
Nature Genetics